# Fast Federated Learning in the Presence of Arbitrary Device Unavailability

**Xinran Gu**[*]
IIIS
Tsinghua University
gxr21@mails.tsinghua.edu.cn

**Kaixuan Huang**[*]
ECE
Princeton University
kaixuanh@princeton.edu

**Jingzhao Zhang**
EECS
Massachusetts Institute of Technology
jzhzhang@mit.edu

**Longbo Huang**[†]
IIIS
Tsinghua University
longbohuang@tsinghua.edu.cn

## Abstract

Federated Learning (FL) coordinates with numerous heterogeneous devices to collaboratively train a shared model while preserving user privacy. Despite its multiple advantages, FL faces new challenges. One challenge arises when devices drop out of the training process beyond the control of the central server. In this case, the convergence of popular FL algorithms such as FedAvg is severely influenced by the straggling devices. To tackle this challenge, we study federated learning algorithms under arbitrary device unavailability and propose an algorithm named Memory-augmented Impatient Federated Averaging (MIFA). Our algorithm efficiently avoids excessive latency induced by inactive devices, and corrects the gradient bias using the memorized latest updates from the devices. We prove that MIFA achieves minimax optimal convergence rates on non-i.i.d. data for both strongly convex and non-convex smooth functions. We also provide an explicit characterization of the improvement over baseline algorithms through a case study, and validate the results by numerical experiments on real-world datasets.

## 1 Introduction

Federated learning is a machine learning setting in which a central server coordinates with a large number of devices to collectively train a shared model [28, 34, 20, 33, 25, 26]. Practical advantages of this training scheme are mainly twofold. First, each device keeps the private data locally and hence preserves its data privacy. Second, federated learning can make use of idle computing resources and lower computation costs. Although federated learning successfully scales up with data sizes and accelerates training via more affordable computing power [43, 38, 36], the collaborative setup leads to new challenges due to large variations among individual computing devices. Our work aims to formulate and investigate the impact of device variations on FL from an optimization perspective.

In FL, a device can differ from its peers in multiple aspects [17, 25]. *First*, the data distribution and local task can be different among devices. To address the data variation, non-i.i.d. objective models were proposed and analyzed by [26, 18, 19, 42, 25]. We follow this line of work and formulate our optimization objective as a sum of stochastic functions on individual devices (See Eqn. (1)).

---

[*]Equal contribution.
[†]Corresponding author.

35th Conference on Neural Information Processing Systems (NeurIPS 2021).

A *second* variation among devices is caused by different computing and communication speeds. One natural way to formulate the variation in computation speeds is to allow asynchronous updates and model the updates as delayed responses. Lots of novel research has studied the problem with different delay models, e.g., [35, 4, 27, 11, 45, 1, 5, 13]. However, the delayed setup assumes that all devices make roughly the same number of (delayed) responses in the end. This behavior may deviate largely from the FL practice, where each device, e.g., personal cell phones, can have very different active duration when participating in the FL training, and hence make different numbers of responses. For this reason, our work aims to address this *third* discrepancy among devices caused by individual availability patterns.

The *third* device heterogeneity caused by different availability patterns is less studied in optimization for federated learning problems. In this model, instead of making a delayed response, devices can abort the training halfway, e.g., due to battery level, incoming calls, etc, and fail to return their responses upon the central server's requests [28, 7, 17]. To handle missing responses, researchers propose algorithms where the central server may collect responses from only a fraction of the devices and make updates [18, 28, 42, 25, 26, 17, 30, 14].

Previous works on collecting responses from a fraction of devices can be divided into two categories. When the response distribution is known, one could collect only the fastest responses and re-weight according to their response probability [17, 26, 30]. This model can be restrictive, as in practice, the exact distribution may not be available and may evolve. Another line of work assumes that the server can arbitrarily decide and sample a set of devices to collect responses accordingly in every communication round [18, 28, 42, 25, 14]. This model does not require knowing the response possibility. However, the response time can be very long if the selected subset contains unavailable devices.

In this work, we address the above limitations by studying federated learning in the presence of arbitrary device unavailability. Within this practical setup, we propose an algorithm that automatically accommodates for the underlying unavailability and allows patterns of the device unavailability to be non-stationary and even adversarial. Furthermore, our algorithm can achieve optimal convergence rates in the presence of device inactivity and automatically reduce to best-known rates if all devices are active. Our contributions are summarized as follows.

- We investigate the federated learning problem with a practical formulation of device participation, which does not require each device to be online according to an (either known or unknown) distribution.

- We propose the *Memory-augmented Impatient Federated Averaging* (MIFA) algorithm that is agnostic to the availability pattern. It efficiently avoids excessive latency induced by inactive devices, successfully exploits the information about the descent direction in stale and noisy gradients, and corrects the gradient bias using the memorized latest updates.

- We prove that MIFA achieves minimax optimal convergence rates $\mathcal{O}\left(\frac{\bar{\tau}_T + 1}{NKT}\right)$ for smooth, strongly convex functions, and $\mathcal{O}\left(\sqrt{\frac{\bar{\nu}+1}{NKT}}\right)$ for smooth, non-convex functions, and establish matching lower bounds. Here, $N, K$ and $T$ stand for the number of devices, local updates and communication rounds respectively. $\bar{\tau}_T$ and $\bar{\nu}$ characterize how actively devices participate in training (see formal definitions in Sections 3, 5 and 6). MIFA also achieves optimal convergence rates in the ideal case when all devices are active.

- We provide an explicit characterization of the improvement over baseline algorithms through a case study and empirically verify our results on real-world datasets.

## 2    Related work

**Federated learning.**    Federated Averaging (FedAvg) was first proposed in [28]. [26, 19, 18, 42] provided convergence analysis for FedAvg on non-i.i.d. data and quantified how data heterogeneity degrades the convergence rate. Several variants of FedAvg were designed to deal with data heterogeneity. FedProx [25] adds a proximal term to local objective functions, while FSVRG [21] and SCAFFOLD [18] employ variance reduction techniques.

One line of work focused on variations in computation capabilities among devices [39, 31, 40]. These models assume that responses are delayed but not missing. To address the missing response, some

work assumes that the server can actively sample a subset of devices to respond [18, 28, 42, 25, 14] or that the pattern of device availability is known [26, 17, 10, 30]. These results do not generalize to adversarial inactive patterns. [32] discussed the impact of device inactivity on convergence but their proposed algorithm diverges if there exists an inactive device in each round of communication. However, our setup allows adversarial patterns under certain non-distributional assumptions (see Section 5) while our proposed algorithm still achieves convergence.

**Asynchronous distributed optimization.** Our work is related to literature in the field of traditional asynchronous distributed optimization in that our proposed algorithm uses stale gradients. The problem setup for asynchronous distributed algorithms can be divided into two categories [13]. One is the shared-data (i.i.d.) setting, where all workers can access the whole dataset. In this setting, the local gradient is an unbiased estimator of the global gradient [35, 4, 27, 11, 45, 1]. In contrast, we assume each worker has non-i.i.d. data, and hence the local stochastic gradient can not be viewed as an unbiased estimator of the global gradient.

The other less studied setting in distributed optimization is the distributed-data setting (non-i.i.d.), where data are partitioned among workers. Specifically, [5] proposed an asynchronous incremental aggregated gradient algorithm that uses buffered gradients to update the global model. Unlike our setup, this algorithm evaluates full local gradients, performs only one local step, and was analyzed under the bounded delay assumption. [13] models the delay as stochastic and assumes that the server has knowledge of the distribution, but our formulation is distribution-free. [6] allows workers to perform multiple local steps and communicate with the server at different times, but the authors assume that all workers are available and compute at the same rate.

**Comparison with an independent work.** While preparing the manuscript, we were unaware of an independent work [41] that investigated the same setup and proposed a similar algorithm called `FedLaAvg`. Their main theorem established the convergence rate of $\mathcal{O}\left(\sqrt{\frac{\nu_{\max}}{N^{0.5}T}}(G^2 + \sigma^2)\right)$ for smooth and non-convex problems, where $G^2$ is the uniform upper bound for the squared norm of stochastic gradients and $\nu_{max}$ is the maximum number of inactive rounds. In comparison, we prove the minimax optimal rate of $\mathcal{O}(\sqrt{\frac{\bar{\nu}}{NKT}}\sigma^2)$ without the bounded gradient assumption, also improving $\nu_{\max}$ to $\bar{\nu}$. Furthermore, our result achieves a linear speedup in $N$ and $K$.

Apart from non-convex functions, we also derive the minimax optimal rates for strongly convex smooth functions under the mild assumption that allows for arbitrary and unbounded number of inactive rounds. Both of our results achieve linear speedups in terms of $N$ and $K$, and automatically recover the best-known rates of `FedAvg` when all devices are active. We also show that our proposed algorithm achieves acceleration over unbiased baseline algorithms in the presence of stragglers.

## 3  Problem Setup

We consider optimizing the following problem in a Federated Learning setting:

$$\min_{w\in\mathbb{R}^d} f(w) := \tfrac{1}{N}\sum_{i=1}^N f_i(w) := \tfrac{1}{N}\sum_{i=1}^N \mathbb{E}_{\xi_i}[f_i(w,\xi_i)], \tag{1}$$

where $w$ is the optimization variable, e.g., parameters of a machine learning model, $N$ is the number of participating devices, $f_i$ is the local loss function on device $i$, and $\xi_i$ describes the randomness in local data distribution.

In the ideal federated learning setup (see Figure 1 (a)), all devices return responses within similar time, and hence the central server collects all the local updates. In this case, the computation cost is usually measured by the number of local stochastic oracle evaluations, which is proportional to the number of rounds. In a delayed FL setup (see Figure 1 (b)), devices are always active upon the central server's request but may return responses with a delay. Here, all devices return almost the same number of responses in the long term.

As we discussed, the above setups do not depict a real-world scenario in which a device can have a longer inactive duration than active duration. In such cases, the communication interval is much longer than the local computation time required for each update, and each device generates an unequal number of responses [28, 17]. This motivates our setup in Figure 1 (c).

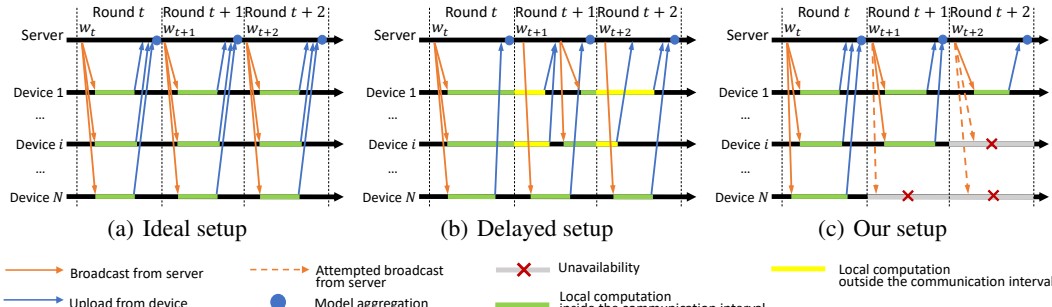

Figure 1: An illustration of setup. (a) Ideal setup: all devices return their responses within similar time. (b) Delayed setup: all devices are available, but may return responses with a delay. (c) Our setup: devices can be unavailable arbitrarily, and the communication interval is long enough for active devices to return responses.

In our proposed setup, we use $t$ to index the global communication rounds. We say a device *participates* or is *active* at round $t$ if it can complete the computation task and send back the update at the end of round $t$. We define $\mathcal{A}(t)$ as the set of all active devices at round $t$. Notice that we make no assumptions on the distribution of the participation patterns of devices and allow them to be arbitrary.

Directly applying `FedAvg` to the proposed setup can be problematic due to the existence of inactive devices. To accommodate for inactive devices, we discuss three natural variants of `FedAvg` and their limitations. The detailed algorithms can be found in Appendix A.

- Biased `FedAvg`. At each communication round, the global model is updated with a direct average of local updates from the active devices. This naive approach induces bias when data distribution and response patterns vary among devices.

- `FedAvg` with device sampling. The server selects a subset of $S$ devices randomly without replacement, and then waits until all devices in the subset $\mathcal{S}$ respond. This is how original `FedAvg` [28] addresses device unavailability. Note that over $T$ communication rounds, the global model is updated less than $T$ times due to waiting. This approach is prone to stragglers and we refer the readers to Section 5.1 for a detailed discussion.

- `FedAvg` with importance sampling [26, 13, 17]. The local updates from the active devices are weighted by the reciprocal of the participation probabilities to avoid bias. This approach is only applicable when the response of each device is i.i.d. over rounds and it requires the knowledge of participation probabilities.

## 4 Memory-augmented Impatient Federated Averaging (MIFA)

In this section, we introduce our algorithm — *Memory-augmented Impatient Federated Averaging* (MIFA). MIFA maintains an update-array $\{G^i\}$ in the memory that stores the latest updates for all devices. As the name suggests, MIFA has two components. First, the algorithm is impatient and avoids waiting for any specific device when facing heterogeneous devices with arbitrary availability. Second, the algorithm augments the received updates of the active devices with the stored updates of the inactive devices to perform averaging.

Specifically, at the beginning of round $t$, the server broadcasts the latest model parameter $w_t$ to all active devices $\mathcal{A}(t)$. After receiving $w_t$, each active device, say, the $i$-th device, sets $w_{t,0}^i = w_t$ and performs $K$ steps of SGD with respect to the local objective function to get $w_{t,K}^i$:

$$w_{t,k+1}^i = w_{t,k}^i - \eta_t \tilde{\nabla} f_i(w_{t,k}^i), \ k = 0, \cdots, K-1,$$

where $\eta_t$ is the learning rate and $\tilde{\nabla} f_i(w_{t,k}^i)$ is the stochastic gradient evaluated on device $i$. Next, the server stores the received update $\frac{1}{\eta_t}(w_t - w_{t,K}^i)$ in $G^i$. Denote by $\{G_t^i\}$ the update-array after round $t$, then we have

$$G_t^i = \begin{cases} G_{t-1}^i, & \text{if } i \notin \mathcal{A}(t), \\ \frac{1}{\eta_t}(w_t - w_{t,K}^i), & \text{if } i \in \mathcal{A}(t). \end{cases}$$

At the end of round $t$, the server updates the global model with the average of $\{G_t^i\}$ (line 9). In other words, our algorithm `MIFA` updates the model with the latest available accumulated gradients for all devices.

---

**Algorithm 1** Memory-augmented Impatient Federated Averaging (`MIFA`)

---

1: **Input**: initial $w_1$, learning rate $\{\eta_t\}$
2: **Server executes**:
3:   initialize $G^i \leftarrow 0, i \in [N]$
4: **for** $t = 1, \cdots, T - 1$ **do**
5:   broadcast $w_t$ to all active devices $i \in \mathcal{A}(t)$
6:   **for** each active device $i$ **do**
7:     $G^i \leftarrow$ DeviceUpdate$(i, w_t, \eta_t)$
8:   **end for**
9:   $w_{t+1} \leftarrow w_t - \frac{\eta_t}{N} \sum_{i=1}^N G^i$
10: **end for**

1: **DeviceUpdate**$(i, w_t, \eta_t)$:
2:   $w_{t,0}^i \leftarrow w_t$
3: **for** local step $k = 0, \cdots, K - 1$ **do**
4:   compute stochastic gradient $\tilde{\nabla} f_i(w_{t,k}^i)$
5:   $w_{t,k+1}^i \leftarrow w_{t,k}^i - \eta_t \tilde{\nabla} f_i(w_{t,k}^i)$
6: **end for**
7: Return $\frac{1}{\eta_t}(w_t - w_{t,K}^i)$ to the server

---

`MIFA` efficiently progresses without waiting for inactive devices and re-uses their latest updates as the surrogate for missing responses. Being impatient accelerates convergence, whereas memory augmentation corrects the update bias. Our algorithm differs from asynchronous algorithms in traditional distributed optimization [35, 4, 27, 11, 45, 1, 13, 6] in that we utilize the *noisy* updates of inactive devices *more than once* to avoid biasing against stragglers. In the following part of the paper, we show that `MIFA` successfully exploits information about the descent direction contained in the stale and noisy gradients.

**Discussion on implementation.** In practice, to implement `MIFA`, the server needs to maintain a huge array to store the latest update for each device, which scales with the model size and the total number of devices. To avoid exhausting the server's memory, one strategy is to distribute the memory consumption among devices. Specifically, each device, say the $i$-th, stores its previous update $G_{t_i'}^i$ computed at round $t_i'$ in its local memory. When it becomes active and computes $G_t^i$, the device sends $G_t^i - G_{t_i'}^i$ to the server, which is the difference between the current update and the previous one. In this case, the server only needs to maintain the average $\bar{G}$ in the memory and updates it by $\bar{G}_t = \bar{G}_{t-1} + \frac{1}{N} \sum_{i \in \mathcal{A}(t)}(G_t^i - G_{t_i'}^i)$ at round $t$. Then the server updates the global model by $w_{t+1} = w_t - \eta_t \bar{G}_t$.

## 5 Convergence Analysis for strongly convex objective functions

In this section, we present the convergence results for `MIFA` on $\mu$-strongly convex $L$-smooth functions. Typical examples for the strongly convex case are $\ell_2$ regularized logistic regression and linear regression problems.

In order to capture how the unavailability of devices affects algorithm performance, we introduce the following notion to quantify the dynamics of devices in our setting.

**Definition 5.1** (Number of inactive rounds). *We define the **number of inactive rounds** of device $i$ at round $t$ as $\tau(t, i) = t - \max\{t' \mid t' \leq t, i \in \mathcal{A}(t')\}$, which is the difference between current round $t$ and the latest round when device $i$ is active.*

It can be seen that $\tau(t, i) = 0$ if device $i$ is active at round $t$ and $\tau(t, i) = \tau(t - 1, i) + 1$ otherwise. Also, $t - \tau(t, i)$ is the latest round when the device $i$ is active. Next, we present the assumptions made for establishing our convergence theorem.

**Assumption 1.** $f_1, \cdots, f_N$ *are all $L$-smooth, i.e., for all $w$ and $v$, $f_i(v) \leq f_i(w) + \langle \nabla f_i(w), v - w \rangle + \frac{L}{2} \|w - v\|^2$.*

**Assumption 2.** $\tilde{\nabla} f_i(w)$ *is an unbiased estimator of $\nabla f_i$ with variance bounded by $\sigma^2$, i.e.,* $\mathbb{E}_\xi \left[ \tilde{\nabla} f_i(w) \right] = \nabla f_i(w), \mathbb{E}_\xi \left[ \left\| \tilde{\nabla} f_i(w) - \nabla f_i(w) \right\|^2 \right] \leq \sigma^2$.

**Assumption 3.** $f_1, \cdots, f_N$ *are all $\mu$-strongly convex: for all $w$ and $v$, $f_i(v) \geq f_i(w) + \langle \nabla f_i(w), v - w \rangle + \frac{\mu}{2} \|w - v\|^2$.*

**Assumption 4.** *There exists a constant $t_0 > 0$, such that for all $t \geq 1$ and $i \in [N]$, the number of inactive rounds of device $i$ at communication round $t$ satisfies $\tau(t,i) \leq t_0 + \frac{1}{b}t$, where $b = 40\left(L/\mu\right)^{1.5}$.*

Assumptions 1, 2, and 3 are standard and common in the FL literature, e.g., [26, 18, 19, 42, 36]. In Assumption 2, we relax the bounded gradient assumption that is often required in prior work, e.g., [6, 26, 40, 1]. Lastly, Assumption 4 is a very mild assumption on device availability, since it allows the number of inactive rounds to grow as $\mathcal{O}(t)$. In contrast, existing results on asynchronous updates mostly assume a bounded or fixed latency, e.g., [6, 1, 5, 40, 35, 4].

We are now ready to present our first convergence result. Define $D = \frac{1}{N}\sum_{i=1}^{N} \|\nabla f_i(w_*)\|^2$ to measure data dissimilarity, where $w_* = \arg\min f(w)$ is the global optimum. Also, define $\bar{\tau}_T$ and $\tau_{\max,T}$ to be the average and maximum numbers of inactive rounds $\tau(t,i)$ across all devices and rounds, respectively. That is,

$$\bar{\tau}_T = \frac{1}{N(T-1)}\sum_{t=1}^{T-1}\sum_{i=1}^{N}\tau(t,i), \quad \tau_{\max,T} = \max_{i\in[N]}\max_{1\leq t\leq T-1}\tau(t,i).$$

The following theorem summarizes the performance of `MIFA` in this case.

**Theorem 5.1.** *Assume that Assumptions 1 to 3 hold. Further assume that the device availability sequence $\tau(t,i)$ satisfies Assumption 4 and $\tau(1,i) = 0$ for all $i \in [N]$. By setting the learning rate $\eta_t = \frac{4}{\mu K(t+a)}$ with $a = \max\{100, 40t_0\}(L/\mu)^{1.5}$, after $T-1$ communication rounds, `MIFA` satisfies:*

$$\mathbb{E}_\xi\left[f(\overline{w}_T)\right] - f(w_*) = \mathcal{O}\left(\frac{\bar{\tau}_T + 1}{\mu NKT}\sigma^2 + \frac{\tau_{\max,T}^2 A_1 + (K-1)^2 A_2 + A_3}{\mu^2 T^2}\right),$$

*where $\overline{w}_T$ is a weighted average of $w_t$ defined as:*

$$\overline{w}_T = \frac{1}{W_T}\sum_{t=1}^{T}(t+a-1)(t+a-2)w_t, \quad W_T = \sum_{t=1}^{T}(t+a-1)(t+a-2),$$

*and $A_1 = L(D + L\sigma^2/\mu)$, $A_2 = L(D/K^2 + \sigma^2/K^3)$, $A_3 = t_0^2 L^3 \|w_1 - w_*\|^2$.*

Our results hold under Assumption 4, which allows for arbitrary device availability sequences with $\tau_{\max,T} = \mathcal{O}(T)$. However, for `MIFA` to converge, we require $\tau_{\max,T} = o(T)$ and $t_0 = o(T)$. When $T = \Omega(\frac{NK(\tau_{\max,T}^2 + t_0^2)}{\bar{\tau}+1})$, the first term dominates and the impact of the second $\mathcal{O}(1/T^2)$ term is negligible. In fact the first term in Theorem 5.1 is minimax optimal by our information-theoretic lower bound for the problem in the next proposition.

**Proposition 5.1.** *Let $c_0 > 0$ be a universal constant. For any potentially randomized algorithm, there exists a stochastic strongly convex problem satisfying Assumptions 1 to 3, such that the output $w_T$ after $T$ rounds of communication has expected sub-optimality lower bounded by*

$$\mathbb{E}[f(w_T) - f(w^*)] \geq c_0 \frac{\bar{\tau}_T \sigma^2}{\mu NKT}.$$

The proof is based on the observation that the number of gradient evaluation can scale inversely with $\bar{\tau}_T$ and that the oracle complexity is tight even for centralized stochastic optimization problems. The optimality of the first term in Theorem 5.1 is independent of the distributed or the FL setup.

The second term in Theorem 5.1 converges at the rate $\mathcal{O}(1/T^2)$ and consists of three parts, where the first part reflects the slowdown caused by device unavailability through $\tau_{\max,T}$, the second part shows the effect of multiple ($K > 1$) local steps, and the third part tells how the initial error decreases.

**Remark 5.1.** *When $\tau(t,i) = 0$ for all $i$ and $t$, our setup reduces to FedAvg with full device participation, and we have $\bar{\tau}_T = 0$ and $\tau_{\max,T} = 0$. In this case, Theorem 5.1 yields bound $\mathcal{O}\left(\frac{\sigma^2}{\mu KNT} + \frac{L(\sigma^2/K + D + L^2\|w_1 - w_*\|^2)}{\mu^2 T^2}\right)$, matching the rate $\mathcal{O}\left(\frac{\sigma^2 \log T}{\mu KNT} + \frac{L(\sigma^2/K + D)(\log T)^2}{\mu^2 T^2} + \mu \|w_1 - w_*\|^2 \exp(-\frac{\mu}{48L}T)\right)$ in [18] (Thm. V. $B^2 = 2, \eta_g = 1$) up to logarithmic terms. Besides, in the general case, our $\mathcal{O}(\tau_{\max,T}^2 A_1/T^2)$ term matches the last term in [6] (Cor. 5).*

**Remark 5.2.** *Our analysis relies on the technical assumption that all devices respond in the first round. Intuitively, this is because we need at least one valid stochastic gradient evaluation for each device to get a complete picture of the global objective, or otherwise any update would be biased. In practice, this can be achieved by waiting for the updates from all devices on $w_1$ at the very beginning.*

## 5.1 Case Study: i.i.d. Bernoulli participation

Though our algorithm can be applied to non-stationary and non-independent response patterns, we show in this subsection that even in the simple i.i.d. Bernoulli participation scenario our algorithm can achieve considerable improvement compared to known algorithms. In particular, we consider a setup where each device becomes active independently with a fixed probability $p_i$. It serves as the first motivating example towards modeling the participation patterns of devices, and provides a clean view of how the heterogeneity of the device participation influences the Federated optimization algorithms.

We will show that in this scenario, Assumption 4 holds with high probability, and the terms involving the inactive rounds $\tau(t,i)$ in Theorem 5.1 can also be bounded. Furthermore, we theoretically demonstrate that algorithms such as `FedAvg` [28] and `SCAFFOLD` [18], which sample $S$ devices for each global update, are more prone to stragglers than our algorithm.

**Definition 5.2.** *Assume that for all $i \in [N]$, the $i$-th device is assigned with a probability $p_i$. We say the participation of the devices follows **i.i.d. Bernoulli participation** model with participation probabilities $\{p_i\}$, if (1). at the first round, all devices are active, and (2). at round $t > 1$, device $i$ is active with probability $p_i$, which is independent of the history and other devices.*

Next theorem shows that under i.i.d. Bernoulli participation scenario, with high probability, $\tau(t,i)$ only grows logarithmically in $t$. Also Assumption 4 holds for a mild choice of $t_0$.

**Theorem 5.2.** *For i.i.d. Bernoulli participation model defined in Definition 5.2, given any $\delta > 0$, with probability at least $1 - \delta$, we have the following holds for all $t \geq 1$ and $i \in [N]$ simultaneously,*

$$\tau(t,i) \leq \mathcal{O}\Big(\frac{1}{p_i}(\log(Nt/\delta) + 1)\Big).$$

*Furthermore, (1). Assumption 4 holds true if $t_0 = \Omega\Big(\frac{1}{p_{min}} \log \frac{bN}{p_{min}\delta}\Big)$, where $p_{min} = \min\{p_i\}$, and $b = 40(L/\mu)^{1.5}$;(2). $\tau_{\max,T}$ can be upper bounded as*

$$\tau_{\max,T} \leq \mathcal{O}\Big(\frac{1}{p_{min}} \cdot \big(\log(TN/\delta) + 1\big)\Big).$$

The next theorem provides a high probability upper bound for $\bar{\tau}_T$.

**Theorem 5.3.** *For i.i.d. Bernoulli participation model defined in Definition 5.2, given any $\delta > 0$ and $T > 1$, with probability at least $1 - \delta$, we have*

$$\bar{\tau}_T \leq \Big(\frac{1}{N} \sum_{i=1}^{N} \frac{1}{p_i}\Big) \cdot \mathcal{O}\Big(1 + \log \frac{1}{\delta}\Big).$$

By Theorem 5.2 and Theorem 5.3, we conclude that the dominant term of our convergence bound is $\widetilde{\mathcal{O}}\Big(\frac{1}{N} \sum_{i=1}^{N} \frac{1}{p_i} \cdot \frac{\sigma^2}{\mu NKT}\Big)$. Therefore, to achieve $\epsilon$ accuracy, the dominant term of the number of the required rounds is

$$T_\epsilon^{\text{(MIFA)}} = \widetilde{\mathcal{O}}\Big(\frac{1}{N} \sum_{i=1}^{N} \frac{1}{p_i} \cdot \frac{\sigma^2}{\mu NK\epsilon}\Big). \tag{2}$$

For both `FedAvg` and `SCAFFOLD` that sample $S$ devices uniformly at random, [18] (Thm I. & III.) showed that the dominant term of the number of global updates needed to achieve $\epsilon$ accuracy is $R_\epsilon = \widetilde{\mathcal{O}}\Big(\frac{\sigma^2}{\mu SK\epsilon}\Big)$. Notice that in our setting, to accomplish each global update, the server needs to wait for a few rounds for the $S$ devices to respond. Let $T(\mathcal{S})$ be the expected rounds for which the server needs to wait for the selected devices $\mathcal{S}$ to be active. Then the expected total rounds to achieve

$\epsilon$ accuracy is $R_\epsilon \cdot \mathbb{E}_\mathcal{S}[T(\mathcal{S})]$. For i.i.d. Bernoulli participation model, we have $T(\mathcal{S}) \geq \frac{1}{\min\{p_i | i \in \mathcal{S}\}}$, and we can further show that $\mathbb{E}_\mathcal{S}[T(\mathcal{S})] \geq \frac{1}{p_{min}} \frac{S}{N}$ (see Appendix D.3 for details). Therefore,

$$\mathbb{E}\left[T_\epsilon^{(\texttt{FedAvg,SCAFFOLD})}\right] \geq \frac{S}{N} \frac{1}{p_{min}} \widetilde{\mathcal{O}}\left(\frac{\sigma^2}{\mu S K \epsilon}\right) = \widetilde{\mathcal{O}}\left(\frac{1}{p_{min}} \cdot \frac{\sigma^2}{\mu N K \epsilon}\right). \tag{3}$$

By comparing Eqn. 2 and Eqn. 3, we see that both FedAvg and SCAFFOLD are more vulnerable to stragglers, that is, the devices with very small participation probabilities; on the contrary, the convergence rate of MIFA only depends on the average of $1/p_i$ instead of $1/p_{min}$. We also provide empirical experiments showing that MIFA converges faster than FedAvg in Section 7.

## 6 Convergence result for non-convex objective functions

In this section, we present the convergence guarantee of MIFA for the non-convex case. First we list the additional assumptions as below.

**Assumption 5** (Hessian Lipschitz). *$f_1, \cdots, f_N$ are all $\rho$-Hessian Lipschitz: for all $w$ and $v$, $\left\| \nabla^2 f_i(w) - \nabla^2 f_i(v) \right\| \leq \rho \|w - v\|$.*

**Assumption 6** (Bounded noise). *The noise of the local stochastic gradients is upper bounded by a constant $\delta$ almost surely: $\left\| \tilde{\nabla} f_i(w) - \nabla f_i(w) \right\| \leq \delta$ a.s., $\forall\, i \in [N]$.*

**Assumption 7** (Bounded gradient dissimilarity). *There exist $\alpha > 0$ and $\beta_i > 0$ such that for all $w$ and $i \in [N]$: $\|\nabla f_i(w)\|^2 \leq \alpha \|\nabla f(w)\|^2 + \beta_i$. Furthermore, we define $\beta = \frac{1}{N} \sum_{i=1}^N \beta_i$.*

**Assumption 8.** *There exists a constant $\nu_i$ such that $\tau(t, i) \leq \nu_i$, for all $i \in [N]$ and $t \geq 1$. Furthermore, define $\bar{\nu} = \frac{1}{N} \sum_{i=1}^N \nu_i$ and $\nu_{\max} = \max_{i \in [N]} \nu_i$.*

The analysis of non-convex functions is much more technically involved, and our results rely on strong assumptions that provide a finer control of the gradient difference (Assumption 5), gradient noise (Assumption 6), gradient dissimilarity among devices (Assumption 7), and device unavailability (Assumption 8). We remark that Assumption 5 is also made in [9, 16], and Assumption 7 is also made in [18, 39]. We leave it as future work to study whether and how MIFA converges for non-convex functions with weaker assumptions.

**Theorem 6.1.** *Assume that Assumptions 1, 2, and 5 to 7 hold. Further assume that the device availability sequence $\tau(t, i)$ satisfies Assumption 8 and $\tau(1, i) = 0$ for all $i \in [N]$. By using a learning rate $\eta = \sqrt{\frac{N}{KTL(1+\bar{\nu})}}$, for $T \geq \max\{32\alpha LNK, 16LNK, \frac{8KN\nu_{\max}^2(L^2+\rho\delta)}{L}\}$, after $T - 1$ communication rounds, MIFA satisfies:*

$$\min_{1 \leq t \leq T} \mathbb{E}_\xi\left[\|\nabla f(w_t)\|^2\right] = \mathcal{O}\left(\sqrt{\frac{(1+\bar{\nu})L}{TKN}}(f(w_1) - f^* + \sigma^2) + \frac{A_4 + A_5}{T}\right),$$

*where $f^*$ is the optimal value, and:*

$$A_4 = NKL\left(\alpha\sigma^2\bar{\nu} + \frac{\sigma^2\nu_{\max}}{\sqrt{KN}} + \sigma\nu_{\max}\sqrt{\beta}\right) + \frac{(L^2 + \rho\delta)\sigma^2\nu_{\max}}{L},$$
$$A_5 = \frac{(K-1)NL(\beta + \sigma^2/K)}{\bar{\nu} + 1}.$$

Next, we show that the leading $\mathcal{O}(1/\sqrt{T})$ term is theoretically optimal for zero-respecting algorithms.

**Proposition 6.1.** *Let $c_0 > 0$ be a universal constant. For any randomized zero-respecting algorithm, there exists a stochastic non-convex problem satisfying Assumption1, 2, 5 and 7, such that the output $w_T$ after $T$ rounds of communication has expected sub-optimality lower bounded by*

$$\mathbb{E}[\|\nabla f(w_T)\|^2] \geq \mathbb{E}[\|\nabla f(w_T)\|]^2 \geq c_0\sqrt{\frac{\bar{\nu}L\sigma^2(f(w_0) - f^*)}{NKT}}.$$

The above proposition show that when $\sigma\sqrt{(f(w_0) - f^*)} \sim \sigma^2 + (f(w_0) - f^*)$, the result in Theorem 6.1 is tight. However, note that the counter example we used requires the quantity $\delta$ in Assumption 6 to scale with $T$, hence requiring $\delta$ to be large enough. This does not change the optimality of the first term as the first term is independent of $\delta$. Whether this requirement can be relaxed is left as an open problem.

**Remark 6.1.** *When all $\nu_i = 0$ (i.e. all the devices are active), our convergence bound reduces to* $\mathcal{O}\Big(\sqrt{\frac{L}{TKN}}(f(w_1) - f^* + \sigma^2) + \frac{(K-1)NL(\beta+\sigma^2/K)}{T}\Big)$. *This matches the result in [42] (Thm. 1, $\eta = 1, \eta_L = \sqrt{\frac{N}{KTL}}$).*

## 7 Numerical Experiments

In this section, we conduct numerical experiments[3]. to verify our theoretical results and investigate how the heterogeneity of the device availability influences the Federated optimization algorithms. We compare the performance of the following four algorithms: `FedAvg` with importance sampling (`FedAvg`-IS), Biased `FedAvg`, `FedAvg` with device sampling, and our proposed `MIFA`. For the detailed discussions of the algorithms, we refer the readers to Sections 3 and 4. We remark that for a fair comparison, we deliberately include the first few rounds that `MIFA` needs to wait to receive responses from all devices for initializing the update-array $\{G^i\}$.

Following [26, 25], we construct non-i.i.d. datasets from two commonly used computer vision datasets — MNIST [23] and CIFAR-10 [22] . Specifically, we divide the data into $N = 100$ devices with each device holding samples of only two classes, which creates a high level of data heterogeneity. For simplicity, we ensure that each device holds the same number of samples. We do not use any data augmentation. We use multinomial logistic regression as the convex model and LeNet-5 [24] with ReLU activations as the non-convex model. For all experiments, we use weight decay in the training process, which corresponds to adding $\ell_2$ penalty. We use logistic models for MNIST dataset, while we use LeNet-5 for CIFAR-10. Our code is adapted from [26], which is under MIT License.

We model the availability of the devices as independent Bernoulli random trials. The $i$-th device is assigned with a probability $p_i$, where at each time step, the device becomes active with probability $p_i$. In our experiments, the $p_i$'s are chosen such that devices holding data of smaller labels participate less frequently. Specifically, if the $i$-th device holds the data of label $j$ and $k$, we set $p_i = p_{\min} \min(j, k)/9 + (1 - p_{\min})$, where $p_{\min}$ controls the lower bound of the participation probabilities. The correlation between the participation patterns and local datasets increases the difficulty of the problem [17]. To investigate this phenomenon, we repeat the experiments for $p_{\min} = 0.1$ and $0.2$. We control the randomness of device participation when testing different algorithms.

In all the experiments, we set the initial learning rate to be $\eta_0 = 0.1$ and decay the learning rate as $\eta_t = \eta_0 \cdot \frac{1}{t}$. We set the weight decay to be $0.001$. The local batch size is 100 and each local update consists of 2 epochs. Therefore, the actual number of local steps $K$ depends on the size of the dataset. We run all the experiments with 4 GPUs of type GeForce RTX 2080 Ti. We repeat the experiments for 5 different random seeds, and all of the experiments exhibit similar training curves. We report the averaged training loss and test accuracy with error bars in Figure 2.

We observe that `FedAvg` with device sampling (FedAvg ($S = 50$) and FedAvg ($S = 100$) in Figure 2) is severely influenced by the straggling devices and makes progress relatively slowly compared to the other algorithms. Although biased `FedAvg` converges fast at the beginning, this simple algorithm is biased, and the optimality gaps are prominent for the harder CIFAR-10 dataset and when $p_{\min}$ is small. On the contrary, our proposed `MIFA` avoids waiting for stragglers, converges fast without bias, and is competitive with `FedAvg` with importance sampling, which requires knowledge of the participation probabilities. We refer the readers to Appendix G for additional experiments on CIFAR-10.

## 8 Conclusions and Discussions

In this paper, we study FL algorithms in the presence of arbitrary device unavailability and propose `MIFA`, which avoids waiting for straggling devices and re-uses the memorized latest updates as the surrogate when the device is unavailable. We theoretically analyze `MIFA` without any structural assumptions on the device availability and prove the convergence for strongly convex and non-convex smooth functions. Different from the literature that studies oracle complexity in terms of stochastic gradient evaluations, we argue that in federated learning system, the bottleneck lies in the non-stationary and possibly adversarial pattern of device participation. Therefore, it is important to study how the number of inactive rounds influences the convergence rate. In Theorem 5.1, the dependency

---

[3]Our code is available at `https://github.com/hmgxr128/MIFA_code/`

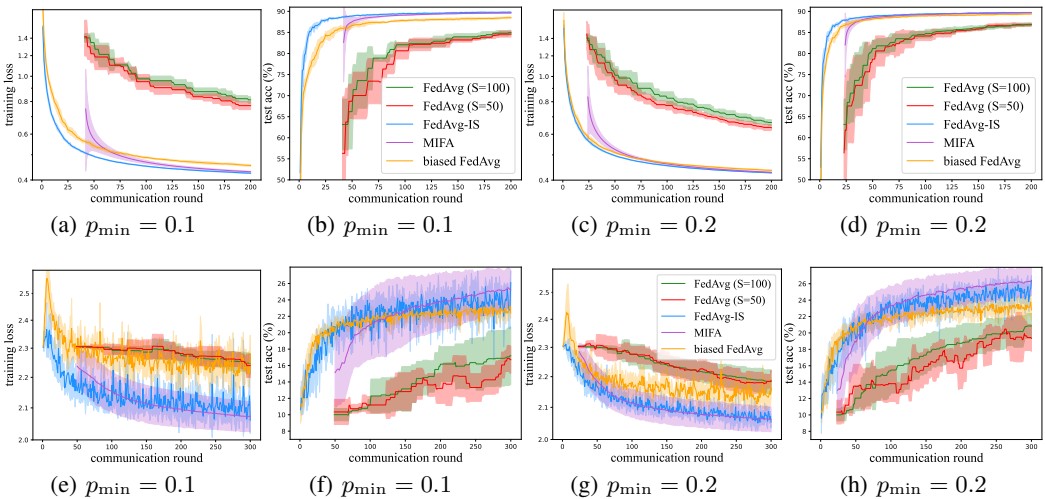

Figure 2: Training losses and test accuracies. Fig. 2(a)–2(d): logistic models on non-iid MNIST. Fig. 2(e)–2(h): LeNet-5 on non-iid CIFAR-10. FedAvg ($S = 50$) and FedAvg ($S = 100$) refer to FedAvg with device sampling that samples $S$ devices for each global update. FedAvg-IS is short for FedAvg with importance sampling, which requires knowledge of the participation probabilities.

upon $\tau_{\max,T}$ might be an artifact of our analysis, and a future direction is to study whether we can remove this dependency. Another important direction is to analyze algorithms for non-convex functions under weaker assumptions.

## Acknowledgments and Disclosure of Funding

The work of Xinran Gu and Longbo Huang is supported in part by the Technology and Innovation Major Project of the Ministry of Science and Technology of China under Grant 2020AAA0108400 and 2020AAA0108403. Xinran Gu would like to thank Kaifeng Lyu for the discussion on the convergence analysis. The authors would like to thank Sai Praneeth Karimireddy for the discussion on the problem settings.

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
