# A Baseline algorithms

The three different baseline algorithms discussed in Section 3 are summarized in the algorithm box below.

---
**Algorithm 2** FedAvg variants

---
1: **Input**: initial $w$, learning rates $\eta_t$, $t' \leftarrow 1$
2: **Server executes**:
3: **for** $t = 1, \cdots, T-1$ **do**
4:     broadcast $w$ to all active devices $i \in \mathcal{A}(t)$
5:     **for** each active device $i$ **do**
6:         $G^i \leftarrow \text{DeviceUpdate}(i, w, \eta_t)$
7:     **end for**

8:     $w \leftarrow w - \eta_t/|\mathcal{A}(t)| \cdot \sum_{i \in \mathcal{A}(t)} G^i$               biased FedAvg

9:     $w \leftarrow w - \eta_t/|\mathcal{A}(t)| \cdot \sum_{i \in \mathcal{A}(t)} \frac{1}{p_i} G^i$       FedAvg with importance sampling

10:     **if** updates from the randomly selected $S$ devices are received **then**
11:         $w \leftarrow w - \eta_{t'}/S \cdot \sum_{i \in \mathcal{S}} G^i$       FedAvg with device sampling
12:         $t' \leftarrow t' + 1$
13:     **end if**

14: **end for**

---

# B Proof of convergence for smooth and strongly convex objective functions

In this section, we analyze the convergence of MIFA for smooth and strongly convex problems. Let $\bar{\tau}_T$ be defined the same as in Section 5. Also, we introduce

$$\bar{d}_{\max,T} = \frac{1}{N} \sum_{i=1}^{N} \left[ \max_{1 \leq t \leq T-1} \tau(t,i) \right]^2,$$

which takes the maximum number of inactive rounds in round $1, \cdots, T-1$ for each device and averages its square over devices. The following theorem is a more general version of Theorem 5.1.

**Theorem B.1.** *Assume that Assumptions 1 to 3 hold. Further assume that the device availability sequence $\tau(t,i)$ satisfies Assumption 4 and $\tau(1,i) = 0$ for all $i \in [N]$. By setting the learning rate $\eta_t = \frac{4}{\mu K(t+a)}$ with $a = \max\{100, 40t_0\}(L/\mu)^{1.5}$. After $T-1$ communication rounds, MIFA satisfies:*

$$\mathbb{E}_\xi [f(\bar{w}_T)] - f(w_*) = \mathcal{O}\left( \frac{\bar{\tau}_T + 1}{\mu N K T} \sigma^2 + \frac{\bar{d}_{\max,T} A'_1 + (K-1)^2 A'_2 + A'_3}{\mu^2 T^2} \right),$$

*where $\overline{w}_T$ is a weighted average of $w_t$ defined as:*

$$\overline{w}_T = \frac{1}{W_T} \sum_{t=1}^{T} (t+a-1)(t+a-2)w_t, \quad W_T = \sum_{t=1}^{T} (t+a-1)(t+a-2),$$

*and $A'_1 = L(D + L\sigma^2/\mu)$, $A'_2 = L(D/K^2 + \sigma^2/K^3)$, $A'_3 = t_0^2 L^3 \|w_1 - w_*\|^2$.*

Note that the only difference between Theorem B.1 and Theorem 5.1 lies in $\bar{d}_{\max,T}$ and $\tau^2_{\max,T}$. Theorem B.1 yields Theorem 5.1 since $\bar{d}_{\max,T} \leq \tau^2_{\max,T}$.

## B.1 Additional notation

Define $\tilde{\eta}_t = K\eta_t$. The update rule of `MIFA` can be summarized as

$$w_{t+1} = w_t - \frac{\eta_t}{N}\sum_{k,i}\tilde{\nabla}f_i(w_{t,k}^i) = w_t - \frac{\tilde{\eta}_t}{KN}\sum_{k,i}\tilde{\nabla}f_i(w_{t-\tau(t,i),k}^i). \tag{4}$$

Further, let $e_{t,k}^i = \tilde{\nabla}f_i(w_{t,k}^i) - \nabla f_i(w_{t,k}^i)$ be the sampling noise of device $i$ at round $t$ and local step $k$. Define $\Delta_t = \mathbb{E}_\xi\left[\|w_t - w_*\|^2\right]$. Next, we introduce the following notation about device unavailability. Define $\tau_t$ and $d_t$ to be the average of the number and squared number of inactive rounds over all devices at round $t$. That is,

$$\tau_t = \frac{1}{N}\sum_{i=1}^N \tau(t,i), \quad d_t = \frac{1}{N}\sum_{i=1}^N [\tau(t,i)]^2.$$

Denote by the sum of $\tau_t$ as $s_T$, i.e., $s_T = \sum_{t=1}^{T-1}\tau_t$. Lastly, define

$$l_t = \max_{i,j}\{\tau(t,i) + \tau(t - \tau(t,i), j)\}.$$

That is, the "oldest" response used to update $w_{t-\tau(t,i)}$ into $w_t$ is received in round $t - l_t$. For convenience, all expectations in this section are taken over sampling noise $\xi$, and the summation $\sum_{k,i}$ is taken over $i = 1, \cdots, N$ and $k = 0, 1, \cdots, K - 1$.

## B.2 Preliminary lemmas

Before starting the proof, we introduce some preliminary lemmas in this subsection.

**Lemma B.1** (Property of smooth functions)**.** *For all functions $f$ that are $L$-smooth with domain $\mathcal{X}$, if $\exists \inf_{x\in\mathcal{X}} f(x) := f^*$, we have:*

$$\frac{1}{2L}\|\nabla f(x)\|^2 \le f(x) - f^*.$$

*Proof.* By definition of $L$-smoothness

$$\begin{aligned}
f^* &\le f(x - \frac{1}{L}\nabla f(x)) \\
&\le f(x) - \left\langle \nabla f(x), \frac{1}{L}\nabla f(x)\right\rangle + \frac{1}{2L}\|\nabla f(x)\|^2 \\
&= f(x) - \frac{1}{2L}\|\nabla f(x)\|^2.
\end{aligned}$$

Rearrange the terms on both sides and we complete the proof. $\square$

The following lemma bounds the norm of local gradient $\|\nabla f_i(w)\|$ by how close the $w$ is to to the global optimum $w_*$.

**Lemma B.2** (Bounding the local gradient)**.** *For all functions $f_i$ satisfying Assumptions 1 and 3, $\|\nabla f_i(w)\|^2$ can be bounded by $\|w - w_*\|^2$. That is,*

$$\|\nabla f_i(w)\|^2 \le 2L^2\|w - w_*\|^2 + 2\|\nabla f_i(w_*)\|^2.$$

*Proof.* By Jensen's inequality and $L$-smoothness, we have

$$\begin{aligned}
\|\nabla f_i(w)\|^2 &\le 2\|\nabla f_i(w) - \nabla f_i(w_*)\|^2 + 2\|\nabla f_i(w_*)\|^2 \\
&\le 2L^2\|w - w_*\|^2 + 2\|\nabla f_i(w_*)\|^2.
\end{aligned}$$

$\square$

The following lemma comes from Lemma 5 in [18].

**Lemma B.3** (Perturbed strong convexity). *The following holds for any $L$-smooth and $\mu$-strongly convex function $f$ and any $x, y, z$ in the domain of $f$:*

$$\langle \nabla f(x), z - y \rangle \geq f(z) - f(y) + \frac{\mu}{4} \|y - z\|^2 - L \|z - x\|^2 .$$

*Proof.* In order for the paper to be self-contained, we restate the proof here.

By smoothness:

$$f(z) \leq f(x) + \langle \nabla f(x), z - x \rangle + \frac{L}{2} \|z - x\|^2 \Rightarrow \langle \nabla f(x), z - x \rangle \geq f(z) - f(x) - \frac{L}{2} \|z - x\|^2 .$$

By strong convexity:

$$f(y) \geq f(x) + \langle \nabla f(x), y - x \rangle + \frac{\mu}{2} \|y - x\|^2 \Rightarrow \langle \nabla f(x), x - y \rangle \geq f(x) - f(y) + \frac{\mu}{2} \|y - x\|^2 .$$

Combining the above inequalities, we have:

$$\langle \nabla f(x), z - y \rangle \geq f(z) - f(y) - \frac{L}{2} \|z - x\|^2 + \frac{\mu}{2} \|y - x\|^2 .$$

By triangle inequality:

$$\|y - x\|^2 \geq \frac{1}{2} \|y - z\|^2 - \|x - z\|^2 .$$

Thus,

$$\langle \nabla f(x), z - y \rangle \geq f(z) - f(y) + \frac{\mu}{4} \|y - z\|^2 - \frac{L + \mu}{2} \|x - z\|^2$$

$$\geq f(z) - f(y) + \frac{\mu}{4} \|y - z\|^2 - L \|x - z\|^2 ,$$

where the second inequality only uses $L \geq \mu$. $\qquad\qquad\square$

The following lemma is slightly modified from Lemma 8 in [18].

**Lemma B.4** (Bounded drift for strongly convex and smooth objective functions). *For all $K \geq 1$ and $0 \leq k \leq K - 1$, when $\tilde{\eta}_t \leq \frac{1}{10L}$, we have bounded drift:*

$$\mathbb{E}\left[\left\|w_{t,k}^i - w_t\right\|^2\right] \leq \frac{8 \tilde{\eta}_t^2 L^2 (K-1)}{K} \Delta_t + \frac{8(K-1)\tilde{\eta}_t^2}{K} \|\nabla f_i(w_*)\|^2 + \frac{2(K-1)\tilde{\eta}_t^2 \sigma^2}{K^2} .$$

*Proof.* For $K = 1$, the bound trivially holds since $w_{t,0}^i = w_t$. For $K \geq 2$,

$$\mathbb{E}\left[\left\|w_{t,k}^i - w_t\right\|^2\right] = \mathbb{E}\left[\left\|w_{t,k-1}^i - w_t - \eta_t \tilde{\nabla} f_i(w_{t,k-1}^i)\right\|^2\right]$$

$$= \mathbb{E}\left[\left\|w_{t,k-1}^i - w_t - \eta_t \nabla f_i(w_{t,k-1}^i)\right\|^2\right] + \eta_t^2 \sigma^2$$

$$\leq \left(1 + \frac{1}{K-1}\right) \mathbb{E}\left[\left\|w_{t,k-1}^i - w_t\right\|^2\right] + K \eta_t^2 \mathbb{E}\left[\left\|\nabla f_i(w_{t,k-1}^i)\right\|^2\right] + \eta_t^2 \sigma^2$$

$$\leq \left(1 + \frac{1}{K-1}\right) \mathbb{E}\left[\left\|w_{t,k-1}^i - w_t\right\|^2\right] + 2K \eta_t^2 \mathbb{E}\left[\left\|\nabla f_i(w_t)\right\|^2\right]$$

$$\quad + 2K \eta_t^2 \mathbb{E}\left[\left\|\nabla f_i(w_{t,k-1}^i) - \nabla f_i(w_t)\right\|^2\right] + \eta_t^2 \sigma^2$$

$$\leq \left(1 + \frac{1}{K-1} + 2KL^2 \eta_t^2\right) \mathbb{E}\left[\left\|w_{t,k-1}^i - w_t\right\|^2\right]$$

$$\quad + 2K \eta_t^2 \mathbb{E}\left[\left\|\nabla f_i(w_t)\right\|^2\right] + \eta_t^2 \sigma^2 .$$

The first inequality uses $\|x + y\|^2 \leq (1 + \frac{1}{\nu}) \|x\|^2 + (1 + \nu) \|y\|^2$, $\forall \nu > 0$ with $\nu = \frac{1}{K-1}$. For $\tilde{\eta}_t \leq \frac{1}{10L}$, i.e., $\eta_t \leq \frac{1}{10KL}$, we have $2KL^2 \eta_t^2 \leq \frac{1}{50(K-1)}$. Plug in the definition of $\tilde{\eta}_t$, we have

$$\underbrace{\mathbb{E}\left[\left\|w_{t,k}^i - w_t\right\|^2\right]}_{Y_k} \leq \underbrace{\left(1 + \frac{51}{50(K-1)}\right)}_{h_1} \underbrace{\mathbb{E}\left[\left\|w_{t,k-1}^i - w_t\right\|^2\right]}_{Y_{k-1}} + \underbrace{\frac{2\tilde{\eta}_t^2}{K} \mathbb{E}\left[\left\|\nabla f_i(w_t)\right\|^2\right] + \frac{\tilde{\eta}_t^2 \sigma^2}{K^2}}_{h_2} .$$

Unrolling the recursion $Y_k \leq h_1 Y_{k-1} + h_2$, where $Y_0 = 0$, we have

$$Y_k \leq h_1^k Y_0 + h_2 \sum_{j=0}^{k-1} h_1^j = \frac{h_2(h_1^k - 1)}{h_1 - 1} \leq \frac{h_2(h_1^{K-1} - 1)}{h_1 - 1}.$$

Since $h_1^{K-1} = (1 + \frac{51}{50(K-1)})^{\frac{50(K-1)}{51} \cdot \frac{51}{50}} \leq \exp(\frac{51}{50}) < 3$ and $h_1 - 1 = \frac{51}{50(K-1)} > \frac{1}{K-1}$, plugging in the value of $h_2$, we have

$$\mathbb{E}\left[\|w_{t,k}^i - w_t\|^2\right] \leq 2(K-1)\left(\frac{2\tilde{\eta}_t^2}{K}\mathbb{E}\left[\|\nabla f_i(w_t)\|^2\right] + \frac{\tilde{\eta}_t^2 \sigma^2}{K^2}\right) \tag{5}$$

$$\leq \frac{8\tilde{\eta}_t^2 L^2(K-1)}{K}\Delta_t + \frac{8(K-1)\tilde{\eta}_t^2}{K}\|\nabla f_i(w_*)\|^2 + \frac{2(K-1)\tilde{\eta}_t^2 \sigma^2}{K^2},$$

where the second inequality uses Lemma B.2. $\qquad\square$

### B.3 The descent lemma for smooth and strongly convex problems

In this subsection, we state the descent lemma and provide a proof.

**Lemma B.5** (Descent lemma for smooth and strongly convex problems)**.** *Assume that Assumptions 1 to 3 hold. Further assume that $\tau(1, i) = 0$ for all $i \in [N]$. For any learning rate satisfying $\eta_t \leq \frac{1}{25KL}$, i.e., $\tilde{\eta}_t \leq \frac{1}{25L}$, the updates of* MIFA *satisfy:*

$$\Delta_{t+1} \leq \left(1 - \frac{1}{2}\mu\tilde{\eta}_t\right)\Delta_t - \frac{44}{25}\tilde{\eta}_t\left(\mathbb{E}[f(w_t)] - f(w_*)\right)$$

$$+ \frac{2\tilde{\eta}_t\sigma^2}{KN^2}\sum_{i=1}^{N}\sum_{j=t-\tau(t,i)}^{t-1}\tilde{\eta}_j + \frac{3\tilde{\eta}_t^2\sigma^2}{KN} + \frac{53}{50}\mathcal{H} + \mathcal{SQ}, \tag{6}$$

*where*

$$\mathcal{H} = \frac{64L^3(K-1)^2}{K^2 N}\tilde{\eta}_t\sum_{i=1}^{N}\tilde{\eta}_{t-\tau(t,i)}^2\Delta_{t-\tau(t,i)} + \frac{16L^3}{N^2}\tilde{\eta}_t\sum_{i=1}^{N}\tau(t,i)\left(\sum_{j=t-\tau(t,i)}^{t-1}\sum_{i'=1}^{N}\tilde{\eta}_j^2\Delta_{j-\tau(j,i)}\right)$$

$$+ \frac{16LD}{N}\tilde{\eta}_t\sum_{i=1}^{N}\tau(t,i)\left(\sum_{j=t-\tau(t,i)}^{t-1}\tilde{\eta}_j^2\right) + \frac{64(K-1)^2L}{K^2 N}\tilde{\eta}_t\sum_{i=1}^{N}\tilde{\eta}_{t-\tau(t,i)}^2\|\nabla f_i(w_*)\|^2$$

$$+ \frac{16(K-1)^2L\sigma^2}{K^3 N}\tilde{\eta}_t\sum_{i=1}^{N}\tilde{\eta}_{t-\tau(t,i)}^2 + \frac{8L\sigma^2}{KN}\tilde{\eta}_t\sum_{i=1}^{N}\tau(t,i)\left(\sum_{j=t-\tau(t,i)}^{t-1}\tilde{\eta}_j^2\right)$$

$$+ \frac{64L^5(K-1)^2}{K^2 N^2}\tilde{\eta}_t\sum_{i=1}^{N}\tau(t,i)\left(\sum_{j=t-\tau(t,i)}^{t-1}\sum_{i'=1}^{N}\tilde{\eta}_j^2\tilde{\eta}_{j-\tau(j,i')}^2\Delta_{j-\tau(j,i')}\right)$$

$$+ \frac{64L^3(K-1)^2}{K^2 N^2}\tilde{\eta}_t\sum_{i=1}^{N}\tau(t,i)\left(\sum_{j=t-\tau(t,i)}^{t-1}\sum_{i'=1}^{N}\tilde{\eta}_j^2\tilde{\eta}_{j-\tau(j,i')}^2\|\nabla f_{i'}(w_*)\|^2\right)$$

$$+ \frac{16L^3(K-1)^2\sigma^2}{K^3 N^2}\tilde{\eta}_t\sum_{i=1}^{N}\tau(t,i)\left(\sum_{j=t-\tau(t,i)}^{t-1}\tilde{\eta}_j^2\sum_{i'=1}^{N}\tilde{\eta}_{j-\tau(j,i')}^2\right),$$

*and*

$$\mathcal{SQ} = \frac{2\sigma L\tilde{\eta}_t}{N}\sum_{i=1}^{N}\sqrt{\frac{\tau(t,i)}{N}\sum_{j=t-\tau(t,i)}^{t-1}\tilde{\eta}_j^2\left(\sum_{i'=1}^{N}\Delta_{j-\tau(j,i')}\right)} +$$

$$\frac{2\sigma L\tilde{\eta}_t}{N}\sum_{i=1}^{N}\sqrt{\frac{\tau(t,i)(K-1)^2}{K^2 N}\sum_{j=t-\tau(t,i)}^{t-1}\tilde{\eta}_j^2\sum_{i'=1}^{N}\tilde{\eta}_{j-\tau(j,i')}^2\left(8L^2\Delta_{j-\tau(j,i')} + 8\|\nabla f_{i'}(w_*)\|^2 + \frac{2\sigma^2}{K}\right)}.$$

**Proof of the descent lemma.** According to the update rule in (4), we can expand $\|w_{t+1} - w_*\|^2$ as

$$\|w_{t+1} - w_*\|^2 = \left\| w_t - w_* - \frac{\tilde{\eta}_t}{KN} \sum_{k,i} \tilde{\nabla} f_i(w^i_{t-\tau(t,i),k}) \right\|^2$$

$$= \|w_t - w_*\|^2 \underbrace{- \frac{2\tilde{\eta}_t}{KN} \sum_{k,i} \left\langle \tilde{\nabla} f_i(w^i_{t-\tau(t,i),k}), w_t - w_* \right\rangle}_{\mathcal{A}_1}$$

$$+ \underbrace{\frac{\tilde{\eta}_t^2}{K^2 N^2} \left\| \sum_{k,i} \tilde{\nabla} f_i(w^i_{t-\tau(t,i),k}) \right\|^2}_{\mathcal{A}_2}.$$

To bound the expectation of $\|w_{t+1} - w_*\|^2$, we bound expectations of $\mathcal{A}_1$ and $\mathcal{A}_2$ respectively.

### B.3.1 Bounding the first term

Note that $\tilde{\nabla} f_i(w^i_{t-\tau(t,i),k})$ can be expanded as $\nabla f_i(w^i_{t-\tau(t,i),k}) + e^i_{t-\tau(t,i),k}$. Thus, $\mathcal{A}_1$ can be split as

$$\mathcal{A}_1 = -\frac{2\tilde{\eta}_t}{KN} \sum_{k,i} \left\langle \nabla f_i(w^i_{t-\tau(t,i),k}), w_t - w_* \right\rangle - \frac{2\tilde{\eta}_t}{KN} \sum_{k,i} \left\langle e^i_{t-\tau(t,i),k}, w_t - w_* \right\rangle.$$

Due to reuse of noisy updates, $e^i_{t-\tau(t,i),k}$ is correlated with $w_t$ and $\mathbb{E}\left[\left\langle e^i_{t-\tau(t,i),k}, w_t - w_* \right\rangle\right]$ is not necessarily zero. Further expanding $w_t - w_*$ as $(w_t - w^i_{t-\tau(t,i),k}) + (w^i_{t-\tau(t,i),k} - w_*)$, we obtain

$$\mathcal{A}_1 = \underbrace{-\frac{2\tilde{\eta}_t}{KN} \sum_{k,i} \left\langle \nabla f_i(w^i_{t-\tau(t,i),k}), w_t - w_* \right\rangle}_{\mathcal{B}_1} \underbrace{- \frac{2\tilde{\eta}_t}{KN} \sum_{k,i} \left\langle e^i_{t-\tau(t,i),k}, w_t - w^i_{t-\tau(t,i),k} \right\rangle}_{\mathcal{B}_2}$$

$$\underbrace{- \frac{2\tilde{\eta}_t}{KN} \sum_{k,i} \left\langle e^i_{t-\tau(t,i),k}, w^i_{t-\tau(t,i),k} - w_* \right\rangle}_{\mathcal{B}_3}.$$

Due to independence of $e^i_{t-\tau(t,i),k}$ and $w^i_{t-\tau(t,i),k}$, we have $\mathbb{E}[\mathcal{B}_3] = 0$. By Lemma B.3,

$$\mathcal{B}_1 \leq -2\tilde{\eta}_t \left(f(w_t) - f(w_*)\right) - \frac{\mu \tilde{\eta}_t}{2} \|w_t - w_*\|^2 + \underbrace{\frac{2L\tilde{\eta}_t}{KN} \sum_{k,i} \left\| w^i_{t-\tau(t,i),k} - w_t \right\|^2}_{\mathcal{C}_1}.$$

To estimate the bound for $\mathcal{C}_1$, we take a closer look at one summand of $\mathcal{C}_1$. Note that $\left(w^i_{t-\tau(t,i),k} - w_t\right)$ can be split in the following way.

$$w^i_{t-\tau(t,i),k} - w_t$$

$$= w^i_{t-\tau(t,i),k} - w_{t-\tau(t,i)} + w_{t-\tau(t,i)} - w_t$$

$$= w^i_{t-\tau(t,i),k} - w_{t-\tau(t,i)} - \frac{1}{KN} \sum_{j=t-\tau(t,i)}^{t-1} \tilde{\eta}_j \sum_{k',i'} \tilde{\nabla} f_{i'}(w^{i'}_{j-\tau(j,i'),k'})$$

$$= w^i_{t-\tau(t,i),k} - w_{t-\tau(t,i)} - \frac{1}{KN} \sum_{j=t-\tau(t,i)}^{t-1} \tilde{\eta}_j \sum_{k',i'} \left(\nabla f_{i'}(w^{i'}_{j-\tau(j,i'),k'}) - \nabla f_{i'}(w_{j-\tau(j,i')})\right)$$

$$- \frac{1}{KN} \sum_{j=t-\tau(t,i)}^{t-1} \tilde{\eta}_j \sum_{i'=1}^{N} \nabla f_{i'}(w_{j-\tau(j,i')}) - \frac{1}{N} \sum_{j=t-\tau(t,i)}^{t-1} \tilde{\eta}_j \sum_{k',i'} e^{i'}_{j-\tau(j,i')}.$$

By Jensen's inequality, we expand $\left\| w_{t-\tau(t,i),k}^i - w_t \right\|^2$ as four parts.

$$\left\| w_{t-\tau(t,i),k}^i - w_t \right\|^2 \le 4 \underbrace{\left\| w_{t-\tau(t,i),k}^i - w_{t-\tau(t,i)} \right\|^2}_{\mathcal{D}_1}$$

$$+ 4 \underbrace{\left\| \frac{1}{KN} \sum_{j=t-\tau(t,i)}^{t-1} \tilde{\eta}_j \left[ \sum_{k',i'} \left( \nabla f_{i'}(w_{j-\tau(j,i'),k'}^{i'}) - \nabla f_{i'}(w_{j-\tau(j,i')}) \right) \right] \right\|^2}_{\mathcal{D}_2}$$

$$+ 4 \underbrace{\left\| \frac{1}{KN} \sum_{j=t-\tau(t,i)}^{t-1} \tilde{\eta}_j \sum_{k',i'} \nabla f_{i'}(w_{j-\tau(j,i')}) \right\|^2}_{D_3}$$

$$+ 4 \underbrace{\left\| \frac{1}{KN} \sum_{j=t-\tau(t,i)}^{t-1} \tilde{\eta}_j \sum_{k',i'} e_{j-\tau(j,i'),k'}^{i'} \right\|^2}_{\mathcal{D}_4}.$$

According to Lemma B.4, for $k = 0$, $\mathcal{D}_1 = 0$. For $k \ge 1$,

$$\mathbb{E}\left[\mathcal{D}_1\right] \le \frac{32L^2(K-1)\tilde{\eta}_{t-\tau(t,i)}^2}{K}\Delta_{t-\tau(t,i)}$$
$$+ \frac{32(K-1)\tilde{\eta}_{t-\tau(t,i)}^2}{K}\left\|\nabla f_i(w_*)\right\|^2 + \frac{8(K-1)\tilde{\eta}_{t-\tau(t,i)}^2\sigma^2}{K^2}.$$

Repeatedly applying Jensen's inequality and further using $L$-smoothness,

$$\mathbb{E}\left[\mathcal{D}_2\right] \le \frac{4\tau(t,i)}{K^2N^2} \sum_{j=t-\tau(t,i)}^{t-1} \tilde{\eta}_j^2 \left\| \sum_{k',i'} \left( \nabla f_{i'}(w_{j-\tau(j,i'),k'}^{i'}) - \nabla f_{i'}(w_{j-\tau(j,i')}) \right) \right\|^2$$

$$\le \frac{4\tau(t,i)}{KN} \sum_{j=t-\tau(t,i)}^{t-1} \tilde{\eta}_j^2 \sum_{k',i'} \left\| \nabla f_{i'}(w_{j-\tau(j,i'),k'}^{i'}) - \nabla f_{i'}(w_{j-\tau(j,i')}) \right\|^2$$

$$\le \frac{4L^2\tau(t,i)}{KN} \sum_{j=t-\tau(t,i)}^{t-1} \tilde{\eta}_j^2 \sum_{k',i'} \left\| w_{j-\tau(j,i'),k'}^{i'} - w_{j-\tau(j,i')} \right\|^2.$$

By Lemma B.4,

$$\mathbb{E}\left[\mathcal{D}_2\right] \le \frac{32L^4\tau(t,i)(K-1)^2}{K^2N} \sum_{j=t-\tau(t,i)}^{t-1} \tilde{\eta}_j^2 \sum_{i'=1}^{N} \tilde{\eta}_{j-\tau(j,i')}^2 \Delta_{j-\tau(j,i')}$$

$$+ \frac{32L^2\tau(t,i)(K-1)^2}{K^2N} \sum_{j=t-\tau(t,i)}^{t-1} \tilde{\eta}_j^2 \sum_{i'=1}^{N} \tilde{\eta}_{j-\tau(j,i')}^2 \left\|\nabla f_{i'}(w_*)\right\|^2$$

$$+ \frac{8L^2\tau(t,i)(K-1)^2\sigma^2}{K^3N} \sum_{j=t-\tau(t,i)}^{t-1} \tilde{\eta}_j^2 \sum_{i'=1}^{N} \tilde{\eta}_{j-\tau(j,i')}^2.$$

Expanding $\mathcal{D}_3$ by Jensen's inequality and applying Lemma B.2,

$$\mathbb{E}\left[\mathcal{D}_3\right] \le \frac{4\tau(t,i)}{KN} \sum_{j=t-\tau(t,i)}^{t-1} \tilde{\eta}_j^2 \sum_{k',i'} \left\|\nabla f_{i'}(w_{j-\tau(j,i)})\right\|^2$$

$$\le \frac{8\tau(t,i)L^2}{N} \sum_{j=t-\tau(t,i)}^{t-1} \tilde{\eta}_j^2 \sum_{i'=1}^{N} \Delta_{j-\tau(j,i)} + 8\tau(t,i)D \sum_{j=t-\tau(t,i)}^{t-1} \tilde{\eta}_j^2.$$

Due to independence of $e^i_{j,k}$ and $e^{i'}_{j,k'}$ for $i' \neq i$ or $k \neq k'$, $\mathbb{E}\left[\left\|\sum_{k',i'} e^{i'}_{j-\tau(j,i'),k'}\right\|^2\right] \leq KN\sigma^2$.

Still by Jensen's inequality, the expectation of $\mathcal{D}_4$ can be bounded as follows.

$$\mathbb{E}\left[\mathcal{D}_4\right] \leq \frac{4\tau(t,i)}{K^2N^2} \sum_{j=t-\tau(t,i)}^{t-1} \left\|\tilde{\eta}_j \sum_{k',i'} e^{i'}_{j-\tau(j,i'),k'}\right\|^2 \leq \frac{4\tau(t,i)\sigma^2}{KN} \sum_{j=t-\tau(t,i)}^{t-1} \tilde{\eta}_j^2.$$

Intuitively, $\mathcal{D}_1$ quantifies the drift induced by multiple local steps. $\mathcal{D}_3$ and $\mathcal{D}_4$ correspond to errors caused by inactivity. $\mathcal{D}_2$ is induced by both local steps and inactivity. Note that $\mathcal{D}_2$ to $\mathcal{D}_4$ vanish when $\tau(t,i) = 0$ and $\mathcal{D}_1$ and that $\mathcal{D}_2$ vanish when $K = 1$. Combining the expectation of $\mathcal{D}_1$ to $\mathcal{D}_4$, we have

$$\mathbb{E}\left[\left\|w^i_{t-\tau(t,i),k} - w_t\right\|^2\right]$$

$$\leq \frac{32L^2(K-1)^2\tilde{\eta}_{t-\tau(t,i)}^2}{K^2}\Delta_{t-\tau(t,i)} + \frac{8\tau(t,i)L^2}{N} \sum_{j=t-\tau(t,i)}^{t-1} \tilde{\eta}_j^2 \sum_{i'=1}^{N} \Delta_{j-\tau(j,i)}$$

$$+ 8\tau(t,i)D \sum_{j=t-\tau(t,i)}^{t-1} \tilde{\eta}_j^2 + \frac{32(K-1)^2\tilde{\eta}_{t-\tau(t,i)}^2}{K^2} \left\|\nabla f_i(w_*)\right\|^2$$

$$+ \frac{8(K-1)^2\tilde{\eta}_{t-\tau(t,i)}^2\sigma^2}{K^3} + \frac{4\tau(t,i)\sigma^2}{KN} \sum_{j=t-\tau(t,i)}^{t-1} \tilde{\eta}_j^2$$

$$+ \frac{32L^4\tau(t,i)(K-1)^2}{K^2N} \sum_{j=t-\tau(t,i)}^{t-1} \tilde{\eta}_j^2 \sum_{i'=1}^{N} \tilde{\eta}_{j-\tau(j,i')}^2 \Delta_{j-\tau(j,i')}$$

$$+ \frac{32L^2\tau(t,i)(K-1)^2}{K^2N} \sum_{j=t-\tau(t,i)}^{t-1} \tilde{\eta}_j^2 \sum_{i'=1}^{N} \tilde{\eta}_{j-\tau(j,i')}^2 \left\|\nabla f_{i'}(w_*)\right\|^2$$

$$+ \frac{8L^2\tau(t,i)(K-1)^2\sigma^2}{K^3N} \sum_{j=t-\tau(t,i)}^{t-1} \tilde{\eta}_j^2 \sum_{i'=1}^{N} \tilde{\eta}_{j-\tau(j,i')}^2.$$

Since when $i$ and $t$ are fixed, $\mathbb{E}\left[\left\|w^i_{t-\tau(t,i),k} - w_t\right\|^2\right]$ can be uniformly bounded for all $0 \leq k \leq K - 1$, we can bound the expectation of $\mathcal{C}_1$.

$$\mathbb{E}\left[\mathcal{C}_1\right]$$

$$\leq \frac{64L^3(K-1)^2}{K^2N}\tilde{\eta}_t \sum_{i=1}^{N} \tilde{\eta}_{t-\tau(t,i)}^2 \Delta_{t-\tau(t,i)} + \frac{16L^3}{N^2}\tilde{\eta}_t \sum_{i=1}^{N} \tau(t,i) \left(\sum_{j=t-\tau(t,i)}^{t-1} \sum_{i'=1}^{N} \tilde{\eta}_j^2 \Delta_{j-\tau(j,i)}\right)$$

$$+ \frac{16LD}{N}\tilde{\eta}_t \sum_{i=1}^{N} \tau(t,i) \left(\sum_{j=t-\tau(t,i)}^{t-1} \tilde{\eta}_j^2\right) + \frac{64(K-1)^2L}{K^2N}\tilde{\eta}_t \sum_{i=1}^{N} \tilde{\eta}_{t-\tau(t,i)}^2 \left\|\nabla f_i(w_*)\right\|^2$$

$$+ \frac{16(K-1)^2L\sigma^2}{K^3N}\tilde{\eta}_t \sum_{i=1}^{N} \tilde{\eta}_{t-\tau(t,i)}^2 + \frac{8L\sigma^2}{KN}\tilde{\eta}_t \sum_{i=1}^{N} \tau(t,i) \left(\sum_{j=t-\tau(t,i)}^{t-1} \tilde{\eta}_j^2\right)$$

$$+ \frac{64L^5(K-1)^2}{K^2N^2}\tilde{\eta}_t \sum_{i=1}^{N} \tau(t,i) \left(\sum_{j=t-\tau(t,i)}^{t-1} \sum_{i'=1}^{N} \tilde{\eta}_j^2 \tilde{\eta}_{j-\tau(j,i')}^2 \Delta_{j-\tau(j,i')}\right)$$

$$+ \frac{64L^3(K-1)^2}{K^2N^2}\tilde{\eta}_t \sum_{i=1}^{N} \tau(t,i) \left(\sum_{j=t-\tau(t,i)}^{t-1} \sum_{i'=1}^{N} \tilde{\eta}_j^2 \tilde{\eta}_{j-\tau(j,i')}^2 \left\|\nabla f_{i'}(w_*)\right\|^2\right)$$

$$+ \frac{16L^3(K-1)^2\sigma^2}{K^3N^2}\tilde{\eta}_t \sum_{i=1}^{N} \tau(t,i) \left( \sum_{j=t-\tau(t,i)}^{t-1} \tilde{\eta}_j^2 \sum_{i'=1}^{N} \tilde{\eta}_{j-\tau(j,i')}^2 \right).$$

Here we denote the RHS of the above inequality as $\mathcal{H}$. Therefore,

$$\mathbb{E}\left[\mathcal{B}_1\right] \leq -\frac{1}{2}\mu\tilde{\eta}_t\Delta_t - 2\tilde{\eta}_t\left(\mathbb{E}\left[f(w_t)\right] - f(w_*)\right) + \mathcal{H}. \tag{7}$$

Next we estimate the bound for $\mathcal{B}_2$. Unrolling one summand of $\mathcal{B}_2$,

$$-\left\langle e_{t-\tau(t,i),k}^i, w_t - w_{t-\tau(t,i),k}^i \right\rangle = \underbrace{-\left\langle e_{t-\tau(t,i),k}^i, w_t - w_{t-\tau(t,i)} \right\rangle}_{\mathcal{C}_2}$$

$$\underbrace{-\left\langle e_{t-\tau(t,i),k}^i, w_{t-\tau(t,i)} - w_{t-\tau(t,i),k}^i \right\rangle}_{\mathcal{C}_3}.$$

Due to independence of $e_{t-\tau(t,i),k}^i$ and $w_{t-\tau(t,i)} - w_{t-\tau(t,i),k}^i$, $\mathbb{E}\left[\mathcal{C}_3\right] = 0$. Then we turn to $\mathcal{C}_2$,

$$\mathcal{C}_2 = \frac{1}{KN} \left\langle e_{t-\tau(t,i),k}^i, \sum_{j=t-\tau(t,i)}^{t-1} \tilde{\eta}_j \sum_{k',i'} \left( \nabla f_{i'}(w_{j-\tau(j,i'),k'-1}^{i'}) + e_{j-\tau(j,i'),k'}^i \right) \right\rangle$$

$$= \underbrace{\frac{1}{KN} \left\langle e_{t-\tau(t,i),k}^i, \sum_{j=t-\tau(t,i)}^{t-1} \tilde{\eta}_j e_{j-\tau(j,i),k}^i \right\rangle}_{\mathcal{D}_5}$$

$$+ \underbrace{\frac{1}{KN} \left\langle e_{t-\tau(t,i),k}^i, \sum_{j=t-\tau(t,i)}^{t-1} \tilde{\eta}_j \sum_{\substack{k'\neq k \\ \text{or } i'\neq i}} e_{j-\tau(j,i'),k'}^{i'} \right\rangle}_{\mathcal{D}_6}$$

$$+ \underbrace{\frac{1}{KN} \left\langle e_{t-\tau(t,i),k}^i, \sum_{j=t-\tau(t,i)}^{t-1} \tilde{\eta}_j \sum_{k',i'} \nabla f_{i'}(w_{j-\tau(j,i'),k'}^{i'}) \right\rangle}_{\mathcal{D}_7}.$$

Applying the identity $e_{t-\tau(t,i),k}^i = e_{t-1-\tau(t-1,i),k}^i = \cdots = e_{t-\tau(t,i)-\tau(t-\tau(t,i),i),k}^i$, the expectation of $\mathcal{D}_5$ can be bounded by

$$\mathbb{E}\left[\mathcal{D}_5\right] \leq \frac{\sigma^2}{KN} \sum_{j=t-\tau(t,i)}^{t-1} \tilde{\eta}_j.$$

Due to independence of $e_{j,k}^i$ and $e_{j,k'}^{i'}$ for $i' \neq i$ or $k \neq k'$, $\mathbb{E}\left[\mathcal{D}_6\right] = 0$. Note that $\nabla f_{i'}(w_{j-\tau(j,i'),k'}^{i'})$ can be split as $(\nabla f_{i'}(w_{j-\tau(j,i'),k'}^{i'}) - \nabla f_{i'}(w_{j-\tau(j,i')})) + (\nabla f_{i'}(w_{j-\tau(j,i')}) - \nabla f_{i'}(w_*))$, where the first part is the difference between the gradient on the local parameter and on the global parameter, and the second part is the difference between the gradient on the global parameter and on the global optimum. By Cauchy-Schwartz inequality $\mathbb{E}\left[\langle X, Y\rangle\right] \leq \sqrt{\mathbb{E}\left[\|X\|^2\right]\mathbb{E}\left[\|Y\|^2\right]}$, we bound the expectation of $\mathcal{D}_7$,

$$\mathbb{E}\left[\mathcal{D}_7\right]$$

$$= \frac{1}{KN}\mathbb{E}\left[\left\langle e_{t-\tau(t,i),k}^i, \sum_{j=t-\tau(t,i)}^{t-1} \tilde{\eta}_j \sum_{k',i'} \left( \nabla f_{i'}(w_{j-\tau(j,i'),k'}^{i'}) - \nabla f_{i'}(w_{j-\tau(j,i')}) \right) \right\rangle\right]$$

$$+ \frac{1}{KN}\mathbb{E}\left[\left\langle e_{t-\tau(t,i),k}^i, \sum_{j=t-\tau(t,i)}^{t-1} \tilde{\eta}_j \sum_{k',i'} \left( \nabla f_{i'}(w_{j-\tau(j,i')}) - \nabla f_{i'}(w_*) \right) \right\rangle\right]$$

$$\leq \frac{1}{KN}\sigma \sqrt{\mathbb{E}\left[\left\|\sum_{j=t-\tau(t,i)}^{t-1}\tilde{\eta}_j \sum_{k',i'}\left(\nabla f_{i'}(w^{i'}_{j-\tau(j,i'),k'}) - \nabla f_{i'}(w_{j-\tau(j,i')})\right)\right\|^2\right]}$$

$$+ \frac{1}{KN}\sigma \sqrt{\mathbb{E}\left[\left\|\sum_{j=t-\tau(t,i)}^{t-1}\tilde{\eta}_j \sum_{k',i'}\left(\nabla f_{i'}(w_{j-\tau(j,i')}) - \nabla f_{i'}(w_*)\right)\right\|^2\right]}.$$

By Jensen's inequality and $L$-smoothness, the term inside the first square root can be bounded as follows.

$$\left\|\sum_{j=t-\tau(t,i)}^{t-1}\tilde{\eta}_j \sum_{k',i'}\left(\nabla f_{i'}(w^{i'}_{j-\tau(j,i'),k'}) - \nabla f_{i'}(w_{j-\tau(j,i')})\right)\right\|^2$$

$$\leq \tau(t,i)\sum_{j=t-\tau(t,i)}^{t-1}\tilde{\eta}_j^2 \left\|\sum_{k',i'}\left(\nabla f_{i'}(w^{i'}_{j-\tau(j,i'),k'}) - \nabla f_{i'}(w_{j-\tau(j,i')})\right)\right\|^2$$

$$\leq KN\tau(t,i)\sum_{j=t-\tau(t,i)}^{t-1}\tilde{\eta}_j^2 \left(\sum_{k',i'}\left\|\nabla f_{i'}(w^{i'}_{j,k'}) - \nabla f_{i'}(w_j)\right\|^2\right)$$

$$\leq KNL^2\tau(t,i)\sum_{j=t-\tau(t,i)}^{t-1}\tilde{\eta}_j^2 \left(\sum_{k',i'}\left\|w^{i'}_{j-\tau(j,i'),k'} - w_{j-\tau(j,i')}\right\|^2\right).$$

Similarly, the term inside the second square root can be bounded as follows.

$$\left\|\sum_{j=t-\tau(t,i)}^{t-1}\tilde{\eta}_j \sum_{k',i'}\left(\nabla f_{i'}(w_{j-\tau(j,i')}) - \nabla f_{i'}(w_*)\right)\right\|^2$$

$$\leq K^2NL^2\tau(t,i)\sum_{j=t-\tau(t,i)}^{t-1}\tilde{\eta}_j^2 \left(\sum_{i'=1}^{N}\left\|w_{j-\tau(j,i')} - w_*\right\|^2\right).$$

Therefore,

$$\mathbb{E}\left[\mathcal{D}_7\right]$$

$$\leq \sigma L\sqrt{\frac{\tau(t,i)}{KN}\sum_{j=t-\tau(t,i)}^{t-1}\tilde{\eta}_j^2 \sum_{k',i'}\mathbb{E}\left[\left\|w^{i'}_{j-\tau(j,i'),k'} - w_{j-\tau(j,i')}\right\|^2\right]}$$

$$+ \sigma L\sqrt{\frac{\tau(t,i)}{N}\sum_{j=t-\tau(t,i)}^{t-1}\tilde{\eta}_j^2 \left(\sum_{i'=1}^{N}\Delta_{j-\tau(j,i')}\right)}$$

$$\leq \sigma L\sqrt{\frac{\tau(t,i)(K-1)^2}{K^2N}\sum_{j=t-\tau(t,i)}^{t-1}\tilde{\eta}_j^2 \sum_{i'=1}^{N}\tilde{\eta}_{j-\tau(j,i')}^2 \left(8L^2\Delta_{j-\tau(j,i')} + 8\left\|\nabla f_{i'}(w_*)\right\|^2 + \frac{2\sigma^2}{K}\right)}$$

$$+ \sigma L\sqrt{\frac{\tau(t,i)}{N}\sum_{j=t-\tau(t,i)}^{t-1}\tilde{\eta}_j^2 \left(\sum_{i'=1}^{N}\Delta_{j-\tau(j,i')}\right)},$$

where the second inequality uses Lemma B.4. Combining the expectation of $\mathcal{D}_5$ to $\mathcal{D}_7$, we have

$$\mathbb{E}\left[\mathcal{C}_2\right]$$

$$\leq \frac{\sigma^2}{KN}\sum_{j=t-\tau(t,i)}^{t-1}\tilde{\eta}_j + \sigma L\sqrt{\frac{\tau(t,i)}{N}\sum_{j=t-\tau(t,i)}^{t-1}\tilde{\eta}_j^2 \left(\sum_{i'=1}^{N}\Delta_{j-\tau(j,i')}\right)}$$

$$+ \sigma L \sqrt{\frac{\tau(t,i)(K-1)^2}{K^2 N} \sum_{j=t-\tau(t,i)}^{t-1} \tilde{\eta}_j^2 \sum_{i'=1}^{N} \tilde{\eta}_{j-\tau(j,i')}^2 \left(8L^2 \Delta_{j-\tau(j,i')} + 8\|\nabla f_i(w_*)\|^2 + \frac{2\sigma^2}{K}\right)}.$$

The first term can be interpreted as the accumulated noise due to reuse of noisy gradients. The expression inside the square root of the second term stands for the effect of inactivity and it vanishes when $\tau(t,i) = 0$. The expression inside the square root of the second term stands for the effect of unavailability and local updates and it vanishes when $\tau(t,i) = 0$ or $K = 1$. Then the expectation of $\mathcal{B}_2$ can be bounded by

$$\mathbb{E}\left[\mathcal{B}_2\right]$$

$$\leq \frac{2\tilde{\eta}_t \sigma^2}{KN^2} \sum_{i=1}^{N} \sum_{j=t-\tau(t,i)}^{t-1} \tilde{\eta}_j + \underbrace{\frac{2\sigma L \tilde{\eta}_t}{N} \sum_{i=1}^{N} \sqrt{\frac{\tau(t,i)}{N} \sum_{j=t-\tau(t,i)}^{t-1} \tilde{\eta}_j^2 \left(\sum_{i'=1}^{N} \Delta_{j-\tau(j,i')}\right)}}_{\mathcal{SQ}_1} +$$

$$\underbrace{\frac{2\sigma L \tilde{\eta}_t}{N} \sum_{i=1}^{N} \sqrt{\frac{\tau(t,i)(K-1)^2}{K^2 N} \sum_{j=t-\tau(t,i)}^{t-1} \tilde{\eta}_j^2 \sum_{i'=1}^{N} \tilde{\eta}_{j-\tau(j,i')}^2 \left(8L^2 \Delta_{j-\tau(j,i')} + 8\|\nabla f_i(w_*)\|^2 + \frac{2\sigma^2}{K}\right)}}_{\mathcal{SQ}_2}.$$

$$(8)$$

Combining (B.3.1) and (8), we bound the expectation of $\mathcal{A}_1$.

$$\mathbb{E}\left[\mathcal{A}_1\right] \leq \mathbb{E}\left[\mathcal{B}_1\right] + \mathbb{E}\left[\mathcal{B}_2\right]$$

$$\leq -\frac{\mu \tilde{\eta}_t}{2} \Delta_t - 2\tilde{\eta}_t \left(\mathbb{E}\left[f(w_t)\right] - f(w_*)\right) + \frac{2\tilde{\eta}_t \sigma^2}{KN^2} \sum_{i=1}^{N} \sum_{j=t-\tau(t,i)}^{t-1} \tilde{\eta}_j + \mathcal{H} + \mathcal{SQ},$$

where $\mathcal{SQ} = \mathcal{SQ}_1 + \mathcal{SQ}_2$.

### B.3.2 Bounding the second term

Note that $\tilde{\nabla} f_i(w_{t-\tau(t,i),k}^i)$ can be split into three terms, i.e.,

$$\tilde{\nabla} f_i(w_{t-\tau(t,i),k}^i) = \left(\nabla f_i(w_{t-\tau(t,i),k}^i) - \nabla f_i(w_t)\right) + \nabla f_i(w_t) + e_{t-\tau(t,i),k}^i.$$

By Jensen's inequality,

$$\mathbb{E}\left[\mathcal{A}_2\right] \leq \underbrace{\frac{3\tilde{\eta}_t^2}{K^2 N^2} \mathbb{E}\left[\left\|\sum_{k,i} \left(\nabla f_i(w_{t-\tau(t,i),k}^i) - \nabla f_i(w_t)\right)\right\|^2\right]}_{\mathcal{B}_4}$$

$$+ \underbrace{\frac{3\tilde{\eta}_t^2}{K^2 N^2} \mathbb{E}\left[\left\|\sum_{k,i} \nabla f_i(w_t)\right\|^2\right]}_{\mathcal{B}_5} + \underbrace{\frac{3\tilde{\eta}_t^2}{K^2 N^2} \mathbb{E}\left[\left\|\sum_{k,i} e_{t-\tau(t,i),k}^i\right\|^2\right]}_{\mathcal{B}_6}.$$

Due to independence of $e_{t-\tau(t,i),k}^i$ and $e_{t-\tau(t,i),k'}^{i'}$ for $i \neq i'$ or $k \neq k'$, we have $\mathcal{B}_6 \leq \frac{3\tilde{\eta}_t^2 \sigma^2}{KN}$. Recall $\mathcal{C}_1 = \frac{2L\tilde{\eta}_t}{KN} \sum_{k,i} \left\|w_{t-\tau(t,i),k}^i - w_t\right\|^2$. By Jensen's inequality and $L$-smoothness, we then bound $\mathcal{B}_4$.

$$\mathcal{B}_4 \leq \frac{3L^2 \tilde{\eta}_t^2}{KN} \sum_{k,i} \mathbb{E}\left[\left\|w_{t-\tau(t,i),k}^i - w_t\right\|^2\right] = \frac{3}{2} L\tilde{\eta}_t \mathbb{E}\left[\mathcal{C}_1\right] \leq \frac{3}{2} L\tilde{\eta}_t \mathcal{H}.$$

By Lemma B.1, we have

$$\mathcal{B}_5 \leq 3\tilde{\eta}_t^2 \mathbb{E}\left[\|\nabla f(w_t)\|^2\right] \leq 6L\tilde{\eta}_t^2 \left(\mathbb{E}\left[f(w_t)\right] - f(w_*)\right).$$

Therefore

$$\mathbb{E}\left[\mathcal{A}_2\right] \le \frac{3}{2}L\tilde{\eta}_t\mathcal{H} + \frac{3\tilde{\eta}_t^2\sigma^2}{KN} + 6L\tilde{\eta}_t^2\left(\mathbb{E}\left[f(w_t)\right] - f(w_*)\right).$$

Combining Appendix B.3.1 and Appendix B.3.2, we have

$$\Delta_{t+1} \le \left(1 - \frac{1}{2}\mu\tilde{\eta}_t\right)\Delta_t - 2\tilde{\eta}_t\left(1 - 3L\tilde{\eta}_t\right)\left(\mathbb{E}\left[f(w_t)\right] - f(w_*)\right)$$

$$+ \frac{2\tilde{\eta}_t\sigma^2}{KN^2}\sum_{i=1}^{N}\sum_{j=t-\tau(t,i)}^{t-1}\tilde{\eta}_j + \frac{3\tilde{\eta}_t^2\sigma^2}{KN} + \left(1 + \frac{3}{2}L\tilde{\eta}_t\right)\mathcal{H} + \mathcal{SQ}.$$

Since when $\tilde{\eta}_t \le \frac{1}{25L}$, $-2\tilde{\eta}_t(1 - 3L\tilde{\eta}_t) \le -\frac{44}{25}\tilde{\eta}_t$ and $\frac{3}{2}L\tilde{\eta}_t \le \frac{3}{50}$, Lemma B.5 holds.

## B.4 Deriving the convergence bound

In this subsection, we obtain Theorem B.1 based on the descent lemma. We provide a bound for $\Delta_t$ in Appendix B.4.1 $\forall 1 \le t \le T$ and further bound $\mathbb{E}\left[f(\bar{\tau}_T)\right] - f(w_*)$ in Appendix B.5.

### B.4.1 Bounding the distance from the global optimum

**Lemma B.6** (A bound for the expected squared $l_2$-distance from the global optimum). *Assume that Assumptions 1 to 3 hold. Further assume that the device availability sequence $\tau(t, i)$ satisfies Assumption 4 and $\tau(t, i) = 0$, for all $i \in [N]$. By setting the learning rate $\eta_t = \frac{4}{\mu K(t+a)}$ with $a = \max\{100, 40t_0\}(\frac{L}{\mu})^{1.5}$. For all $1 \le t \le T$, after $t - 1$ communication rounds, $\Delta_t$ satisfies:*

$$\Delta_t \le \frac{Es_t\sigma^2}{(t+a)^2} + \frac{G}{t+a} + \frac{F}{(t+a)^2} := B_t, \tag{9}$$

*where*

$$E = \frac{35\sigma^2}{\mu^2 NK}, G = \frac{32\sigma^2}{\mu^2 NK}, F = \frac{\bar{d}_{\max,T}C_1 + (K-1)^2C_2 + C_3}{\mu^3 K^2},$$

*and*

$$C_1 = 2500LK^2(D + 2L\sigma^2/\mu), C_2 = 5000L(D + \sigma^2/K), C_3 = \max\{1600t_0^2, 10000\}L^3K^2\Delta_1^2.$$

We prove Lemma B.6 by induction. We first show that (9) holds when $t = 1$. Then assuming that $\Delta_{t'} \le B_{t'}$ holds for all $1 \le t' \le t$, we prove $\Delta_{t+1} \le B_{t+1}$ by verifying

$$B_{t+1} \overset{(a)}{\ge} F(B_t, \tilde{\eta}_t) \overset{(b)}{\ge} \text{RHS of (6)} \ge \Delta_{t+1}, \tag{10}$$

where $F$ is a function of $B_t$ and $\tilde{\eta}_t$. To validate (b), we prove that for all $0 \le m \le l_t$, $B_{t-m}$ and $\tilde{\eta}_{t-m}$ can be bounded by $B_t$ and $\tilde{\eta}_t$ respectively in Appendix B.4.2. We simplify terms of higher degree in Appendix B.4.3 and simplify terms with square roots in Appendix B.4.4. Finally, relation (a) is verified in Appendix B.4.6. A formal proof is provided as follows.

**Proof of Lemma B.6.** Note that (9) holds trivially when $t = 1$ since $\frac{C_3}{\mu^3 K^2} \ge a^2\Delta_1$. Now we assume $\forall 1 \le t' \le t, \Delta_{t'} \le B_{t'}$ holds.

### B.4.2 Connecting bounds and learning rates at different rounds

According to Assumption 4, $\tau(t, i) \le t_0 + \frac{1}{40}t, l_t \le 2t_0 + \frac{1}{20}t$. Combining with $t_0 \le \frac{1}{40}a$, we have

$$\frac{1}{t + a - \tau(t,i)} \le \frac{40}{39(t+a)} \quad \text{and} \quad \frac{1}{t + a - l_t} \le \frac{20}{19(t+a)}.$$

Therefore, for all $0 \le n \le \tau(t, i)$, we have

$$B_{t-n} = \frac{s_{t-n}E}{(t+a-n)^2} + \frac{G}{t+a-n} + \frac{F}{(t+a-n)^2}$$

$$\leq \frac{s_t E}{(t + a - \tau(t,i))^2} + \frac{G}{t + a - \tau(t,i)} + \frac{F}{(t + a - \tau(t,i))^2}$$

$$\leq \left(\frac{40}{39}\right)^2 B_t.$$

For all $0 \leq m \leq l_t$,

$$B_{t-m} \leq \frac{s_t E}{(t + a - l_t)^2} + \frac{G}{t + a - l_t} + \frac{F}{(t + a - l_t)^2} \leq \left(\frac{20}{19}\right)^2 B_t,$$

and

$$\tilde{\eta}_{t-n} \leq \tilde{\eta}_{t-\tau(t,i)} \leq \frac{40}{39}\tilde{\eta}_t, \forall 0 \leq n \leq \tau(t,i),$$

$$\tilde{\eta}_{t-m} \leq \tilde{\eta}_{t-l_t} \leq \frac{20}{19}\tilde{\eta}_t, \forall 0 \leq m \leq l_t.$$

Also, we have

$$\frac{2\tilde{\eta}_t \sigma^2}{KN^2} \sum_{i=1}^{N} \sum_{j=t-\tau(t,i)}^{t-1} \tilde{\eta}_j \leq \frac{80\tau_t \sigma^2}{39KN}\tilde{\eta}_t^2. \tag{11}$$

### B.4.3 Simplifying terms of higher degree

In this section, we simplify $\mathcal{H}$ in (6) and bound it by $B_t$ and $\tilde{\eta}_t$. Rearranging $\mathcal{H}$, we have

$$\mathcal{H} = \underbrace{\frac{16L^3}{N^2}\tilde{\eta}_t \sum_{i=1}^{N} \tau(t,i) \left( \sum_{j=t-\tau(t,i)}^{t-1} \sum_{i'=1}^{N} \tilde{\eta}_j^2 \Delta_{j-\tau(j,i)} \right)}_{\mathcal{I}_1} + \underbrace{\frac{16LD}{N}\tilde{\eta}_t \sum_{i=1}^{N} \tau(t,i) \left( \sum_{j=t-\tau(t,i)}^{t-1} \tilde{\eta}_j^2 \right)}_{\mathcal{I}_2}$$

$$+ \underbrace{\frac{8L\sigma^2}{KN}\tilde{\eta}_t \sum_{i=1}^{N} \tau(t,i) \left( \sum_{j=t-\tau(t,i)}^{t-1} \tilde{\eta}_j^2 \right)}_{\mathcal{I}_3}$$

$$+ \underbrace{\frac{64L^3(K-1)^2}{K^2N}\tilde{\eta}_t \sum_{i=1}^{N} \tilde{\eta}_{t-\tau(t,i)}^2 \Delta_{t-\tau(t,i)}}_{\mathcal{I}_4}$$

$$+ \underbrace{\frac{64L^5(K-1)^2}{K^2N^2}\tilde{\eta}_t \sum_{i=1}^{N} \tau(t,i) \left( \sum_{j=t-\tau(t,i)}^{t-1} \sum_{i'=1}^{N} \tilde{\eta}_j^2 \tilde{\eta}_{j-\tau(j,i')}^2 \Delta_{j-\tau(j,i')} \right)}_{\mathcal{I}_5}$$

$$+ \underbrace{\frac{64(K-1)^2L}{K^2N}\tilde{\eta}_t \sum_{i=1}^{N} \tilde{\eta}_{t-\tau(t,i)}^2 \|\nabla f_i(w_*)\|^2}_{\mathcal{I}_6}$$

$$+ \underbrace{\frac{64L^3(K-1)^2}{K^2N^2}\tilde{\eta}_t \sum_{i=1}^{N} \tau(t,i) \left( \sum_{j=t-\tau(t,i)}^{t-1} \sum_{i'=1}^{N} \tilde{\eta}_j^2 \tilde{\eta}_{j-\tau(j,i')}^2 \|\nabla f_{i'}(w_*)\|^2 \right)}_{\mathcal{I}_7}$$

$$+ \underbrace{\frac{16(K-1)^2L\sigma^2}{K^3N}\tilde{\eta}_t \sum_{i=1}^{N} \tilde{\eta}_{t-\tau(t,i)}^2}_{\mathcal{I}_8}$$

$$+ \underbrace{\frac{16L^3(K-1)^2\sigma^2}{K^3N^2}\tilde{\eta}_t\sum_{i=1}^{N}\tau(t,i)\left(\sum_{j=t-\tau(t,i)}^{t-1}\tilde{\eta}_j^2\sum_{i'=1}^{N}\tilde{\eta}_{j-\tau(j,i')}^2\right)}_{\mathcal{I}_9}.$$

We first show that $\mathcal{I}_1$, $\mathcal{I}_4$ and $\mathcal{I}_5$ can be bounded by $\mu\tilde{\eta}_tB_t$. According to Assumption 4,

$$\tau(t,i)\tilde{\eta}_t \leq \frac{4[t_0+(1/b)t]}{\mu(a+t)} \leq \frac{4[t_0+(1/b)t]}{\mu(bt_0+t)} \leq \frac{4}{\mu b} \leq \frac{\mu^{0.5}}{10L^{1.5}} \leq \frac{1}{10L}. \tag{12}$$

Combining the result in Appendix B.4.2, we can bound $\mathcal{I}_1$ in the following way.

$$\mathcal{I}_1 \leq 16\left(\frac{40}{39}\right)^2\left(\frac{20}{19}\right)^2 L\tilde{\eta}_t\left(L^2\tilde{\eta}_t^2\frac{1}{N}\sum_{i=1}^{N}\tau(t,i)^2\right)B_t \leq 0.19\mu\tilde{\eta}_tB_t.$$

Similarly, $\mathcal{I}_5$ and $\mathcal{I}_4$ can be bounded as follows.

$$\mathcal{I}_5 \leq \frac{64L^5(K-1)^2}{K^2N^2}\tilde{\eta}_t\sum_{i=1}^{N}\tau(t,i)\sum_{j=t-\tau(t,i)}^{t-1}\sum_{i'=1}^{N}\left(\frac{40}{39}\right)^2\left(\frac{20}{19}\right)^4\tilde{\eta}_t^4B_t$$

$$\leq \frac{83L^5(K-1)^2}{K^2}\tilde{\eta}_t^3\left(\tilde{\eta}_t\tau(t,i)\right)^2B_t$$

$$\leq \frac{0.83L^3(K-1)^2}{K^2}\tilde{\eta}_t^3B_t,$$

$$\mathcal{I}_4 \leq 64\left(\frac{40}{39}\right)^4\frac{L^3(K-1)^2}{K^2}\tilde{\eta}_t^3B_t \leq \frac{71L^3(K-1)^2}{K^2}\tilde{\eta}_t^3B_t.$$

Further using $\tilde{\eta}_t \leq \frac{4}{\mu a} \leq \frac{\mu^{0.5}}{25L^{1.5}}$, we have

$$\mathcal{I}_4+\mathcal{I}_5 \leq \frac{68.13L^3(K-1)^2}{K^2}\tilde{\eta}_t^3B_t \leq \left(\frac{71.83L^3}{\mu}\tilde{\eta}_t^2\right)\mu\tilde{\eta}_tB_t = 0.12\mu\tilde{\eta}_tB_t.$$

By the same token,

$$\mathcal{I}_6+\mathcal{I}_7 \leq \frac{68.1(K-1)^2L}{K^2}\tilde{\eta}_t^3D,$$

$$\mathcal{I}_8+\mathcal{I}_9 \leq \frac{17.03(K-1)^2L}{K^3}\tilde{\eta}_t^3\sigma^2.$$

Still using the result in Appendix B.4.2, we can bound $\mathcal{I}_2$ and $\mathcal{I}_3$.

$$\mathcal{I}_2 \leq 16\left(\frac{40}{39}\right)^2\left(\frac{1}{N}\sum_{i=1}^{N}\tau(t,i)^2\right)LD\tilde{\eta}_t^3 \leq 16.84LDd_t\tilde{\eta}_t^3,$$

$$\mathcal{I}_3 \leq 8\left(\frac{40}{39}\right)^2\left(\frac{1}{N}\sum_{i=1}^{N}\tau(t,i)^2\right)\frac{L\sigma^2}{K}\tilde{\eta}_t^3 \leq \frac{8.42d_tL\sigma^2\tilde{\eta}_t^3}{K}.$$

Therefore,

$$\mathcal{H} \leq 0.31\mu\tilde{\eta}_tB_t + 16.84LDd_t\tilde{\eta}_t^3 + \frac{68.1(K-1)^2L}{K^2}\tilde{\eta}_t^3D$$

$$+ \frac{8.42d_tL\sigma^2\tilde{\eta}_t^3}{K} + \frac{17.03(K-1)^2L}{K^3}\tilde{\eta}_t^3\sigma^2 \tag{13}$$

### B.4.4 Simplifying terms with square roots

In this section, we bound terms with square roots on RHS of (6), i.e., $\mathcal{SQ}$. We apply the results in Appendix B.4.2 to bound the first term.

$$2\sigma\tilde{\eta}_tL\frac{1}{N}\sum_{i=1}^{N}\sqrt{\tau(t,i)\sum_{j=t-\tau(t,i)}^{t-1}\tilde{\eta}_j^2\frac{1}{N}\sum_{i'=1}^{N}\Delta_{j-\tau(j,i')}}$$

$$\leq \frac{80}{39}\sigma\tilde{\eta}_t^2 L \frac{1}{N}\sum_{i=1}^N \sqrt{\tau(t,i)\frac{1}{N}\sum_{j=t-\tau(t,i)}^{t-1}\sum_{i'=1}^N B_{j-\tau(j,i')}}$$

$$\leq \frac{80}{39}\sigma\tilde{\eta}_t^2 \tau_t L \frac{1}{N}\sum_{i=1}^N \sqrt{\left(\frac{20}{19}\right)^2 B_t}$$

$$\leq 2.16\sigma\tilde{\eta}_t^2 \tau_t L \sqrt{B_t}.$$

Recall

$$B_t \geq \frac{\bar{d}_{\max,T}C_1}{\mu^3(t+a)^2} \geq \frac{5000\bar{d}_{\max,T}L^2\sigma^2}{\mu^4(t+a)^2}.$$

Since $\tau_t^2 = \left[\frac{1}{N}\sum_{i=1}^N \tau(t,i)\right]^2 \leq \frac{1}{N}\sum_{i=1}^N \tau(t,i)^2 \leq \bar{d}_{\max,T}$, we have

$$\sqrt{B_t} \geq \frac{70\tau_t L\sigma}{\mu^2(t+a)} \geq \frac{1}{\mu}\cdot 8 \cdot \frac{4}{\mu(t+a)}(2.16\sigma\tau_t L) = \frac{8}{\mu}(2.16\sigma\tilde{\eta}_t\tau_t L).$$

Therefore,

$$\frac{1}{8}\mu\tilde{\eta}_t B_t \geq 2.16\sigma\tilde{\eta}_t^2 \tau_t L\sqrt{B_t}.$$

Next, we bound the second term.

$$\frac{2\sigma L\tilde{\eta}_t}{N}\sum_{i=1}^N \sqrt{\frac{\tau(t,i)(K-1)^2}{K^2 N}\sum_{j=t-\tau(t,i)}^{t-1}\tilde{\eta}_j^2\sum_{i'=1}^N \tilde{\eta}_{j-\tau(j,i')}^2\left(8L^2\Delta_{j-\tau(j,i')}+8\|\nabla f_{i'}(w_*)\|^2+\frac{2\sigma^2}{K}\right)}$$

$$\leq \frac{2\sigma L\tilde{\eta}_t}{N}\sum_{i=1}^N \sqrt{\frac{\tau(t,i)(K-1)^2}{K^2 N}\sum_{j=t-\tau(t,i)}^{t-1}\tilde{\eta}_{t-\tau(t,i)}^2\sum_{i'=1}^N \tilde{\eta}_{t-l_t}^2\left(8L^2 B_{j-\tau(j,i')}+8\|\nabla f_{i'}(w_*)\|^2+\frac{2\sigma^2}{K}\right)}$$

$$\leq \frac{2.16\sigma L\tilde{\eta}_t^3}{N}\sum_{i=1}^N \sqrt{\frac{\tau(t,i)^2(K-1)^2}{K^2}\left(8\left(\frac{20}{19}\right)^2 L^2 B_t+8D+\frac{2\sigma^2}{K}\right)}$$

$$\leq \frac{(K-1)2.16\sigma L\tau_t\tilde{\eta}_t^3}{K}\sqrt{\left(8\left(\frac{20}{19}\right)^2 L^2 B_t+8D+\frac{2\sigma^2}{K}\right)}.$$

To show that $\frac{(K-1)2.16\sigma L\tau_t\tilde{\eta}_t^3}{K}\sqrt{\left(8\left(\frac{20}{19}\right)^2 L^2 B_t+8D+\frac{2\sigma^2}{K}\right)} \leq \frac{1}{8}\mu\tilde{\eta}_t B_t$, we only have to prove

$$B_t^2 \geq \frac{64(2.16)^2(K-1)^2\sigma^2 L^2\tau_t^2}{\mu^2 K^2}\tilde{\eta}_t^4\left[8\left(\frac{20}{19}\right)^2 L^2 B_t+8D+\frac{2\sigma^2}{K}\right]. \tag{14}$$

To let (14) hold, we only have to verify

$$\frac{1}{2}B_t \geq 2647\frac{(K-1)^2\sigma^2 L^4\tau_t^2}{\mu^2 K^2}\tilde{\eta}_t^4, \tag{15}$$

$$\frac{1}{4}B_t^2 \geq 2500\frac{(K-1)^2\sigma^2 L^2\tau_t^2 D}{\mu^2 K^2}\tilde{\eta}_t^4 \Leftrightarrow B_t \geq \frac{100(K-1)\sigma L\tau_t\sqrt{D}}{\mu K}\tilde{\eta}_t^2 = \frac{1600(K-1)\sigma L\tau_t\sqrt{D}}{\mu^3 K(t+a)^2}, \tag{16}$$

$$\frac{1}{4}B_t^2 \geq 625\frac{(K-1)^2\sigma^2 L^2\tau_t^2}{\mu^2 K^2}\tilde{\eta}_t^4\left(\frac{\sigma^2}{K}\right) \Leftrightarrow B_t \geq \frac{50(K-1)L\sigma\tau_t}{\mu K}\tilde{\eta}_t^2\frac{\sigma}{\sqrt{K}} = \frac{800(K-1)L\sigma^2\tau_t}{\mu^3 K^{1.5}(t+a)^2}. \tag{17}$$

Since $\tilde{\eta}_t \leq \frac{\mu^{0.5}}{25L^{1.5}}$, we have

$$2647\frac{(K-1)^2\sigma^2 L^4\tau_t^2}{\mu^2 K^2}\tilde{\eta}_t^4 \leq \frac{4.24(K-1)^2 L\bar{d}_{\max,T}\sigma^2}{\mu^3 K^2(t+a)^2} \leq \frac{2500\bar{d}_{\max,T}L^2\sigma^2}{\mu^4(t+a)^2} \leq \frac{1}{2}B_t.$$

Therefore (15) holds. Also note that

$$\frac{1600(K-1)\sigma L\tau_t\sqrt{D}}{\mu^3 K(t+a)2} = \frac{1600L}{\mu^3(t+a)^2}\left[\left(\frac{K-1}{K}\sqrt{D}\right)(\tau_t\sigma)\right]$$

$$\leq \frac{800L(K-1)^2 D}{\mu^3 K^2(t+a)^2} + \frac{800L\bar{d}_{\max,T}\sigma^2}{\mu^3(t+a)^2}$$

$$\leq B_t.$$

Hence, (16) holds. Similarly,

$$\frac{800(K-1)L\sigma^2\tau_t}{\mu^3 K^{1.5}(t+a)^2} = \frac{800\sigma^2 L}{\mu^3(t+a)^2}\left[\left(\frac{K-1}{K^{1.5}}\right)\right]\tau_t \leq \frac{400L(K-1)^2\sigma^2}{\mu^3 K^3(t+a)^2} + \frac{400L\bar{d}_{\max,T}\sigma^2}{\mu^3(t+a)^2} \leq B_t.$$

Therefore, (17) holds. Now we have obtained a bound for $\mathcal{SQ}$. That is,

$$\mathcal{SQ} \leq \frac{1}{4}\mu\tilde{\eta}_t B_t. \tag{18}$$

### B.4.5 Verifying relation (b)

In this subsection, we verify relation (b) by using the results in Appendix B.4.2, Appendix B.4.3 and Appendix B.4.4. First apply the definition of strong convexity and therefore,

$$\mathbb{E}\left[f(w_t)\right] - f(w_*) \geq \frac{\mu}{2}\Delta_t. \tag{19}$$

Since $\mu\tilde{\eta}_t \leq \frac{4}{a} \leq \frac{1}{25}$, $1 - 1.38\mu\tilde{\eta}_t \geq 0$. We have

$$\left(1 - \frac{1}{2}\mu\tilde{\eta}_t\right)\Delta_t - \frac{44}{25}\tilde{\eta}_t\left(\mathbb{E}\left[f(w_t)\right] - f(w_*)\right) \leq (1 - 1.38\mu\tilde{\eta}_t)\Delta_t \leq (1 - 1.38\mu\tilde{\eta}_t)B_t. \tag{20}$$

Combining (11), (13), (18) and (20), we obtain

$$\begin{aligned}
\text{RHS of (6)} &\leq (1 - 1.38\mu\tilde{\eta}_t)B_t + 0.25\mu\tilde{\eta}_t B_t + 0.33\mu\tilde{\eta}_t B_t + \frac{80\tau_t\sigma^2}{39KN}\tilde{\eta}_t^2 + \frac{3\sigma^2}{KN}\tilde{\eta}_t^2 \\
&\quad + \left[18\bar{d}_{\max,T} + \frac{73(K-1)^2}{K^2}\right]LD\tilde{\eta}_t^3 + \left[\frac{9\bar{d}_{\max,T}}{K} + \frac{18.1(K-1)^2}{K^3}\right]L\sigma^2\tilde{\eta}_t^3.
\end{aligned} \tag{21}$$

Therefore, relation (b) is verified.

### B.4.6 Verifying relation (a)

To verify relation (a), we only have to show

$$\begin{aligned}
B_{t+1} + 0.8\mu\tilde{\eta}_t B_t &\geq B_t + \frac{80\tau_t\sigma^2}{39KN}\tilde{\eta}_t^2 + \frac{3\sigma^2}{KN}\tilde{\eta}_t^2 + \left[18\bar{d}_{\max,T} + \frac{73(K-1)^2}{K^2}\right]LD\tilde{\eta}_t^3 \\
&\quad + \left[\frac{9\bar{d}_{\max,T}}{K} + \frac{18.1(K-1)^2}{K^3}\right]L\sigma^2\tilde{\eta}_t^3.
\end{aligned} \tag{22}$$

Note that $B_{t+1}$ can be split as

$$B_{t+1} = \frac{\tau_t E}{(t+a+1)^2} + \frac{s_t E}{(t+a+1)^2} + \frac{G}{t+a+1} + \frac{F}{(t+a+1)^2},$$

and that

$$\frac{1}{t+a} - \frac{1}{t+a+1} = \frac{1}{(t+a)(t+a+1)} \leq \frac{1}{(t+a)^2},$$

$$\frac{1}{(t+a)^2} - \frac{1}{(t+a+1)^2} = \frac{2t+2a+1}{(t+a)^2(t+a+1)^2} \leq \frac{2}{(t+a)^3}.$$

Therefore, to prove (22), we only have to show

$$\frac{\tau_t E}{(t+a+1)^2} \geq \frac{80\tau_t\sigma^2}{39KN}\tilde{\eta}_t^2 = \frac{1280\tau_t\sigma^2}{39\mu^2 KN(t+a)^2}, \tag{23}$$

and

$$0.8\mu\tilde{\eta}_t B_t \geq \frac{2Es_t}{(t+a)^3} + \frac{2F}{(t+a)^3} + \frac{G}{(t+a)^2} + \frac{48\sigma^2}{KN\mu^2(t+a)^2}$$

$$+ \left[18\bar{d}_{\max,T} + \frac{73(K-1)^2}{K^2}\right] LD\tilde{\eta}_t^3 \tag{24}$$

$$+ \left[\frac{9\bar{d}_{\max,T}}{K} + \frac{18.1(K-1)^2}{K^3}\right] L\sigma^2\tilde{\eta}_t^3.$$

(23) holds since

$$E = \frac{35\sigma^2}{\mu^2 NK} \geq \frac{1280(40+1)^2}{39(40^2)\mu^2 NK} \geq \frac{1280(t+a+1)^2}{39\mu^2 NK(t+a)^2}.$$

To show that (24) holds, we plug in the value of $B_t$ and $\tilde{\eta}_t$ and make minor adjustments.

$$\frac{1.2Es_{t-1}}{(t+a)^3} + \frac{1.2F}{(t+a)^3} + \frac{2.2G}{(t+a)^2}$$

$$\geq \frac{48\sigma^2}{KN\mu^2(t+a)^2} + \frac{\bar{d}_{\max,T}L(1152D + 576\sigma^2/K)}{\mu^3(t+a)^3} + \frac{(K-1)^2}{K^2} \cdot \frac{L(4672D + 1158.4\sigma^2/K)}{\mu^3(t+a)^3}.$$

Recall

$$G = \frac{22\sigma^2}{\mu^2 NK}, F \geq \frac{\bar{d}_{\max,T}L(2500D + 5000L\sigma^2/\mu)}{\mu^3} + \frac{(K-1)^2}{K^2} \cdot \frac{5000L(D+\sigma^2/K)}{\mu^3}.$$

Thus (24) holds. Now we have completed the induction step and obtain Lemma B.6.

## B.5  Proof of Theorem B.1

In this subsection, we provide a bound for $\mathbb{E}\left[f(\overline{w}_T)\right] - f(w_*)$ based on the bound for $\Delta_T$. Here we restate the descent lemma.

$$\Delta_{t+1} \leq \left(1 - \frac{1}{2}\mu\tilde{\eta}_t\right)\Delta_t - \frac{44}{25}\tilde{\eta}_t\left(\mathbb{E}\left[f(w_t)\right] - f(w_*)\right)$$

$$+ \underbrace{\frac{2\tilde{\eta}_t\sigma^2}{KN^2}\sum_{i=1}^{N}\sum_{j=t-\tau(t,i)}^{t-1}\tilde{\eta}_j + \frac{3\tilde{\eta}_t^2\sigma^2}{KN} + \frac{53}{50}\mathcal{H} + \mathcal{SQ}}_{\mathcal{Q}_t}. \tag{25}$$

Interestingly, the proof in Appendix B.4.1 generates a bound for $\mathcal{Q}_t$. Combining (21) and (22), we find

$$B_{t+1} \geq (1 - 1.38\mu\tilde{\eta}_t)B_t + 0.25\mu\tilde{\eta}_t B_t + 0.33\mu\tilde{\eta}_t B_t + \frac{80\tau_t\sigma^2}{39KN}\tilde{\eta}_t^2$$

$$+ \left[18\bar{d}_{\max,T} + \frac{73(K-1)^2}{K^2}\right] LD\tilde{\eta}_t^3 + \left[\frac{9\bar{d}_{\max,T}}{K} + \frac{18.1(K-1)^2}{K^3}\right] L\sigma^2\tilde{\eta}_t^3$$

$$\geq (1 - 1.38\mu\tilde{\eta}_t)B_t + \mathcal{Q}_t.$$

Hence

$$\mathcal{Q}_t \leq B_{t+1} - B_t + 1.38\mu\tilde{\eta}_t B_t \leq \frac{E\tau_t}{(t+a+1)^2} + 1.38\mu\tilde{\eta}_t B_t.$$

Rearrange (25), we have

$$\frac{44}{25}\tilde{\eta}_t\left(\mathbb{E}\left[f(w_t)\right] - f(w_*)\right) \leq \left(1 - \frac{1}{2}\mu\tilde{\eta}_t\right)\Delta_t - \Delta_{t+1} + \mathcal{Q}_t.$$

Apply (19) and subtract $0.25\mu\tilde{\eta}_t\Delta_t$ on the RHS and $0.5\tilde{\eta}_t(\mathbb{E}\left[f(w_t)\right] - f(w_*))$ on the LHS. Then we have

$$\frac{63}{50}\tilde{\eta}_t(\mathbb{E}\left[f(w_t)\right] - f(w_*)) \leq \left(1 - \frac{3}{4}\mu\tilde{\eta}_t\right)\Delta_t - \Delta_{t+1} + \mathcal{Q}_t.$$

Dividing $\tilde{\eta}_t$ on both sides and multiplying both sides by $(t+a-1)(t+a-2)$, we have

$$(t+a-1)(t+a-2)(\mathbb{E}\left[f(w_t)\right] - f(w_*))$$
$$\leq \frac{\mu(t+a-3)(t+a-2)(t+a-1)}{4}\Delta_t - \frac{\mu(t+a-2)(t+a-1)(t+a)}{4}\Delta_{t+1}$$
$$+ \frac{\mu(t+a-2)(t+a-1)(t+a)}{4}\mathcal{Q}_t$$
$$\leq \frac{\mu(t+a-3)(t+a-2)(t+a-1))}{4}\Delta_t - \frac{\mu(t+a-2)(t+a-1)(t+a)}{4}\Delta_{t+1}$$
$$+ E''\tau_t(t+a) + E's_t + F' + (t+a)G'.$$

where $E'' = \frac{\mu}{4}E$, $E' = 0.345\mu E$, $F' = 0.345\mu F$, $G' = 0.345\mu G$. Telescoping from $t=1$ to $T-1$, we have

$$\sum_{t=1}^{T-1}(t+a-1)(t+a-2)\left\{\mathbb{E}\left[f(w_t)\right] - f(w_*)\right\} + \frac{\mu(T+a-3)(T+a-2)(T+a-1)}{4}\Delta_T$$
$$\leq \frac{\mu a^3}{4}\Delta_1 + E''\sum_{t=1}^{T-1}\tau_t(t+a) + E'\sum_{t=1}^{T-1}s_t + F'T + G'\sum_{t=1}^{T-1}(t+a).$$

$$(26)$$

By $L$-smoothness, $f(w_t) - f(w_*) \leq \frac{L}{2}\Delta_t$. Since $a \geq 100(\frac{L}{\mu})^{1.5}$, $\frac{\mu(a-2)}{4} \geq \frac{L}{2}$. Therefore,

$$\frac{\mu(T+a-3)(T+a-2)(T+a-1)}{4}\Delta_T \geq \frac{\mu(a-2)(T+a-2)(T+a-1)}{4}\Delta_T$$
$$\geq (T+a-2)(T+a-1)\left\{\mathbb{E}\left[f(w_T)\right] - f(w_*)\right\}.$$

Then (26) can be further simplified as

$$\sum_{t=1}^{T}(t+a-1)(t+a-2)\left\{\mathbb{E}\left[f(w_t)\right] - f(w_*)\right\}$$
$$\leq \frac{\mu a^3}{4}\Delta_1 + E''\sum_{t=1}^{T-1}\tau_t(t+a) + E'\sum_{t=1}^{T-1}s_t + F'T + G'\sum_{t=1}^{T-1}(t+a).$$

$$(27)$$

Since $\sum_{t=1}^{T-1}s_t = \sum_{t=1}^{T-1}\sum_{t'=1}^{t-1}\tau_{t'} = \sum_{t=1}^{T-1}(T-1-t)\tau_t$, we have

$$E''\sum_{t=1}^{T-1}\tau_t(t+a) + E'\sum_{t=1}^{T-1}s_t \leq E'(T-1+a)\sum_{t=1}^{T-1}\tau_t$$
$$\leq E'(T+a)s_T.$$

Therefore,

$$\text{RHS of (27)} \leq \frac{\mu a^3}{4}\Delta_1 + E'(T+a)s_T + F'T + G'T(T+a).$$

Define $W_T = \sum_{t=1}^{T}(t+a-1)(t+a-2) = \frac{1}{3}T^3 + (a-1)T^2 + (a^2 - 2a + \frac{2}{3})T$. Note that $W_T \geq \frac{1}{3}T^2(T+a)$. Dividing $W_T$ on both sides, we have

$$\left\{\frac{1}{W_T}\sum_{t=1}^{T}(t+a-1)(t+a-2)\mathbb{E}\left[f(w_t)\right]\right\} - f(w_*)$$
$$\leq \frac{3\mu a^3}{4(T+a)^3}\Delta_1 + \frac{3E's_T}{T^2} + \frac{3G'}{T} + \frac{3F'}{T^2}.$$

Considering $\frac{\mu a^3}{(T+a)^3} \leq \frac{\mu a^2}{T^2}$ and convexity of $f(w)$, we have

$$\mathbb{E}\left[f(\overline{w}_T)\right] - f(w_*) = \mathcal{O}\left(\frac{G' + E'\bar{\tau}_T}{T} + \frac{F'}{T^2}\right).$$

where $\overline{w}_T = \frac{1}{W_T}\sum_{t=1}^{T}(t+a-1)(t+a-2)w_t$. Plugging in $G'$, $E'$ and $F'$, we obtain Theorem B.1. Since $\bar{d}_{\max,T} \leq \tau_{\max,T}^2$, Theorem 5.1 holds.

## C    Proof of convergence for smooth and non-convex objective functions

In this section, we first state a more general version of Theorem 6.1 and then provide a proof. The proof of Theorem 6.1 is provided as a corollary (See Corollary C.1). Regarding the number of inactive rounds, we have the following relaxed assumption.

**Assumption 9.** *There exists a constant $t_0$ such that $\forall t \geq 1$ and $i \in [N]$, $\tau(t, i) \leq \frac{1}{4}\sqrt{\frac{L}{(L^2+\rho\delta)KN}}\max\{\sqrt{t}, \sqrt{t_0}\}$.*

Note that different from Assumption 8, Assumption 9 allows $\tau(t,i)$ to grow as $\mathcal{O}(\sqrt{t})$. Let $\bar{\tau}_T$ and $\tau_{\max,T}$ be be defined the same as in Section 5. Further define

$$\bar{\tau}_{\max,T} = \frac{1}{N}\sum_{i=1}^{N} \max_{1\leq t\leq T-1}\{\tau(t,i)\},$$

which takes the maximum number of inactive rounds over rounds $1, \cdots, T-1$ for each device and takes the average across devices. And define

$$\bar{d}_T = \frac{1}{T-1}\sum_{t=1}^{T-1} d_t,$$

which is the average of squared number of inactive rounds across all devices and rounds. The following theorem summarizes the performance of `MIFA` on smooth and non-convex problems.

**Theorem C.1.** *Let Assumptions 1, 2 and 5 to 7 hold. Further assume that the device availability sequence $\tau(t,i)$ satisfies Assumption 9 and $\tau(t,i) = 0$ for all $i \in [N]$. By setting the learning rate $\eta = c_0\sqrt{\frac{N}{KTL(1+\bar{\tau}_T)}}$, where constant $c_0$ satisfies $0 < c_0 \leq 1$ and $T \geq \max\{\frac{64\alpha^2 KNL^3}{L^2+\rho\delta}, 16LNK, t_0\}$, after communication rounds $1, \cdots, T-1$, `MIFA` satisfies:*

$$\min_{1\leq t\leq T}\mathbb{E}_\xi\left[\|\nabla f(w_t)\|^2\right] = \mathcal{O}\left(\sqrt{\frac{(1+\bar{\tau}_T)L}{TKN}}(f(w_1) - f^* + \sigma^2) + \frac{A_6}{T}\right),$$

*where*

$$A_6 = \frac{1}{(1+\bar{\tau}_T)}\left[\sigma^2\bar{\tau}_{\max,T}^2 NKL\left(1 + \frac{\alpha\bar{\tau}_{\max,T}L^2}{(\rho\delta+L^2)(1+\bar{\tau}_T)}\right) + \frac{(L^2+\rho\delta)\sigma^2}{L}\bar{d}_T \right.$$

$$\left. + (K-1)NL(\beta + \sigma^2/K)\right] + LKN\tau_{\max,T}\sigma\sqrt{\beta + \frac{\sigma^2}{KN}}.$$

### C.1    Additional notation

Define $r_T = \sum_{t=1}^{T-1} d_t$, which is the sum of average squared number of inactive rounds over the first $T-1$ communication rounds. Define $g_t = \frac{1}{KN}\sum_{k,i}\nabla f_i(w_{t-\tau(t,i),k}^i)$, which is the scaled accumulated true gradients at round $t$. Also define $l_{\max,T} = 2\tau_{\max,T}$ and $\tilde{\eta} = K\eta$ for convenience.

### C.2    Preliminary lemmas

Before starting the proof, we introduce some preliminary lemmas in this subsection.

**Lemma C.1** (Property of Hessian Lipschitz functions)**.** *For a $\rho$-Hessian Lipschitz function $f$ and for all $w, v$ and $z$, the following holds.*

$$\langle\nabla f(w) - \nabla f(v), z\rangle \leq \langle\nabla^2 f(v)(w-v), z\rangle + \frac{\rho}{2}\|z\|\|w-v\|^2.$$

*Proof.*

$$\langle\nabla f(w) - \nabla f(v), z\rangle$$

$$= \left\langle\left[\int_0^1 \nabla^2 f(v+\theta(w-v))d\theta\right](w-v), z\right\rangle$$

$$= \left\langle \nabla^2 f(v)(w-v), z \right\rangle + \left\langle \left\{ \int_0^1 \left[ \nabla^2 f(v + \theta(w-v)) - \nabla^2 f(v) \right] d\theta \right\} (w-v), z \right\rangle$$

$$\leq \left\langle \nabla^2 f(v)(w-v), z \right\rangle + \|z\| \|w-v\| \left\| \int_0^1 \left[ \nabla^2 f(v + \theta(w-v)) - \nabla^2 f(v) \right] d\theta \right\|$$

$$\leq \left\langle \nabla^2 f(v)(w-v), z \right\rangle + \|z\| \|w-v\| \int_0^1 \left\| \nabla^2 f(v + \theta(w-v)) - \nabla^2 f(v) \right\| d\theta$$

$$\leq \left\langle \nabla^2 f(v)(w-v), z \right\rangle + \rho \|z\| \|w-v\|^2 \int_0^1 \theta d\theta$$

$$\leq \left\langle \nabla^2 f(v)(w-v), z \right\rangle + \frac{\rho}{2} \|z\| \|w-v\|^2 .$$

$\square$

**Lemma C.2** (Bounded drift for non-convex objective functions). *For all* $K \geq 1, 0 \leq k \leq K-1$, $\tilde{\eta} \leq \frac{1}{10L}$, *we have bounded drift*

$$\mathbb{E}\left[\left\| w_{t,k}^i - w_t \right\|^2\right] \leq \frac{4\alpha\tilde{\eta}^2(K-1)}{K} \mathbb{E}\left[\|\nabla f(w_t)\|^2\right] + \frac{4(K-1)\tilde{\eta}^2\beta_i}{K} + \frac{2(K-1)\tilde{\eta}^2\sigma^2}{K^2}.$$

*Proof.* Simply combining (5) in Lemma B.4 and Assumption 7, we have

$$\mathbb{E}\left[\left\| w_{t,k}^i - w_t \right\|^2\right] \leq 2(K-1)\left( \frac{2\tilde{\eta}^2}{K} \mathbb{E}\left[\|\nabla f_i(w_t)\|^2\right] + \frac{\tilde{\eta}^2\sigma^2}{K^2} \right)$$

$$\leq \frac{4\alpha\tilde{\eta}^2(K-1)}{K} \mathbb{E}\left[\|\nabla f(w_t)\|^2\right] + \frac{4(K-1)\tilde{\eta}^2\beta_i}{K} + \frac{2(K-1)\tilde{\eta}^2\sigma^2}{K^2}.$$

$\square$

**Lemma C.3** (Bounding the difference of parameters at different rounds). *For all* $t \geq t', t - t' \leq l$, *where* $l$ *is a constant and* $\tilde{\eta} \leq \frac{1}{\sqrt{12}L}$, *the following inequality holds.*

$$\mathbb{E}\left[\|w_t - w_{t'}\|^2\right] \leq \frac{4\alpha l\tilde{\eta}^2}{N} \sum_{j=\max\{t-l,1\}}^{t-1} \sum_{i=1}^{N} \mathbb{E}\left[\left\|\nabla f(w_{j-\tau(j,i)})\right\|^2\right]$$

$$+ 4l^2\beta\tilde{\eta}^2 + \frac{4\tilde{\eta}^2 l^2}{KN}\sigma^2.$$

*Proof.* Since $\tilde{\nabla} f_i(w_{j-\tau(j,i),k}^i) = \nabla f_i(w_{j-\tau(j,i),k}^i) - \nabla f_i(w_{j-\tau(j,i)}) + \nabla f_i(w_{j-\tau(j,i)}) + e_{j-\tau(j,i),k}^i,$

$$\mathbb{E}\left[\|w_t - w_{t'}\|^2\right]$$

$$= \tilde{\eta}^2 \mathbb{E}\left[\left\| \sum_{j=t'}^{t-1} \frac{1}{KN} \sum_{k,i} \tilde{\nabla} f_i(w_{j-\tau(j,i),k}^i) \right\|^2\right]$$

$$\leq 3\tilde{\eta}^2 \mathbb{E}\left[\left\| \frac{1}{KN} \sum_{j=t'}^{t-1} \sum_{k,i} \left( \nabla f_i(w_{j-\tau(j,i),k}^i) - \nabla f_i(w_{j-\tau(j,i)}) \right) \right\|^2\right]$$

$$+ 3\tilde{\eta}^2 \mathbb{E}\left[\left\| \frac{1}{N} \sum_{j=t'}^{t-1} \sum_{i=1}^{N} \nabla f_i(w_{j-\tau(j,i)}) \right\|^2\right] + \frac{3\tilde{\eta}^2}{K^2 N^2} \mathbb{E}\left[\left\| \sum_{j=t'}^{t-1} \left( \sum_{k,i} e_{j-\tau(j,i'),k}^i \right) \right\|^2\right]$$

$$\leq \frac{3(t-t')L^2\tilde{\eta}^2}{KN} \sum_{j=t-t'}^{t-1} \sum_{k,i} \mathbb{E}\left[\left\| w_{j-\tau(j,i),k}^i - w_{j-\tau(j,i)} \right\|^2\right]$$

$$+ \frac{3\tilde{\eta}^2(t-t')}{N} \sum_{j=t-t'}^{t-1} \sum_{i=1}^{N} \mathbb{E}\left[\left\| \nabla f_i(w_{j-\tau(j,i)}) \right\|^2\right] + \frac{3\tilde{\eta}^2(t-t')^2}{KN}\sigma^2$$

$$\leq \frac{3\alpha l\tilde{\eta}^2}{N} \sum_{j=\max\{t-l,1\}}^{t-1} \sum_{i=1}^{N} \mathbb{E}\left[\left\|\nabla f(w_{j-\tau(j,i)})\right\|^2\right] + 3l^2\beta\tilde{\eta}^2 + \frac{3\tilde{\eta}^2 l^2}{KN}\sigma^2$$

$$+ \frac{12\alpha l L^2\tilde{\eta}^4(K-1)^2}{NK^2} \sum_{j=\max\{t-l,1\}}^{t-1} \sum_{i=1}^{N} \mathbb{E}\left[\left\|\nabla f(w_{j-\tau(j,i)})\right\|^2\right] + \frac{12(K-1)^2 l^2 L^2\beta\tilde{\eta}^4}{K^2}$$

$$+ \frac{6(K-1)^2 l^2 L^2\tilde{\eta}^4\sigma^2}{K^3}$$

$$\leq \frac{4\alpha l\tilde{\eta}^2}{N} \sum_{j=t-l}^{t-1} \sum_{i=1}^{N} \mathbb{E}\left[\left\|\nabla f(w_{j-\tau(j,i)})\right\|^2\right] + 4l^2\beta\tilde{\eta}^2 + \frac{4\tilde{\eta}^2 l^2}{KN}\sigma^2.$$

The first inequality above uses Jensen's inequality. The second one utilizes $L$-smoothness and Jensen's inequality. The third one uses Lemma C.2 and the last one holds since $\tilde{\eta} \leq \frac{1}{\sqrt{12}L}$. $\qquad\square$

## C.3 The descent lemma for smooth and non-convex problems

In this subsection, we state the descent lemma and provide a proof.

**Lemma C.4** (Descent lemma for non-convex problems)**.** *Assume that Assumptions 1, 2 and 5 to 7 hold. Further assume that $\tau(1,i) = 0$ for all $i \in [N]$. For any learning rate satisfying $\tilde{\eta} \leq \frac{1}{\sqrt{12}L}$, i.e., $\eta \leq \frac{1}{\sqrt{12}KL}$, the following holds for all $1 \leq t \leq T$.*

$$\mathbb{E}\left[f(w_{t+1})\right] - \mathbb{E}\left[f(w_t)\right]$$

$$\leq -\frac{\tilde{\eta}}{2}\mathbb{E}\left[\left\|\nabla f(w_t)\right\|^2\right] + \frac{L(1+\tau_t)\sigma^2}{KN}\tilde{\eta}^2 + (H_1 d_t + H_2\tau_t + H_3)\tilde{\eta}^3$$

$$+ 2\tau_t\sigma L^2\tilde{\eta}^3 \sqrt{\frac{\alpha l}{N} \sum_{j=\max\{t-l_{\max,T},1\}}^{t-1} \sum_{i'=1}^{N} \mathbb{E}\left[\left\|\nabla f(w_{j-\tau(j,i')})\right\|^2\right]}$$

$$+ \frac{(4L^2+\rho\delta)}{N}\tilde{\eta}^3 \sum_{i=1}^{N} \tau(t,i)\left(\sum_{j=t-\tau(t,i)}^{t-1} \mathbb{E}\left[\left\|g_j\right\|^2\right]\right) - \frac{\tilde{\eta}}{2}\left(1 - 2L\tilde{\eta}\right)\mathbb{E}\left[\left\|g_t\right\|^2\right] \qquad (28)$$

$$+ \sigma L\tilde{\eta}^2\tau_t\sqrt{\mathbb{E}\left[\left\|\nabla f(w_t)\right\|^2\right]} + \frac{8\alpha L^2(K-1)\tilde{\eta}^3}{KN} \sum_{i=1}^{N} \mathbb{E}\left[\left\|\nabla f(w_{t-\tau(t,i)})\right\|^2\right],$$

*where $H_1 = \frac{(4L^2+\rho\delta)\sigma^2}{KN}$, $H_2 = 2L^2 l_{\max,T}\sigma\sqrt{\beta + \frac{\sigma^2}{KN}}$ and $H_3 = \frac{4(K-1)L^2(2\beta+\sigma^2/K)}{K}$ .*

**Proof of the descent lemma.** According to the update rule in (4) and $L$-smoothness,

$$f(w_{t+1}) - f(w_t)$$

$$\leq \langle\nabla f(w_t), w_{t+1} - w_t\rangle + \frac{L}{2}\left\|w_{t+1} - w_t\right\|^2$$

$$= -\tilde{\eta}\left\langle\nabla f(w_t), \frac{1}{KN}\sum_{k,i}\tilde{\nabla}f_i(w_{t-\tau(t,i),k}^i)\right\rangle + \frac{L\tilde{\eta}^2}{2}\left\|\frac{1}{KN}\sum_{k,i}\tilde{\nabla}f_i(w_{t-\tau(t,i),k}^i)\right\|^2$$

$$= \underbrace{-\tilde{\eta}\left\langle\nabla f(w_t), \frac{1}{KN}\sum_{k,i}e_{t-\tau(t,i),k}^i\right\rangle}_{\mathcal{T}_1} \underbrace{-\tilde{\eta}\left\langle\nabla f(w_t), \frac{1}{KN}\sum_{k,i}\nabla f_i(w_{t-\tau(t,i),k}^i)\right\rangle}_{\mathcal{T}_2}$$

$$+ \underbrace{\frac{L\tilde{\eta}^2}{2}\left\|\frac{1}{KN}\sum_{k,i}\tilde{\nabla}f_i(w_{t-\tau(t,i),k}^i)\right\|^2}_{\mathcal{T}_3}.$$

### C.3.1 Bounding the first term

Due to reuse of noisy updates, $e^i_{t-\tau(t,i),k}$ is correlated with $w_t$ and $\mathbb{E}[\mathcal{T}_1]$ is not necessarily zero. Unrolling one summand of $\mathcal{T}_1$,

$$-\tilde{\eta}\left\langle \nabla f(w_t), e^i_{t-\tau(t,i),k} \right\rangle = \underbrace{-\tilde{\eta}\left\langle \nabla f(w_t) - \nabla f(w_{t-\tau(t,i)}), e^i_{t-\tau(t,i),k} \right\rangle}_{\mathcal{U}_1}$$

$$\underbrace{-\tilde{\eta}\left\langle \nabla f(w_{t-\tau(t,i)}), e^i_{t-\tau(t,i),k} \right\rangle}_{\mathcal{U}_2}.$$

Since $w_{t-\tau(t,i)}$ and $e^i_{t-\tau(t,i),k}$ are independent, we have $\mathbb{E}[\mathcal{U}_2] = 0$. Plugging $z = -e^i_{t-\tau(t,i),k}$ into Lemma C.1,

$$\mathbb{E}[\mathcal{U}_1]$$

$$\leq \mathbb{E}\left[-\tilde{\eta}\left\langle \nabla^2 f(w_{t-\tau(t,i)})(w_t - w_{t-\tau(t,i)}), e^i_{t-\tau(t,i),k} \right\rangle\right] + \frac{1}{2}\rho\delta\tilde{\eta}\mathbb{E}\left[\left\|w_t - w_{t-\tau(t,i)}\right\|^2\right]$$

$$= \underbrace{\tilde{\eta}^2\mathbb{E}\left[\left\langle \nabla^2 f(w_{t-\tau(t,i)})\frac{1}{KN}\sum_{j=t-\tau(t,i)}^{t-1}\sum_{k',i'}\nabla f_{i'}(w^{i'}_{j-\tau(j,i')}), e^i_{t-\tau(t,i),k} \right\rangle\right]}_{\mathcal{V}_1}$$

$$+ \underbrace{\tilde{\eta}^2\frac{1}{KN}\sum_{j=t-\tau(t,i)}^{t-1}\sum_{k',i'}\mathbb{E}\left[\left\langle \nabla^2 f(w_{t-\tau(t,i)})e^{i'}_{j-\tau(j,i),k'}, e^i_{t-\tau(t,i),k} \right\rangle\right]}_{\mathcal{V}_2}$$

$$+ \frac{1}{2}\rho\delta\tilde{\eta}\underbrace{\mathbb{E}\left[\left\|w_t - w_{t-\tau(t,i)}\right\|^2\right]}_{\mathcal{V}_3}.$$

Using the identity $e^i_{t-\tau(t,i),k} = e^i_{t-1-\tau(t-1,i),k} = \cdots = e^i_{t-\tau(t,i)-\tau(t-\tau(t,i),i),k}$ and independence of $e^i_{j,k}$ and $e^{i'}_{j',k'}$ for all $i \neq i'$ or $k \neq k'$, we can bound $\mathcal{V}_2$.

$$\mathcal{V}_2 = \frac{\tilde{\eta}}{KN}\tau(t,i)\mathbb{E}\left[\left\langle \nabla^2 f(w_{t-\tau(t,i)})e^i_{t-\tau(t,i),k}, e^i_{t-\tau(t,i),k} \right\rangle\right] \leq \tau(t,i)\frac{\tilde{\eta}^2 L}{KN}\sigma^2,$$

where the second inequality uses $L$-smoothness of $f$. Note that $\nabla f_{i'}(w_{j-\tau(j,i')})$ can be split as $\nabla f_{i'}(w_{j-\tau(j,i')}) - \nabla f_{i'}(w_t) + \nabla f_{i'}(w_t)$. Further using Cauchy-Schwartz inequality $\mathbb{E}[\langle X,Y\rangle] \leq \sqrt{\mathbb{E}\left[\|X\|^2\right]\mathbb{E}\left[\|Y\|^2\right]}$ and $L$-smoothness, we can bound $\mathcal{V}_1$ in the following way.

$$\mathcal{V}_1 = \tilde{\eta}^2\frac{1}{KN}\sum_{j=t-\tau(t,i)}^{t-1}\sum_{k',i'}\mathbb{E}\left[\left\langle \nabla^2 f(w_{t-\tau(t,i)})\left(\nabla f_{i'}(w_{j-\tau(j,i')}) - \nabla f_{i'}(w_t)\right), e^i_{t-\tau(t,i),k} \right\rangle\right]$$

$$+ \tilde{\eta}^2\tau(t,i)\mathbb{E}\left[\left\langle \nabla^2 f(w_{t-\tau(t,i)})\nabla f(w_t), e_{i,t-\tau(t,i)} \right\rangle\right]$$

$$\leq \frac{\sigma L\tilde{\eta}^2}{N}\underbrace{\sum_{j=t-\tau(t,i)}^{t-1}\sum_{i'=1}^{N}\sqrt{\mathbb{E}\left[\left\|\nabla f_{i'}(w_t) - \nabla f_{i'}(w_{j-\tau(j,i')})\right\|^2\right]}}_{\mathcal{V}_4}$$

$$+ \sigma L\tilde{\eta}^2\tau(t,i)\sqrt{\mathbb{E}\left[\|\nabla f(w_t)\|^2\right]}.$$

Note that for all $t-\tau(t,i) \leq j \leq t-1$ and $i' \in [N]$, $t-(j-\tau(j,i')) \leq l_{\max,T}$. By $L$-smoothness and Lemma C.3, we obtain an upper bound for $\mathcal{V}_4$.

$$\mathbb{E}[\mathcal{V}_4] \leq \frac{\sigma L^2\tilde{\eta}^2}{N}\sum_{j=t-\tau(t,i)}^{t-1}\sum_{i'=1}^{N}\sqrt{\mathbb{E}\left[\left\|w_t - w_{j-\tau(j,i')}\right\|^2\right]}$$

$$\leq 2\tau(t,i)\sigma L^2\tilde{\eta}^3\sqrt{\frac{\alpha l_{\max,T}}{N}\sum_{j=\max\{t-l_{\max,T},1\}}^{t-1}\sum_{i'=1}^{N}\mathbb{E}\left[\left\|\nabla f(w_{j-\tau(j,i')})\right\|^2\right]}$$

$$+2\sigma L^2 l_{\max,T}\tau(t,i)\tilde{\eta}^3\sqrt{\beta+\frac{\sigma^2}{KN}},$$

where the last in equality uses $\sqrt{x+y} \leq \sqrt{x} + \sqrt{y}, \forall x, y \geq 0$. Now we can obtain an upper bound for $\mathcal{V}_1$.

$$\mathcal{V}_1 \leq 2\tau(t,i)\sigma L^2\tilde{\eta}^3\sqrt{\frac{\alpha l_{\max,T}}{N}\sum_{j=\max\{t-l_{\max,T},1\}}^{t-1}\sum_{i'=1}^{N}\mathbb{E}\left[\left\|\nabla f(w_{j-\tau(j,i')})\right\|^2\right]}$$

$$+\sigma L\tilde{\eta}^2\tau(t,i)\sqrt{\mathbb{E}\left[\left\|\nabla f(w_t)\right\|^2 \mid \mathcal{F}_t\right]}+2\sigma L^2 l_{\max,T}\tau(t,i)\tilde{\eta}^3\sqrt{\beta+\frac{\sigma^2}{KN}}.$$

We proceed to bound $\mathcal{V}_3$ by Jensen's inequality.

$$\mathcal{V}_3 = \mathbb{E}\left[\left\|\frac{\tilde{\eta}}{KN}\sum_{j=t-\tau(t,i)}^{t-1}\sum_{k',i'}\tilde{\nabla}f_{i'}(w^{i'}_{j-\tau(j,i'),k'})\right\|^2\right]$$

$$= \mathbb{E}\left[\left\|\tilde{\eta}\sum_{j=t-\tau(t,i)}^{t-1}g_j+\frac{\tilde{\eta}}{KN}\sum_{j=t-\tau(t,i)}^{t-1}\sum_{k',i'}e^{i'}_{j-\tau(j,i'),k'}\right\|^2\right]$$

$$\leq 2\tilde{\eta}^2\mathbb{E}\left[\left\|\sum_{j=t-\tau(t,i)}^{t-1}g_j\right\|^2\right]+2\tilde{\eta}^2\mathbb{E}\left[\left\|\frac{1}{KN}\sum_{j=t-\tau(t,i)}^{t-1}\sum_{k',i'}e^{i'}_{j-\tau(j,i'),k'}\right\|^2\right]$$

$$\leq 2\tau(t,i)\tilde{\eta}^2\sum_{j=t-\tau(t,i)}^{t-1}\mathbb{E}\left[\left\|g_j\right\|^2\right]+\frac{2\tau(t,i)^2\sigma^2\tilde{\eta}^2}{KN}.$$

Combining $\mathcal{V}_1$ to $\mathcal{V}_3$, we have

$$\mathbb{E}\left[\mathcal{U}_1\right] \leq \frac{\tau(t,i)L\sigma^2}{KN}\tilde{\eta}^2+\frac{\rho\delta\tau(t,i)^2\sigma^2}{KN}\tilde{\eta}^3+2\sigma L^2 l_{\max,T}\tau(t,i)\tilde{\eta}^3\sqrt{\beta+\frac{\sigma^2}{KN}}$$

$$+2\tau(t,i)\sigma L^2\tilde{\eta}^3\sqrt{\frac{\alpha l_{\max,T}}{N}\sum_{j=\max\{t-l_{\max,T},1\}}^{t-1}\sum_{i'=1}^{N}\mathbb{E}\left[\left\|\nabla f(w_{j-\tau(j,i')})\right\|^2\right]}$$

$$+\sigma L\tilde{\eta}^2\tau(t,i)\sqrt{\mathbb{E}\left[\left\|\nabla f(w_t)\right\|^2\right]}+\rho\delta\tau(t,i)\tilde{\eta}^3\sum_{j=t-\tau(t,i)}^{t-1}\mathbb{E}\left[\left\|g_j\right\|^2\right].$$

Finally we bound the expectation of $\mathcal{T}_1$ and conclude this section.

$$\mathbb{E}\left[\mathcal{T}_1\right] \leq \frac{\tau_t L\sigma^2}{KN}\tilde{\eta}^2+\frac{\rho\delta d_t\sigma^2}{KN}\tilde{\eta}^3+2\sigma L^2 l_{\max,T}\tau_t\tilde{\eta}^3\sqrt{\beta+\frac{\sigma^2}{KN}}$$

$$+2\tau_t\sigma L^2\tilde{\eta}^3\sqrt{\frac{\alpha l_{\max,T}}{N}\sum_{j=\max\{t-l_{\max,T},1\}}^{t-1}\sum_{i'=1}^{N}\mathbb{E}\left[\left\|\nabla f(w_{j-\tau(j,i')})\right\|^2\right]}$$

$$+\sigma L\tilde{\eta}^2\tau_t\sqrt{\mathbb{E}\left[\left\|\nabla f(w_t)\right\|^2\right]}+\frac{\rho\delta\tilde{\eta}^3}{N}\sum_{i=1}^{N}\tau(t,i)\sum_{j=t-\tau(t,i)}^{t-1}\mathbb{E}\left[\left\|g_j\right\|^2\right].$$

## C.4 Bounding the second term

Since $\langle x, y \rangle = \frac{1}{2} \|x\|^2 + \frac{1}{2} \|y\|^2 - \frac{1}{2} \|x - y\|^2$,

$$\mathcal{T}_2 = -\frac{\tilde{\eta}}{2} \|\nabla f(w_t)\|^2 - \frac{\tilde{\eta}}{2} \|g_t\|^2 + \frac{\tilde{\eta}}{2} \underbrace{\left\| \nabla f(w_t) - \frac{1}{KN} \sum_{k,i} \nabla f_i(w^i_{t-\tau(t,i),k}) \right\|^2}_{\mathcal{U}_3}.$$

Next we bound $\mathcal{U}_3$. Note that $\nabla f(w_t) - \frac{1}{KN} \sum_{k,i} \nabla f_i(w^i_{t-\tau(t,i),k})$ can be split as

$$\nabla f(w_t) - \frac{1}{KN} \sum_{k,i} \nabla f_i(w^i_{t-\tau(t,i),k})$$

$$= \frac{1}{N} \sum_{i=1}^N \left( \nabla f_i(w_t) - \nabla f_i(w_{t-\tau(t,i)}) \right) + \frac{1}{KN} \sum_{k,i} \left( \nabla f_i(w^i_{t-\tau(t,i),k}) - \nabla f_i(w_{t-\tau(t,i)}) \right).$$

By Jensen's inequality and $L$-smoothness,

$$\mathbb{E}[\mathcal{U}_3]$$

$$\leq \frac{2L^2}{N} \sum_{i=1}^N \mathbb{E}\left[ \|w_t - w_{t-\tau(t,i)}\|^2 \right] + \frac{2L^2}{KN} \sum_{k,i} \left\| w^i_{t-\tau(t,i),k} - w_{t-\tau(t,i)} \right\|^2$$

$$\leq \frac{4L^2 \tilde{\eta}^2}{N} \sum_{i=1}^N \tau(t,i) \sum_{j=t-\tau(t,i)}^{t-1} \mathbb{E}\left[ \|g_j\|^2 \right] + \frac{4d_t L^2 \sigma^2 \tilde{\eta}^2}{KN}$$

$$+ \frac{8\alpha L^2 (K-1)\tilde{\eta}^2}{KN} \sum_{i=1}^N \mathbb{E}\left[ \|\nabla f(w_{t-\tau(t,i)})\|^2 \right] + \frac{8L^2 (K-1)\tilde{\eta}^2 \beta}{K} + \frac{4L^2 (K-1)\tilde{\eta}^2 \sigma^2}{K^2},$$

where we apply Lemma C.2 and plug in the bound for $\mathcal{V}_3$ in the second inequality. To sum up, we derive the following bound for the expectation of $\mathcal{T}_2$.

$$\mathbb{E}[\mathcal{T}_2]$$

$$\leq -\frac{\tilde{\eta}}{2} \mathbb{E}\left[ \|\nabla f(w_t)\|^2 \right] - \frac{\tilde{\eta}}{2} \mathbb{E}\left[ \|g_t\|^2 \right]$$

$$+ \frac{4L^2 \tilde{\eta}^3}{N} \sum_{i=1}^N \tau(t,i) \sum_{j=t-\tau(t,i)}^{t-1} \mathbb{E}\left[ \|g_j\|^2 \right] + \frac{4d_t L^2 \sigma^2 \tilde{\eta}^3}{KN}$$

$$+ \frac{8\alpha L^2 (K-1)\tilde{\eta}^3}{KN} \sum_{i=1}^N \mathbb{E}\left[ \|\nabla f(w_{t-\tau(t,i)})\|^2 \right] + \frac{8L^2 (K-1)\tilde{\eta}^3 \beta}{K} + \frac{4L^2 (K-1)\tilde{\eta}^3 \sigma^2}{K^2}.$$

## C.5 Bounding the third term

By Jensen's inequality,

$$\mathbb{E}[\mathcal{T}_3] = \frac{L\tilde{\eta}^2}{2} \mathbb{E}\left[ \left\| g_t + \frac{1}{KN} \sum_{k,i} e^i_{t-\tau(t,i),k} \right\|^2 \right]$$

$$\leq L\tilde{\eta}_t^2 \mathbb{E}\left[ \|g_t\|^2 \right] + L\tilde{\eta}^2 \mathbb{E}\left[ \left\| \frac{1}{KN} \sum_{i=1}^N e^i_{t-\tau(t,i),k} \right\|^2 \right]$$

$$\leq L\tilde{\eta}^2 \mathbb{E}\left[ \|g_t\|^2 \right] + \frac{L\sigma^2 \tilde{\eta}^2}{KN}.$$

Combining the results in Appendix C.3.1, Appendix C.4 and Appendix C.5, we have

$$\mathbb{E}\left[f(w_{t+1})\right] - \mathbb{E}\left[f(w_t)\right]$$

$$\leq -\frac{\tilde{\eta}}{2}\mathbb{E}\left[\|\nabla f(w_t)\|^2\right] + \frac{L(1+\tau_t)\sigma^2}{KN}\tilde{\eta}^2 + (H_1 d_t + H_2 \tau_t + H_3)\tilde{\eta}^3$$

$$+ 2\tau_t \sigma L^2 \tilde{\eta}^3 \sqrt{\frac{\alpha l_{\max,T}}{N} \sum_{j=\max\{t-l_{\max,T},1\}}^{t-1} \sum_{i'=1}^{N} \mathbb{E}\left[\left\|\nabla f(w_{j-\tau(j,i')})\right\|^2\right]}$$

$$+ \frac{(4L^2+\rho\delta)}{N}\tilde{\eta}^3 \sum_{i=1}^{N} \tau(t,i)\left(\sum_{j=t-\tau(t,i)}^{t-1} \mathbb{E}\left[\|g_j\|^2\right]\right) - \frac{\tilde{\eta}}{2}\left(1 - 2L\tilde{\eta}\right)\mathbb{E}\left[\|g_t\|^2\right]$$

$$+ \sigma L\tilde{\eta}^2 \tau_t \sqrt{\mathbb{E}\left[\|\nabla f(w_t)\|^2\right]} + \frac{8\alpha L^2(K-1)\tilde{\eta}^3}{KN}\sum_{i=1}^{N} \mathbb{E}\left[\left\|\nabla f(w_{t-\tau(t,i)})\right\|^2\right],$$

where $H_1 = \frac{(4L^2+\rho\delta)\sigma^2}{KN}$, $H_2 = 2L^2 l_{\max,T}\sigma\sqrt{\beta + \frac{\sigma^2}{KN}}$ and $H_3 = \frac{4(K-1)L^2(2\beta+\sigma^2/K)}{K}$ . Now we have proved the descent lemma.

## C.6  Deriving the convergence rate

Since $\sum_{t=1}^{T-1} \mathbb{E}\left[\left\|\nabla f(w_{t-\tau(t,i)})\right\|^2\right] \leq (1 + \max_{1\leq t\leq T-1}\{\tau(t,i)\})\sum_{t=1}^{T-1}\mathbb{E}\left[\|\nabla f(w_t)\|^2\right]$, the telescoping sum of (28) from $t=1$ to $T-1$ satisfies

$$\mathbb{E}\left[f(w_T)\right] - \mathbb{E}\left[f(w_1)\right]$$

$$\leq \underbrace{-\tilde{\eta}\left(\frac{1}{2} - \frac{8\alpha L^2(K-1)\bar{\tau}_{\max,T}\tilde{\eta}^2}{K}\right)\sum_{t=1}^{T-1}\mathbb{E}\left[\|\nabla f(w_t)\|^2\right] + \frac{L(T+s_T)\sigma^2}{KN}\tilde{\eta}^2}_{\mathcal{V}_5}$$

$$+ (H_1 r_T + H_2 s_T + H_3 T)\tilde{\eta}^3$$

$$+ \underbrace{2\sum_{t=1}^{T-1}\tau_t\sigma L^2\tilde{\eta}^3\sqrt{\frac{\alpha l_{\max,T}}{N}\sum_{j=\max\{t-l_{\max,T},1\}}^{t-1}\sum_{i'=1}^{N}\mathbb{E}\left[\left\|\nabla f(w_{j-\tau(j,i')})\right\|^2\right]}}_{\mathcal{V}_6} \qquad (29)$$

$$+ \underbrace{\frac{(4L^2+\rho\delta)}{N}\tilde{\eta}^3\sum_{t=1}^{T-1}\sum_{i=1}^{N}\tau(t,i)\left(\sum_{j=t-\tau(t,i)}^{t-1}\mathbb{E}\left[\|g_j\|^2\right]\right) - \frac{\tilde{\eta}}{2}\left(1 - 2L\tilde{\eta}\right)\sum_{t=1}^{T-1}\mathbb{E}\left[\|g_t\|^2\right]}_{\mathcal{V}_7}$$

$$+ \underbrace{\sigma L\tilde{\eta}^2\sum_{t=1}^{T-1}\tau_t\sqrt{\mathbb{E}\left[\|\nabla f(w_t)\|^2\right]}}_{\mathcal{V}_8}.$$

Next, we bound $\mathcal{V}_5$ to $\mathcal{V}_8$ respectively. When $\tilde{\eta} \leq \sqrt{\frac{1}{32\alpha\bar{\tau}_{\max,T}L^2}}$, we have

$$\mathcal{V}_5 \leq -\frac{\tilde{\eta}}{4}\sum_{t=1}^{T}\mathbb{E}\left[\|\nabla f(w_t)\|^2\right].$$

By Jensen's inequality $(\sum_{t=1}^{T}\sqrt{a_t})^2 \leq T\sum_{t=1}^{T} a_t$, i.e. $\sum_{t=1}^{T}\sqrt{a_t} \leq \sqrt{T\sum_{t=1}^{T} a_t}$,

$$\mathcal{V}_6 \leq 2\bar{\tau}_{\max,T}\sigma L^2\tilde{\eta}^3\sum_{t=1}^{T}\sqrt{\frac{\alpha l_{\max,T}}{N}\sum_{j=\max\{t-l_{\max,T},1\}}^{t-1}\sum_{i'=1}^{N}\mathbb{E}\left[\left\|\nabla f(w_{j-\tau(j,i')})\right\|^2\right]}$$

$$\leq 2\bar{\tau}_{\max,T}\sigma L^2\tilde{\eta}^3\sqrt{\frac{\alpha l_{\max,T}^2 T}{N}\sum_{t=1}^{T}\sum_{i'=1}^{N}\mathbb{E}\left[\left\|\nabla f(w_{t-\tau(t,i')})\right\|^2\right]}$$

$$\leq 2\bar{\tau}_{\max,T}l_{\max,T}\sigma L^2\tilde{\eta}^3\sqrt{\alpha T\bar{\tau}_{\max,T}\sum_{t=1}^{T}\mathbb{E}\left[\left\|\nabla f(w_t)\right\|^2\right]}.$$

When $\tilde{\eta} \leq \frac{1}{4L}$ and $\tilde{\eta} \leq \sqrt{\frac{1}{2(4L^2+\rho\delta)\bar{d}_{\max,T}}}$ , we can bound $\mathcal{V}_7$ as

$$\mathcal{V}_7 \leq -\frac{\tilde{\eta}}{2}\left[1 - 2L\tilde{\eta} - (4L^2 + \rho\delta)\bar{d}_{\max,T}\tilde{\eta}^2\right]\left(\sum_{t=1}^{T}\mathbb{E}\left[\|g_t\|^2\right]\right) \leq 0.$$

Using $\tau_t \leq \bar{\tau}_{\max,T}$ for all $1 \leq t \leq T-1$ and Jensen's inequality, we have

$$\mathcal{V}_8 \leq \sigma L\tilde{\eta}^2\bar{\tau}_{\max,T}\sqrt{T\sum_{t=1}^{T}\mathbb{E}\left[\|\nabla f(w_t)\|^2\right]}.$$

After minor rearrangement, (29) can be simplified as

$$\sum_{t=1}^{T-1}\mathbb{E}\left[\|\nabla f(w_t)\|^2\right]$$

$$\leq \frac{4}{\tilde{\eta}}(f(w_1) - f(w_T)) + \frac{4L(T+s_T)\sigma^2}{KN}\tilde{\eta} + 4(H_1 r_T + H_2 s_T + H_3 T)\tilde{\eta}^2 \quad (30)$$

$$+ 4\left(1 + 2l_{\max,T}\sqrt{\alpha\bar{\tau}_{\max,T}}L\tilde{\eta}\right)\sqrt{T}\sigma L\bar{\tau}_{\max,T}\tilde{\eta}\sqrt{\sum_{t=1}^{T}\mathbb{E}\left[\|\nabla f(w_t)\|^2\right]}.$$

By Lemma B.1, $\mathbb{E}\left[\|\nabla f(w_T)\|^2\right] \leq 2L(\mathbb{E}\left[f(w_T)\right] - f^*)$. Multiplying both sides by $\tilde{\eta}$ and further using $\tilde{\eta} \leq \frac{1}{4L}$, we have

$$\tilde{\eta}\mathbb{E}\left[\|\nabla f(w_T)\|^2\right] \leq 2L\tilde{\eta}(\mathbb{E}\left[f(w_T)\right] - f^*)$$

$$\leq \frac{1}{2}(\mathbb{E}\left[f(w_T)\right] - f^*).$$

Then adding $\mathbb{E}\left[\|\nabla f(w_T)\|^2\right]$ to the LHS and $\frac{4}{\tilde{\eta}_t}\mathbb{E}\left[f(w_T)\right] - f(w_*)$ to the RHS, (30) can be further simplified as

$$\sum_{t=1}^{T}\mathbb{E}\left[\|\nabla f(w_t)\|^2\right]$$

$$\leq \frac{4}{\tilde{\eta}}(f(w_1) - f^*) + \frac{4L(T+s_T)\sigma^2}{KN}\tilde{\eta} + 4(H_1 r_T + H_2 s_T + H_3 T)\tilde{\eta}^2$$

$$+ 4\left(1 + 2l_{\max,T}\sqrt{\alpha\bar{\tau}_{\max,T}}L\tilde{\eta}\right)\sqrt{T}\sigma L\bar{\tau}_{\max,T}\tilde{\eta}\sqrt{\sum_{t=1}^{T}\mathbb{E}\left[\|\nabla f(w_t)\|^2\right]}.$$

Define $\Omega_T = \sum_{t=1}^{T}\mathbb{E}\left[\|\nabla f(w_t)\|^2\right]$, $H_4 = \frac{4}{\tilde{\eta}}(f(w_1) - f^*) + \frac{4L(T+s_T)\sigma^2}{KN}\tilde{\eta} + 4(H_1 r_T + H_2 s_T + H_3 T)\tilde{\eta}^2$, $H_5 = 4\left(1 + 2l_{\max,T}\sqrt{\alpha\bar{\tau}_{\max,T}}L\tilde{\eta}\right)\sqrt{T}\sigma L\bar{\tau}_{\max,T}\tilde{\eta}$. Now we solve the following inequality.

$$\Omega_T \leq H_5\sqrt{\Omega_T} + H_4 \Rightarrow \sqrt{\Omega_T} \leq \frac{1}{2}(H_5 + \sqrt{H_5^2 + 4H_4}) \Rightarrow \Omega_T \leq H_5^2 + 2H_4.$$

Therefore,

$$\frac{1}{T}\sum_{t=1}^{T}\mathbb{E}\left[\|\nabla f(w_t)\|^2\right] \leq \frac{8}{T\tilde{\eta}}(f(w_1) - f^*) + \frac{8L(1 + \bar{\tau}_T)\sigma^2}{KN}\tilde{\eta}$$
$$+ 4(H_1\bar{d}_T + H_2\bar{\tau}_T + H_3)\tilde{\eta}^2 \qquad (31)$$
$$+ 32\sigma^2\bar{\tau}_{\max,T}^2 L^2\tilde{\eta}^2 + 128\alpha\sigma^2\bar{\tau}_{\max,T}^3 l_{\max,T}^2 L^4\tilde{\eta}^4.$$

Let $\tilde{\eta} = c_0\sqrt{\frac{KN}{TL(1+\bar{\tau}_T)}}$, where $c_0$ is a constant and $0 < c_0 \leq 1$. We will show that for $T \geq \max\{\frac{64\alpha^2 KNL^3}{L^2+\rho\delta}, 16LNK, t_0\}$, the following holds.

$$\tilde{\eta} \leq \sqrt{\frac{1}{2(4L^2 + \rho\delta)\bar{d}_{\max,T}}}, \qquad (32)$$

$$\tilde{\eta} \leq \frac{1}{4L}, \qquad (33)$$

$$\tilde{\eta} \leq \sqrt{\frac{1}{32\alpha\bar{\tau}_{\max,T}L^2}}. \qquad (34)$$

By Assumption 9, when $T \geq t_0$, $\tau(t,i) \leq \frac{1}{4}\sqrt{\frac{LT}{NK(\rho\delta+L^2)}}$, $\forall t \leq T$. Thus (32) holds. Since $T \geq 16LNK$, (33) holds. To verify (34), we only have to show

$$\tilde{\eta}^2 \leq \frac{1}{32\alpha\bar{\tau}_{\max,T}L^2} \Leftrightarrow \bar{\tau}_{\max,T} \leq \frac{T(1+\bar{\tau}_T)}{32\alpha c_0^2 LKN}.$$

Still by Assumption 9, we only have to show

$$\frac{1}{4}\sqrt{\frac{LT}{NK(\rho\delta+L^2)}} \leq \frac{T(1+\bar{\tau}_T)}{32\alpha LKN},$$

which holds for $T \geq \frac{64\alpha^2 KNL^3}{L^2+\rho\delta}$. Now we only have to plug the value of $\tilde{\eta}$ into (31) and make minor adjustments. Still by Assumption 9, we have

$$(l_{\max,T}\tilde{\eta})^2 = \mathcal{O}\left(\frac{1}{(\rho\delta + L^2)(1 + \bar{\tau}_T)}\right).$$

Since $\min_{1\leq t\leq T}\mathbb{E}\left[\|\nabla f(w_t)\|^2\right] \leq \frac{1}{T}\sum_{t=1}^{T}\mathbb{E}\left[\|\nabla f(w_t)\|^2\right]$, we have

$$\min_{1\leq t\leq T}\mathbb{E}\left[\|\nabla f(w_t)\|^2\right] = \mathcal{O}\left(\sqrt{\frac{(1 + \bar{\tau}_T)L}{TKN}}(f(w_1) - f^* + \sigma^2) + \frac{A_6}{T}\right),$$

where

$$A_6 = \frac{1}{(1 + \bar{\tau}_T)}\left[\sigma^2\bar{\tau}_{\max,T}^2 NKL\left(1 + \frac{\alpha\bar{\tau}_{\max,T}L^2}{(\rho\delta + L^2)(1 + \bar{\tau}_T)}\right) + \frac{(L^2 + \rho\delta)\sigma^2}{L}\bar{d}_T\right.$$
$$\left. + (K-1)NL(\beta + \sigma^2/K)\right] + LKN\tau_{\max,T}\sigma\sqrt{\beta + \frac{\sigma^2}{KN}}.$$

Now we have completed the proof of Theorem C.1. The following corollary is the same as Theorem 6.1, which holds under the assumption of bounded number of inactive rounds.

**Corollary C.1** (Bounded number of inactive rounds). *Assume that Assumptions 1, 2, and 5 to 7 hold. Further assume that the device availability sequence $\tau(t,i)$ satisfies Assumption 8 and $\tau(1,i) = 0$ for all $i \in [N]$. By using a learning rate $\eta = \sqrt{\frac{N}{KTL(1+\bar{\nu})}}$, for $T \geq \max\{32\alpha LNK, 16LNK, \frac{8KN\nu_{\max}^2(L^2+\rho\delta)}{L}\}$, after $T - 1$ communication rounds, MIFA satisfies:*

$$\min_{1\leq t\leq T}\mathbb{E}_\xi\left[\|\nabla f(w_t)\|^2\right] = \mathcal{O}\left(\sqrt{\frac{(1 + \bar{\nu})L}{TKN}}(f(w_1) - f^* + \sigma^2) + \frac{A_4 + A_5}{T}\right),$$

*where $f^*$ is the optimal value, and:*

$$A_4 = NKL\left(\alpha\sigma^2\bar{\nu} + \frac{\sigma^2\nu_{\max}}{\sqrt{KN}} + \sigma\nu_{\max}\sqrt{\beta}\right) + \frac{(L^2 + \rho\delta)\sigma^2\nu_{\max}}{L},$$

$$A_5 = \frac{(K-1)NL(\beta + \sigma^2/K)}{\bar{\nu} + 1}.$$

*Proof.* We first show that (32) to (34) hold. (32) holds because when $T \geq \frac{8KN\nu_{\max}^2(L^2+\rho\delta)}{L}$,

$$\tilde{\eta} \leq \sqrt{\frac{1}{2(4L^2 + \rho\delta)\nu_{\max}^2}}.$$

Also, (33) holds when $T \geq 16LNK$. (34) holds when $T \geq 32\alpha LNK$. Therefore, (31) holds and it can be further simplified as

$$\frac{1}{T}\sum_{t=1}^{T}\mathbb{E}\left[\|\nabla f(w_t)\|^2\right] \leq \frac{4}{T\tilde{\eta}}(f(w_1) - f^*) + \frac{4L(1+\bar{\nu})\sigma^2}{KN}\tilde{\eta}$$

$$+ 4\left[H_1\left(\frac{1}{N}\sum_{i=1}^{N}\nu_i^2\right) + H_2\bar{\nu} + H_3\right]\tilde{\eta}^2 \tag{35}$$

$$+ 32\sigma^2\bar{\nu}^2L^2\tilde{\eta}^2 + 512\alpha\sigma^2\bar{\nu}^3\nu_{\max}^2L^4\tilde{\eta}^4.$$

Since $T \geq \frac{8KN\nu_{\max}^2(L^2+\rho\delta)}{L}$,

$$\tilde{\eta}^2\nu_{\max}^2 = \mathcal{O}\left(\frac{1}{(1+\bar{\nu})(L^2+\rho\delta)}\right).$$

Therefore,

$$\alpha\sigma^2\bar{\nu}^3\nu_{\max}^2L^4\tilde{\eta}^4 = \mathcal{O}\left(\frac{\alpha\sigma^2\bar{\nu}^2L^4}{L^2+\rho\delta}\tilde{\eta}^2\right) = \mathcal{O}\left(\frac{\alpha\sigma^2\bar{\nu}LKN}{T}\right).$$

Besides,

$$32\sigma^2\bar{\nu}^2L^2\tilde{\eta}^2 = \mathcal{O}\left(\frac{LKN\bar{\nu}\sigma^2}{T}\right),$$

$$H_1\left(\frac{1}{N}\sum_{i=1}^{N}\nu_i^2\right) = \mathcal{O}\left(\frac{L^2+\rho\sigma^2}{L(\bar{\nu}+1)}\left(\frac{1}{N}\sum_{i=1}^{N}\nu_i^2\right)\right) = \mathcal{O}\left(\frac{(L^2+\rho\sigma^2)\nu_{\max}}{L}\right),$$

$$H_2\bar{\nu}\tilde{\eta}^2 = \mathcal{O}\left(\frac{LKN\nu_{\max}}{T}\left(\sigma\sqrt{\beta} + \sigma^2/\sqrt{KN}\right)\right),$$

$$H_3\tilde{\eta}^2 = \mathcal{O}\left(\frac{(K-1)NL(\beta + \sigma^2/K)}{1+\bar{\nu}}\right).$$

Now we have completed the proof of Corollary C.1. $\square$

## D   Proofs in Section 5.1

Our analysis is based on the observation that $\tau(t,i)$ is a truncated geometric random variable with success probability $p_i$ for the Bernoulli participation model.

**Lemma D.1.** *For i.i.d. Bernoulli participation model with participation probabilities $\{p_i\}$, we have $\tau(t,i)$ is a truncated geometric random variable taking values in $\{0, 1, \ldots, t-1\}$.*

*Proof.* Notice that for $k < t$, the event $\{\tau(t,i) \geq k\}$ is equivalent to the event that device $i$ is not active at round $t, t-1, \ldots, t-k+1$, which means

$$\mathbb{P}(\tau(t,i) \geq k) = (1-p_i)^k, \text{ for } k < t.$$

Also, since we have assumed that all devices participate at the first round, we have $\mathbb{P}(\tau(t,i) \geq t) = 0.$ $\square$

## D.1 Proof of Theorem 5.2

*Proof.* By Lemma D.1, we know that for all $k$,

$$\mathbb{P}(\tau(t,i) \geq k) \leq (1 - p_i)^k.$$

For any fixed $0 < \delta_t < 1$, by setting $k = \lceil \frac{\log(1/\delta_t)}{\log(1/(1-p_i))} \rceil$, we have $\mathbb{P}(\tau(t,i) \geq k) \leq \delta_t$. This means with probability at least $1 - \delta_t$, we have

$$\tau(t,i) \leq 1 + \frac{\log(1/\delta_t)}{\log(1/(1-p_i))}.$$

By choosing $\delta_t = \frac{6}{\pi^2} \cdot \frac{\delta}{t^2 N}$ and taking union bound over all $t \geq 1$ and $i \in [N]$, we have with probability at least $1 - \delta$,

$$\tau(t,i) \leq 1 + \frac{\log(\frac{\pi^2}{6} \cdot \frac{t^2 N}{\delta})}{\log(1/(1-p_i))} = 1 + \frac{1}{\log(1/(1-p_i))}\Big[ \log(\frac{\pi^2}{6\delta}) + 2\log t + \log N \Big].$$

Using the inequality that $\frac{1}{\log(1/(1-p_i))} \leq 1/p_i$ (which is tight when $p_i \approx 0$), we further have

$$\tau(t,i) \leq 1 + \frac{1}{p_i}\Big( 2\log t + \log N + \log \frac{\pi^2}{6\delta} \Big) = \mathcal{O}\Big( \frac{1}{p_i}(1 + \log(Nt/\delta)) \Big).$$

For Assumption 4 to hold, We need to find a $t_0$ such that for all $t$,

$$1 + \frac{1}{p_{min}}\Big[ \log(\frac{\pi^2}{6\delta}) + 2\log t + \log N \Big] \leq t_0 + \frac{t}{b}.$$

Solving this inequality, we get

$$t_0 \geq \frac{2}{p_{min}}\Big( \log \frac{2b}{p_{min}} - 1 \Big) + \frac{1}{p_{min}} \log \frac{\pi^2 N}{6\delta} + 1,$$

which is satisfied if

$$t_0 \geq C \frac{1}{p_{min}} \log \frac{bN}{p_{min}\delta}$$

for an absolute constant $C > 0$. $\qquad\square$

## D.2 Proof of Theorem 5.3

*Proof.* By Lemma D.1, we have

$$\mathbb{E}\left[ \tau(t,i) \right] = \sum_{k=1}^{\infty} \mathbb{P}\Big( \tau(t,i) \geq k \Big) = \sum_{k=1}^{t-1}(1 - p_i)^k \leq \frac{1}{p_i}.$$

Therefore, we can upper bound the expectation of $\bar{\tau}_T = \frac{1}{N(T-1)} \sum_{t=1}^{T-1} \sum_{i=1}^{N} \tau(t,i)$ as

$$\mathbb{E}\left[ \bar{\tau}_T \right] \leq \frac{1}{N} \sum_{i=1}^{N} \frac{1}{p_i}.$$

Furthermore, we know that $\tau(t,i)$ is sub-exponential with $\|\tau(t,i)\|_{\psi_1} \leq C_1 \frac{1}{p_i}$ [37]. Then we know that $\bar{\tau}_T - \mathbb{E}\left[ \bar{\tau}_T \right]$ is sub-exponential with $\|\bar{\tau}_T - \mathbb{E}\left[ \bar{\tau}_T \right]\|_{\psi_1} \leq C_2 \frac{1}{N} \sum_{i=1}^{N} \frac{1}{p_i}$. Therefore, by Bernstein's inequality [37], we have with probability at least $1 - \delta$,

$$\bar{\tau}_T - \mathbb{E}\left[ \bar{\tau}_T \right] \leq C_3\Big( \frac{1}{N} \sum_{i=1}^{N} \frac{1}{p_i} \Big) \cdot \max\Big( \log \frac{1}{\delta}, 1 \Big).$$

We conclude that

$$\bar{\tau}_T \leq \Big( \frac{1}{N} \sum_{i=1}^{N} \frac{1}{p_i} \Big) \cdot \mathcal{O}\Big( 1 + \log \frac{1}{\delta} \Big).$$

Remark: $C_1, C_2, C_3 > 0$ are absolute constants. $\qquad\square$

### D.3 Additional Discussion on the Expected Waiting Time

To accomplish a single global update, algorithms such as FedAvg and SCAFFOLD need to receive the local updates from a randomly sampled subset $\mathcal{S}$ of devices. In our setting, the server needs to wait for a few rounds so that all devices in $\mathcal{S}$ become active and return the computation result during the these rounds. For i.i.d. Bernoulli participation model, the expected rounds for the $i$-th device to become active is $1/p_i$. Therefore, the expected rounds for all the devices in $\mathcal{S}$ to become active is at least $\frac{1}{\min\{p_i | i \in \mathcal{S}\}}$.

Denote by $T(\mathcal{S})$ the expected rounds for all the devices in $\mathcal{S}$ to become active, under the setting that $\mathcal{S}$ is randomly selected from $N$ devices without replacement, we have

$$\mathbb{E}_{\mathcal{S}}\left[T(\mathcal{S})\right] \geq \frac{1}{p_{min}} \mathbb{P}_{\mathcal{S}}\left(\text{ the device with minimal } p_i \text{ is selected }\right) = \frac{S}{N}\frac{1}{p_{min}}.$$

## E Proof of Proposition 5.1

*Proof.* This lower bound actually holds even for centralized algorithms. We first show that a lower bound for centralized optimization implies a lower bound on our case. We then analyze the lower bound for the standard optimization setup.

**Number of gradient evaluations.** Assume that we have $N$ devices, and each device respond every $2\tau$ rounds of communication. Then by definition $\bar{\tau}_T = \Theta(\tau)$, and only $\Theta(NKT/\bar{\tau}_T)$ stochastic gradients are evaluated. Hence, the theorem is proved if we can show that no algorithms can output a (potentially random) $w_T$ within $\mathcal{T}$ stochastic gradients evaluations satisfying

$$\mathbb{E}[f(w_T) - f(w^*)] \geq c_0 \frac{\sigma^2}{\mu \mathcal{T}}.$$

**Uncontrained stochastic optimization lower bound.** The constrained version of the above inequality has been formally proved by multiple works (e.g.[2, 29]). These results do not readily applied as we did not assume the function to be Lipschitz continuous. The smooth but not Liptschitz continuous case is a *folklore* in optimization community (e.g. see [12] equation 1.3). We provide a short proof *for completeness* following [8, 44].

For a given $\mu \in (0,1], \sigma > 1$, we consider the following simple one-dimensional function class parameterized by $b$:

$$\min_x \left\{ f_b(x) := \frac{\mu}{2}(x-b)^2 \right\}, \text{ for } b \in [0, 1/2]. \tag{36}$$

Note that $f_b$ is 1-smooth and $\mu$-strongly convex.

Also suppose that for $b \in [0, 1/2]$ the stochastic gradients are of the form:

$$g(x) \sim \nabla f_b(x) + \chi_b, \, \mathbb{E}[g(x)] = \nabla f_b(x), \text{ and } \mathbb{E}[|g(x) - \nabla f_b(x)|^2] \leq \sigma^2. \tag{37}$$

Note that the function class (36) has optimum value $f_b(b) = 0$. Thus, we want to prove the following:

**Theorem E.1.** *There exists a distribution $\chi_b$ such that the stochastic gradients satisfy* (37). *Further, for any (possibly randomized) algorithm $\mathcal{A}$, define $\mathcal{A}_k(f_b + \chi_b)$ to be the output of the algorithm $\mathcal{A}$ after $k$ queries to the stochastic gradient $g(x)$, then:*

$$\max_{b \in [0, 1/2]} \mathbb{E}[f_b(\mathcal{A}_k(f_b + \chi_b))] \geq \frac{c_0 \sigma^2}{k\mu}.$$

We assume the algorithm of interest is stable, i.e. $\max_{b \in [0, 1/2]} \mathbb{E}[f_b(\mathcal{A}_k(f_b + \chi_b))] \leq \infty$. Otherwise, the theorem is true.

Let $\mathcal{A}_k(f_b + \chi_b)$ denote the output of any possibly randomized algorithm $\mathcal{A}$ after processing $k$ stochastic gradients of the function $f_b$ (with noise drawn i.i.d. from distribution $\chi_b$). Similarly, let $\mathcal{D}_k(f_b + \chi_b)$ denote the output of a *deterministic* algorithm after processing the $k$ stochastic gradients. Then from Yao's minimax principle we know that for any fixed distribution $\mathcal{B}$ over $[0, 1/2]$,

$$\min_{\mathcal{A}} \max_{b \in [0, 1/2]} \mathbb{E}_{\mathcal{A}}[\mathbb{E}_{\chi_b} f_b(\mathcal{A}_k(f_b + \chi_b))] \geq \min_{\mathcal{D}} \mathbb{E}_{b \sim \mathcal{B}}[\mathbb{E}_{\chi_b} f_b(\mathcal{D}_k(f_b + \chi_b))].$$

Here we denote $\mathbb{E}_{\mathcal{A}}$ to be expectation over the randomness of the algorithm $\mathcal{A}$ and $\mathbb{E}_{\chi_b}$ to be over the stochasticity of the the noise distribution $\chi_b$. Hence, we only have to analyze deterministic algorithms to establish the lower-bound. Further, since $\mathcal{D}_k$ is deterministic, for any *bijective* transformation $h$ which transforms the stochastic gradients, there exists a deterministic algorithm $\tilde{\mathcal{D}}$ such that $\tilde{\mathcal{D}}_k(h(f_b + \chi_b)) = \mathcal{D}_k(f_b + \chi_b)$. This implies that for any bijective transformation $h(\cdot)$ of the gradients:

$$\min_{\mathcal{D}} \mathbb{E}_{b\sim\mathcal{B}}[\mathbb{E}_{\chi_b} f_b(\mathcal{D}_k(f_b + \chi_b))] = \min_{\mathcal{D}} \mathbb{E}_{b\sim\mathcal{B}}\left[\mathbb{E}_{\chi_b} f_b(\tilde{\mathcal{D}}_k(h(f_b + \chi_b)))\right].$$

In this rest of the proof, we will try obtain a lower bound for the right hand side above.

We now describe our construction of the three quantities to be defined: the problem distribution $\mathcal{B}$, the noise distribution $\chi_b$, and the bijective mapping $h(\cdot)$. All of our definitions are parameterized by $\epsilon \in (0, 1/8]$ (which represents the desired target accuracy). We will pick $\epsilon$ to be a fixed constant which depends on the problem parameters (e.g. $k$) and should be thought of as being small.

- Problem distribution: $\mathcal{B}$ picks $b_0 = 2\epsilon\sigma/\mu$ or $b_1 = \epsilon\sigma/\mu$ at random i.e. $\nu \in \{0, 1\}$ is chosen by an unbiased coin toss and then we pick

$$b_\nu = (2 - \nu)\epsilon\frac{\sigma}{\mu}. \tag{38}$$

- Noise distribution: Define a constant $\gamma = 4\epsilon/\sigma$ and $p_\nu = (16\epsilon^2 - 8\nu\epsilon^2)$. Simple computations verify that $\gamma \in (0, 1/2]$ and that

$$p_\nu = (4 - 2\nu)\left(4\epsilon^2\right) \in (0, 1).$$

Then, for a given $\nu \in \{0, 1\}$ the stochastic gradient $g(x)$ is defined as

$$g(x) = \begin{cases} \mu x - \frac{1}{2\gamma} & \text{with prob. } p_\nu, \\ \mu x & \text{with prob. } 1 - p_\nu. \end{cases} \tag{39}$$

To see that we have the correct gradient in expectation verify that

$$\mathbb{E}[g(x)] = \mu x - \frac{p_\nu}{2\gamma} = \mu x - \mu b_\nu = \nabla f_{b_\nu}(x).$$

Next to bound the variance of $g(x)$. We see that

$$\mathbb{E}[|g(x) - \nabla f_b(x)|^2] \leq p_\nu\left(\frac{1}{2\gamma}\right)^2 + (1 - p_\nu)\mu^2 b_\nu^2 \leq \sigma^2.$$

Thus $g(x)$ defined in (39) satisfies condition (37).

- Bijective mapping: Note that here the only unknown variable is $\nu$ which only affects $p_\nu$. Thus the mapping is bijective as long as the *frequencies* of the events are preserved. Hence given a stochastic gradient $g(x_i)$ the mapping we use is:

$$h(g(x_i)) = \begin{cases} 0 & \text{if } g(x_i) = \mu x_i, \\ 1 & \text{otherwise.} \end{cases} \tag{40}$$

Given the definitions above, the output of algorithm $\mathcal{D}_k$ is thus simply a function of $k$ i.i.d. samples drawn from the Bernoulli distribution with parameter $p_\nu$ (which is denoted by $\text{Ber}(p_\nu)$). We now show how achieving a small optimization error implies being able to guess the value of $\nu$.

**Lemma E.1.** *Suppose we are given problem and noise distributions defined as in* (38) *and* (39)*, and an bijective mapping $h(\cdot)$ as in* (40)*. Further suppose that there is a deterministic algorithm $\mathcal{D}_k$ whose output after processing $k$ stochastic gradients satisfies*

$$\mathbb{E}_{b\sim\mathcal{B}}[\mathbb{E}_{\chi_b} f_b(\mathcal{D}_k(h(f_b + \chi_b)))] < \frac{\epsilon^2\sigma^2}{64\mu}.$$

*Then, there exists a deterministic function $\tilde{\mathcal{D}}_k$ which given $k$ independent samples of $\text{Ber}(p_\nu)$ outputs $\nu' = \tilde{\mathcal{D}}_k(\text{Ber}(p_\nu)) \in \{0, 1\}$ such that*

$$\Pr\left[\tilde{\mathcal{D}}_k(\text{Ber}(p_\nu)) = \nu\right] \geq \frac{3}{4}.$$

*Proof.* Suppose that we are given access to $k$ samples of $\text{Ber}(p_\nu)$. Use these $k$ samples as the input $h(f_b + \chi_b)$) to the procedure $\mathcal{D}_k$ (this is valid as previously discussed), and let the output of $\mathcal{D}_k$ be $x_k^{(\nu)}$. The assumption in the lemma states that

$$\mathbb{E}_\nu\left[\mathbb{E}_{\chi_b}\frac{\mu}{2}|x_k^{(\nu)} - b_\nu|^2\right] < \frac{\epsilon^2\sigma^2}{64\mu}, \text{ which implies that } \mathbb{E}_{\chi_b}|x_k^{(\nu)} - b_\nu|^2 < \frac{\epsilon^2\sigma^2}{16\mu^2} \text{ almost surely.}$$

Then, using Markov's inequality (and then taking square-roots on both sides) gives

$$\Pr\left[|x_k^{(\nu)} - b_\nu| \geq \frac{\epsilon\sigma}{2\mu}\right] \leq \frac{1}{4}.$$

Consider a simple procedure $\tilde{\mathcal{D}}_k$ which outputs $\nu' = 0$ if $x_k^{(\nu)} \geq \frac{3\epsilon\sigma}{2\mu}$, and $\nu' = 1$ otherwise. Recall that $|b_0 - b_1| = \epsilon\sigma/\mu$ with $b_0 = 2\epsilon\sigma/\mu$ and $b_1 = \epsilon\sigma/\mu$. With probability $\frac{3}{4}$, $|x_k^{(\nu)} - b_\nu| < \frac{\epsilon}{2}\sigma/\mu$ and hence the output $\nu'$ is correct. $\qquad\square$

Lemma E.1 shows that if the optimization error of $\mathcal{D}_k$ is small, there exists a procedure $\tilde{\mathcal{D}}_k$ which distinguishes between the Bernoulli distributions with parameters $p_0$ and $p_1$ using $k$ samples. To argue that the optimization error is large, one simply has to argue that a large number of samples are required to distinguish between $\text{Ber}(p_0)$ and $\text{Ber}(p_1)$.

**Lemma E.2.** *For any deterministic procedure $\tilde{\mathcal{D}}_k(\text{Ber}(p_\nu))$ which processes $k$ samples of $\text{Ber}(p_\nu)$ and outputs $\nu'$*

$$\Pr[\nu' = \nu] \leq \frac{1}{2} + \sqrt{k(4\epsilon)^2}.$$

*Proof.* Here it would be convenient to make the dependence on the samples explicitly. Denote $\mathbf{s}_k^{(\nu)} = \left(s_1^{(\nu)}\right),\dots,s_k^{(\nu)} \in \{0,1\}^k$ to be the $k$ samples drawn from $\text{Ber}(p_\nu)$ and denote the output as $\nu' = \tilde{\mathcal{D}}(\mathbf{s}_k^{(\nu)})$. With some slight abuse of notation where we use the same symbols to denote the realization and their distributions, we have:

$$\Pr\left[\tilde{\mathcal{D}}(\mathbf{s}_k^{(\nu)}) = \nu\right] = \frac{1}{2}\Pr\left[\tilde{\mathcal{D}}(\mathbf{s}_k^{(1)}) = 1\right] + \frac{1}{2}\Pr\left[\tilde{\mathcal{D}}(\mathbf{s}_k^{(0)}) = 0\right] = \frac{1}{2} + \frac{1}{2}\mathbb{E}\left[\tilde{\mathcal{D}}(\mathbf{s}_k^{(1)}) - \tilde{\mathcal{D}}(\mathbf{s}_k^{(0)})\right].$$

Next using Pinsker's inequality we can upper bound the right hand side as:

$$\mathbb{E}\left[\tilde{\mathcal{D}}(\mathbf{s}_k^{(1)}) - \tilde{\mathcal{D}}(\mathbf{s}_k^{(0)})\right] \leq \left|\tilde{\mathcal{D}}(\mathbf{s}_k^{(1)}) - \tilde{\mathcal{D}}(\mathbf{s}_k^{(0)})\right|_{TV} \leq \sqrt{\frac{1}{2}\text{KL}\left(\tilde{\mathcal{D}}\left(\mathbf{s}_k^{(1)}\right), \tilde{\mathcal{D}}\left(\mathbf{s}_k^{(0)}\right)\right)},$$

where $|\cdot|_{TV}$ denotes the total-variation distance and $\text{KL}(\cdot, \cdot)$ denotes the KL-divergence. Recall two properties of KL-divergence: i) for a product measures defined over the same measurable space $(p_1,\dots,p_k)$ and $(q_1,\dots,q_k)$,

$$\text{KL}((p_1,\dots,p_k),(q_1,\dots,q_k)) = \sum_{i=1}^k \text{KL}(p_i, q_i),$$

and ii) for any deterministic function $\tilde{\mathcal{D}}$,

$$\text{KL}(p, q) \geq \text{KL}(\tilde{\mathcal{D}}(p), \tilde{\mathcal{D}}(q)).$$

Thus, we can simplify as

$$\begin{aligned}
\Pr\left[\tilde{\mathcal{D}}(\mathbf{s}_k^{(\nu)}) = \nu\right] &\leq \frac{1}{2} + \sqrt{\frac{k}{8}\text{KL}(\text{Ber}(p_1), \text{Ber}(p_0))} \\
&\leq \frac{1}{2} + \sqrt{\frac{k}{8}\frac{(p_0 - p_1)^2}{p_0(1 - p_0)}} \\
&\leq \frac{1}{2} + \sqrt{k(4\epsilon)^2}
\end{aligned}$$

$\qquad\square$

If we pick $\epsilon$ to be

$$\epsilon = \frac{1}{16k^{1/2}}\,,$$

we have that

$$\frac{1}{2} + \sqrt{k(4\epsilon)^2} = \frac{3}{4}\,.$$

Given Lemmas E.1 and E.2, this implies that for the above choice of $\epsilon$,

$$\mathbb{E}_{b \sim \mathcal{B}}[\mathbb{E}_{\chi_b} f_b(\mathcal{D}_k(h(f_b + \chi_b)))] \geq \epsilon^2 \frac{\sigma^2}{64\mu} = \frac{\sigma^2}{\mu 2^{14} k}\,.$$

$\square$

## F  Proof of Proposition 6.1

This proof is almost the same as the proof in Appendix E, except that we use the result from Theorem 3 of [3] instead of from [8].

## G  Additional Experiments on CIFAR-10

For the CIFAR-10 dataset, we conduct more thorough experiments under the same setup as in Section 7. We further experiment with various degrees of data heterogeneity to explore the effect of data heterogeneity on algorithm performance.

### G.1  Improving the experiments in Section 7

Using the same setup as that of Section 7, we conduct more comprehensive experiments on CIFAR-10. Specifically, we perform hyperparameter search of the inital learning rate over the grid $\{0.01, 0.02, \cdots, 0.17\}$ to make sure the optimal learning rate falls in the middle of the interval. We add two more baselines, i.e., Federated Averaging that samples 10 and 25 devices for each global update, denoted as `FedAvg`$(S = 10)$ and `FedAvg`$(S = 25)$ respectively. We increase the total number of rounds to 2000. We plot the average training loss and test accuracy curves over 5 random seeds with error bars in Figure 3. Note that we smooth the curves by simple moving average for better visualization. From the training results, we can reach the same conclusions as in Section 7. Besides, we observe more volatile training loss and test accuracy for `FedAvg`$(S = 10)$ and `FedAvg`$(S = 25)$ due to increased noise induced by the device sampling.

### G.2  Results on various degrees of data heterogeneity

To further explore how levels of data heterogeneity influence algorithm performance, we use the Dirichlet allocation proposed in [15] with parameter $\alpha$ to partition the dataset and experiment on data generated with different $\alpha$'s. Specifically, a smaller $\alpha$ indicates a higher level of heterogeneity. We run all algorithms with $\alpha \in \{0.01, 0.05, 0.1, 0.2\}$ for 2000 rounds. For each combination of the algorithm and the data heterogeneity level, we search for the optimal initial learning rate over the grid $\{0.01, 0.02, \cdots, 0.17\}$ and repeat the experiment on 5 random seeds. We artificially set $p_i$ by the following rule: if the $i$-th device holds more samples of class $j$ than all other classes, its participation probability is set as $p_i = p_{\min}j/9 + (1 - p_{\min})$, where $p_{\min}$ controls the lower bound of participation probabilities. Throughout all experiments in this subsection, $p_{\min}$ is set as 0.1. The model type and other hyperparameters are the same as those in Section 7. We report the average test accuracy with standard deviation and the gap between the average test accuracy achieved by the baseline algorithm and that achieved by `MIFA` in Table 1.

We observe that the performance of all algorithms degrades as the degree of data heterogeneity increases. For all levels of data heterogeneity, `MIFA` is consistently competitive to `FedAvg` with importance sampling and outperforms all other baselines. The advantage of `MIFA` over biased `FedAvg` is more significant under higher levels of data heterogeneity ($\alpha \leq 0.1$).

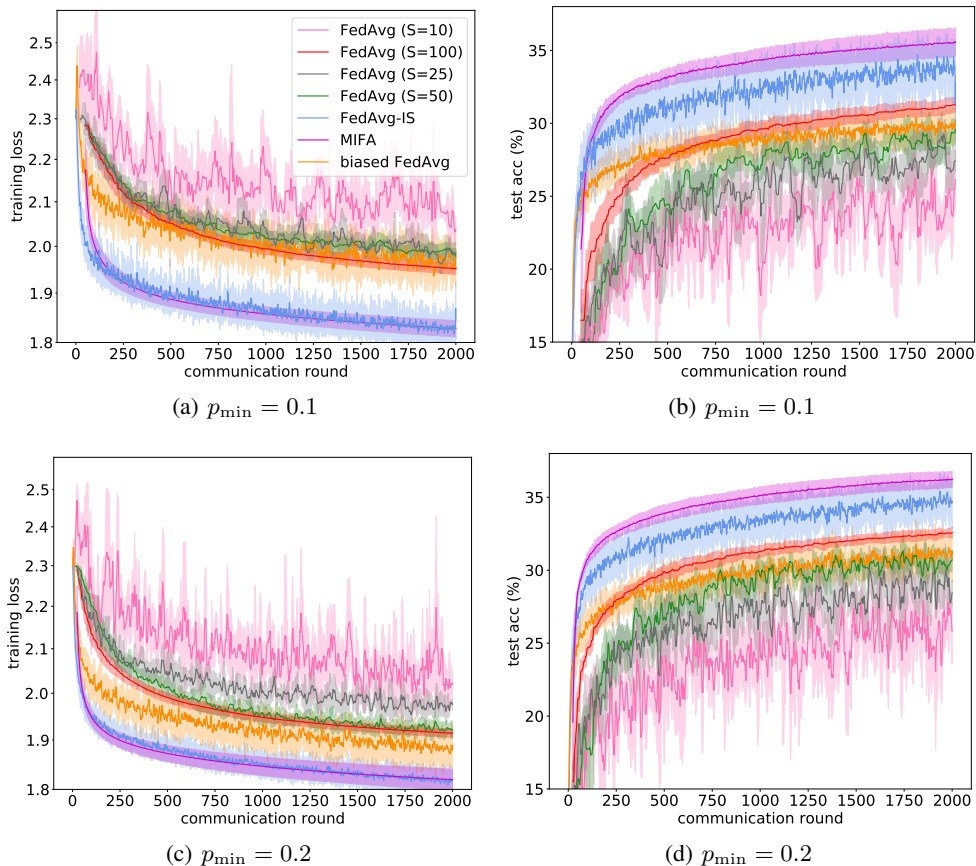

Figure 3: Training losses and test accuracies on non-iid CIFAR-10. Fig. 3(a) and 3(b): minimum participation rate set as 0.1. Fig. 3(c)–3(d): minimum participation rate set as 0.2. FedAvg ($S = 10$), FedAvg ($S = 25$), FedAvg ($S = 50$) and FedAvg ($S = 100$) refer to FedAvg with device sampling that samples $S$ devices for each global update. FedAvg-IS is short for FedAvg with importance sampling, which requires knowledge of the participation probabilities.

Table 1: Average test accuracy under different levels of data heterogeneity. The standard deviation across different random seeds is reported in the parenthesis. A smaller $\alpha$ indicates a higher level of data heterogeneity. The 'gap' column reports the difference between the average accuracy achieved by baseline algorithms and that achieved by MIFA.

| Algorithm | $\alpha = 0.01$ | | $\alpha = 0.05$ | |
|---|---|---|---|---|
| | acc.(stdev.) | gap | acc.(stdev.) | gap |
| MIFA | 33.33(0.39) | – | 36.98(0.44) | – |
| FedAvg-IS | 32.40(0.85) | $-0.92$ | 34.15(3.07) | $-2.84$ |
| Biased FedAvg | 28.04(0.71) | $-5.29$ | 31.51(0.71) | $-5.47$ |
| FedAvg ($S = 100$) | 27.63(0.27) | $-5.70$ | 31.14(0.45) | $-5.84$ |
| FedAvg($S = 50$) | 27.01(0.45) | $-6.31$ | 29.97(0.52) | $-7.02$ |
| FedAvg($S = 25$) | 23.09(1.77) | $-10.24$ | 29.15(2.01) | $-7.84$ |
| FedAvg($S = 10$) | 20.71(2.93) | $-12.62$ | 26.46(1.83) | $-10.52$ |
| Algorithm | $\alpha = 0.1$ | | $\alpha = 0.2$ | |
| | acc.(stdev.) | gap | acc.(stdev.) | gap |
| MIFA | 40.04(0.27) | – | 42.30(1.58) | – |
| FedAvg-IS | 39.50(0.40) | $-0.54$ | 42.71(0.68) | $+0.41$ |
| Biased FedAvg | 35.22(0.34) | $-4.82$ | 40.49(0.99) | $-1.81$ |
| FedAvg ($S = 100$) | 32.73(0.40) | $-7.32$ | 35.49(0.57) | $-6.81$ |
| FedAvg($S = 50$) | 32.45(0.84) | $-7.60$ | 34.89(0.67) | $-7.42$ |
| FedAvg($S = 25$) | 31.89(1.36) | $-8.15$ | 34.64(1.22) | $-7.67$ |
| FedAvg($S = 10$) | 28.39(1.18) | $-11.65$ | 33.93(2.09) | $-8.38$ |