# OpenReview forum: "Fast Federated Learning in the Presence of Arbitrary Device Unavailability"
_NeurIPS.cc/2021/Conference — NeurIPS 2021 Poster_

### Official Review · Reviewer_LFYH · 2021-07-07

**Rating:** 7
**Confidence:** 4

**Summary:**

The paper tries to overcome the challenge of federated learning that devices may drop out of the training process arbitrarily ( which doesn't follow any known or unknown distribution). This paper proposes the Memory-augmented Impatient Federated Averaging (MIFA) algorithm and provides its theoretical analysis under arbitrary device unavailability. Experiments have been done to show the advantages of the proposed method.

**Limitations And Societal Impact:**

Given assumption 4, we have $\tau_{max, T} =O(T)$, So, in Theorem 5.1, there is a constant term $A_1/\mu^2$. It means the MIFA converges to the optimal solution with a constant error term, right? Similarly,  Proposition 5.1 has the same question.

In the experiment section, the initial learning rate only uses 0.1 and the weight delay only checks 0.001.  So, the comparison of performance among all methods may be unfaired.  It is better to check other learning rates and weight delays, and select the best performance for each method.

In Figure 2(a-d), FedAvg-IS always gives the best performance in terms of communication round, rather than the proposed MIFA method. Instead of using communication round, the authors may want to show running time, since the intuitive advantage of the proposed method MIFA (or "asynchronous method") is the better running time since it doesn't need to wait for unavailable devices.





**Main Review:**

Originality:  The assumption on the device availability ( similar to Assumption 4 and 8) has been widely used in the theoretical analysis for asynchronous algorithms. It is new to see the assumption is used in FL to analyze arbitrary device unavailability.

Quality:  The submission technically sound and the claims are well supported by theoretical analysis and experimental results.  Theoretical analyses with convex and nonconvex functions are provided. This is a complete piece of work.


Clarity: The submission is clearly written and well organized.

Significance:  The analysis with assumption 4 is new and significant to federated learning.

**Time Spent Reviewing:**

N/A

---

> ### Author Response · Authors · 2021-08-10
> **Response to Reviewer LFYH**
>
> We sincerely thank the reviewer for the valuable review.
>
> **Q1**: Given assumption 4, we have $\tau_{\max, T} = O(T)$. So, in Theorem 5.1, there is a constant term. It means that MIFA converges to the optimal solution with a constant error term, right? Similarly, Proposition 5.1 has the same question.
>
> **A1**: Theorem 5.1 and Proposition 5.1 hold as long as $\tau_{\max, T}=O(T)$. If  $\tau_{\max, T}$ grows linearly in $T$, MIFA indeed converges to within a constant error of the optimum. Our theorem guarantees MIFA to converge when $\tau_{\max, T} = o(T)$. This is also discussed in Lines 201-205. In fact, this requirement on $\tau_{\max, T}$ is mild since it allows an unbounded number of inactive rounds while existing results on asynchronous updates mostly assume a bounded number of inactive rounds. (e.g., [1, 2, 3]).
>
> **Q2**: In the experiment section, the initial learning rate only uses 0.1 and the weight decay only checks 0.001. So, the comparison of performance among all methods may be unfair. It is better to check other learning rates and weight decays, and select the best performance for each method.
>
> **A2**: Thank you for your valuable suggestion. We have indeed tried different hyperparameters and the results remained almost unchanged. We agree with the reviewer that performing a hyper-parameter search for each algorithm is better for comparing algorithms. We will conduct more comprehensive experiments in our revision.
>
> **Q3**: In Figure 2(a-d), FedAvg-IS always gives the best performance in terms of communication round, rather than the proposed MIFA method.
>
> **A3**: We remark that FedAvg-IS is not applicable in realistic settings since it requires the knowledge of the device participation probabilities. In our experiment setting, FedAvg-IS serves as an oracle algorithm to be targeted. By contrast, MIFA is agnostic to the participation pattern and its performance is still comparable to FedAvg-IS, which shows its superiority.
>
> **Q4**: Instead of using communication round, the authors may want to show running time, since the intuitive advantage of the proposed method MIFA (or "asynchronous method") is the better running time since it doesn't need to wait for unavailable devices.
>
> **A4**: In our paper, one round can be viewed as a unit time. Therefore, running time and communication rounds refer to the same metric. Indeed, as the reviewer indicates, MIFA achieves better running time/communication rounds because it avoids waiting for unavailable devices and corrects the biases in the delayed updates.
>
> [1] Agarwal, Alekh, and John C. Duchi. "Distributed delayed stochastic optimization." 2012 IEEE 51st IEEE Conference on Decision and Control (CDC). IEEE, 2012.
>
> [2] Aytekin, Arda, Hamid Reza Feyzmahdavian, and Mikael Johansson. "Analysis and implementation of an asynchronous optimization algorithm for the parameter server." arXiv preprint arXiv:1610.05507 (2016).
>
> [3] Basu, Debraj, et al. "Qsparse-local-SGD: Distributed SGD with quantization, sparsification, and local computations." arXiv preprint arXiv:1906.02367 (2019).

---

### Official Review · Reviewer_gyQJ · 2021-07-13

**Rating:** 6
**Confidence:** 3

**Summary:**

The article introduces a new algorithm called MIFA (Memory-augmented Impatient Federated Averaging) for Federated learning with arbitrary device participation. They derive theoretical guarantees in terms of average convergence of the algorithm, which are proved minimax for strongly and non strongly convex smooth losses. Numerical experiments illustrate the empirical behaviour of the method in comparison to a baseline FL algorithm.

**Limitations And Societal Impact:**

The authors adequately addressed the limitations and potential negative societal impact of their work.

**Main Review:**

The paper is sound, well written and clear, the theoretical results are also precise.  My one main concern is about the novelty of the setting, as detailed below.
- The authors claim that existing methods only consider delayed responses of devices rather than full drop out, but I am not sure it is true. There are several FL algorithms, more recent thatn FedAvg, which handle partial participation of devices, including with missing devices (see, e.g. [1] and [2]), and I am not sure whether the setting you describe is really new. Please clarify.
- It is a bit frustrating that the obtained theoretical rates are not compared qualitatively to existing work.
- It is also a bit frustrating that the method is empirically challenged only with FedAvg, when many algorithms have been proposed since then, including with partial device participation (see references [1] and [2]).


[1] Eduard Gorbunov, Konstantin Burlachenko, Zhize Li, Peter Richtarik (2021). MARINA: Faster Non-Convex Distributed Learning with Compression. ICML.
[2] [1]Philippenko, C. and Dieuleveut, A., “Bidirectional compression in heterogeneous settings for distributed or federated learning with partial participation: tight convergence guarantees”, <i>arXiv e-prints</i>, 2020.


**Time Spent Reviewing:**

2

---

> ### Author Response · Authors · 2021-08-10
> **Response to Reviewer gyQJ**
>
> We sincerely thank the reviewer for acknowledging our result as sound and clear. We also thank the reviewer for pointing out some missing related works. Compared to [1] and [2], our setup is indeed new in significant aspects that motivated our algorithm design.  In our revision, we will add the missing references and discuss the differences. Details are as follows.
>
> **Q1**: The authors claim that existing methods only consider delayed responses of devices rather than full drop out, but I am not sure it is true. There are several FL algorithms, more recent than FedAvg, which handle partial participation of devices, including with missing devices (see, e.g. [1] and [2]), and I am not sure whether the setting you describe is really new. Please clarify.
>
> **A1**: We are sorry for the confusion but we did not claim that existing methods only consider delayed responses. In fact, we have discussed in detail how previous works model missing responses/partial participation. Please refer to Section 1 (Lines 37 to 49) and Section 2 (Lines 76 to 82).
> Our setup allows arbitrary participation patterns which can be distribution-free and even adversarial. In contrast, the settings of [1] and [2] both make assumptions on the distribution of device participation. Specifically, [1] assumes that all devices become active at each round independently with a fixed probability, and [2] assumes that the index set of participating devices contains  $r$ i.i.d. samples from the distribution over $\{1, \cdots, n\}$. [1] and [2] provide insightful solutions via communication compression, whereas we focus on dealing with unavailability patterns. It is possible that our method can benefit from techniques in [1, 2]. Thank you for pointing to the missing references. We will add the discussion in our revision.
>
> **Q2**: It is a bit frustrating that the obtained theoretical rates are not compared qualitatively to existing work.
>
> **A2**: To our best knowledge, there are no existing theoretical results in the exact same setup as ours. For some special cases of our setup, we have compared our results with existing works in Remark 5.1 and Remark 6.1, and the comparisons show that our bounds are correct and tight.
>
> **Q3**: It is also a bit frustrating that the method is empirically challenged only with FedAvg, when many algorithms have been proposed since then, including with partial device participation (see references [1] and [2]).
>
> **A3**: The focus of our work is to achieve fast convergence under a practical formulation of device participation, which allows arbitrary device participation. To our best knowledge, there are no existing algorithms targeting this setting. MIFA provides a new direction of improvement for existing algorithms in this setting. That is, we can utilize the stale updates of inactive devices to accelerate convergence and correct bias. By contrast, [1] and [2] mainly focus on reducing communication costs by compression, which is another dimension of improvement.
>
> [1] Eduard Gorbunov, Konstantin Burlachenko, Zhize Li, Peter Richtarik (2021). MARINA: Faster Non-Convex Distributed Learning with Compression. ICML.
>
> [2] Philippenko, C. and Dieuleveut, A., “Bidirectional compression in heterogeneous settings for distributed or federated learning with partial participation: tight convergence guarantees”, <i>arXiv e-prints</i>, 2020.

---

### Official Review · Reviewer_JW3q · 2021-07-16

**Rating:** 6
**Confidence:** 3

**Summary:**

This work studies federated learning setup where clients might participate and drop out from the training arbitrarily. Broadly speaking, this is partial participation setup for FL, where there is no direct control on device availability. For such a setup, this paper designs MIFA algorithm, which stores latest gradients from devices and does not wait for strugglers during the training. The method is analyzed for strongly convex and non-convex cases with an additional (weak) assumption on device participation.

**Limitations And Societal Impact:**

Yes

**Main Review:**

Partial participation (PP) is one of the characterizing aspects of federated learning (FL) as clients participate during the training only when they are available. Prior works usually model PP by assuming that a small portion of the devices (generated randomly) are available for the training. However, this ignores possibilities that some (or most and maybe even all) selected devices will be off, and in such case the ongoing communication round is failed. To overcome device unavailability, this work designs MIFA algorithm, which tracks the latest gradient information from each device and, in case of strugglers, updates the global model using stale gradient information of unavailable clients. This approach of handling device unavailability seems interesting.

Any explanation on why $b = 40(L/\mu)^{1.5}$ form is needed in Assumption 4 ? While the number of inactive rounds grow as ${\cal O}(t)$, it seems to grow slowly because of the value $b$.

The discussion in lines 252-262 are not completely fair comparison against FedAvg/SCAFFOLD, since $\frac{S}{N}\frac{1}{p_{\min}}$ iterations for FedAvg/SCAFFOLD are an extra waiting time, but for MIFA it is a normal iteration with communication rounds. In this context, are those waiting rounds counted in the experiments with FedAvg ?

Why the case that all devices are active is referred to as degenerated in line 66 ?


**Time Spent Reviewing:**

4

---

> ### Author Response · Authors · 2021-08-10
> **Response to Reviewer JW3q**
>
> We sincerely thank the reviewer for acknowledging that we provided an interesting solution. We will edit and clarify the potentially confusing statements as suggested.
>
> **Q1**: Why $b=40(L/\mu)^{1.5}$ form is needed in A4? While the number of inactive rounds grow as $O(t)$, it seems to grow slowly because of the value .
>
> **A1**: Although $b$ seems to be large, Assumption 4 is a mild assumption, since it allows arbitrary participation patterns and an unbounded number of inactive rounds that grows as $O(t)$. In contrast, existing results on asynchronous updates mostly assume a bounded number of inactive rounds. (e.g., [1, 2, 3]). The intuition behind the form of $b$ is that the device unavailability incurs errors in the gradient estimation, and a large condition number $L/\mu$ means that the problem is sensitive to the gradient errors. The more sensitive the problem is to gradient errors, the smaller number of inactive rounds it allows.
>
> **Q2.1**: The discussion in lines 252-262 is not completely fair comparison against FedAvg/SCAFFOLD, since $\frac{S}{N}\frac{1}{p_{\min}}$ iterations for FedAvg/SCAFFOLD are an extra waiting time, but for MIFA it is a normal iteration with communication rounds.
>
> **A2.1**: In realistic Federated Learning settings, the bottleneck is device unavailablity rather than  local computation. As pointed out by [4], edge devices (e.g., smartphones) have relatively small local datasets and fast processors (e.g., GPUs), but each device only participates in a small number of rounds per day. Therefore, the time spent waiting for some devices to participate is generally longer than the local computation time. Our theoretical analysis shows that the better performance of MIFA stems from the fact that MIFA does not need to spend extra waiting time and the updates are unbiased.
>
> **Q2.2**: In this context, are those waiting rounds counted in the experiments with FedAvg ?
>
> **A2.2**: Yes, in experiments with FedAvg, the waiting rounds are counted. For a fair comparison, the first few waiting rounds for MIFA to collect responses from all devices are also counted.
>
> **Q3**: Why is the case that all devices are active referred to as degenerated in line 66 ?
>
> **A3**: Thank you for pointing out this confusion and we will modify the wording in our revision. The case where all devices are active is a special case of our setup (corresponds to  $\tau(t,i)=0$ for all $t$  and $i$). Our results recover the optimal rate in this special case.
>
> [1] Agarwal, Alekh, and John C. Duchi. "Distributed delayed stochastic optimization." 2012 IEEE 51st IEEE Conference on Decision and Control (CDC). IEEE, 2012.
>
> [2] Aytekin, Arda, Hamid Reza Feyzmahdavian, and Mikael Johansson. "Analysis and implementation of an asynchronous optimization algorithm for the parameter server." arXiv preprint arXiv:1610.05507 (2016).
>
> [3] Basu, Debraj, et al. "Qsparse-local-SGD: Distributed SGD with quantization, sparsification, and local computations." arXiv preprint arXiv:1906.02367 (2019).
>
> [4] McMahan, Brendan, et al. "Communication-efficient learning of deep networks from decentralized data." Artificial intelligence and statistics. PMLR, 2017.

---

### Official Review · Reviewer_nrJT · 2021-08-01

**Rating:** 6
**Confidence:** 3

**Summary:**

For a heterogeneous distributed optimization setting in which data of different characteristics is available to different workers, the paper proposes the usage of the latest reported updates from workers when unavailable. As claimed in the paper, this amends the bias introduced when utilizing only data from available workers (as in the previous state of the art methods), improves the convergence bounds for convex and non-convex smooth optimization settings, and does not require up-front knowledge of the workers-unavailability statistics.

**Limitations And Societal Impact:**

yes

**Main Review:**

Well written paper, clearly presenting the state of the art and the problems addresses by the proposed approach. The proposed algorithm nevertheless, although claimed to be adaptive, requires upfront knowledge of the smoothness and strong convexity ($L$ and $\mu$ respectively), or searching for the correct values (i.e., to set the learning rate). Finally, although illustrating the superiority of the proposed methods over state of the art, the experimental section seems somewhat limited. To gain more confidence in the method and the related bounds, such experiments should have been run over several meta-parameters and unavailability schemes, datasets and model types.

A few comments and questions:
The results presented in the introduction (lines 63, 64) use notation yet to be defined.
The notion of a communication round (as first mentioned in the introduction - line 48) is not reflected in Algorithm 1. E.g., how is the set of active devices determined? is there a watchdog/timeout to be employed? when is a 'round' declared? what if an update arrives later, out of the round (that is, a combination of the delayed setup and 'our setup' of Figure 1)?
Typos? - founded (line 119), till (line 124).



**Time Spent Reviewing:**

3

---

> ### Author Response · Authors · 2021-08-10
> **Response to Reviewer nrJT**
>
> We sincerely thank the reviewer for pointing out that our work provides a clear solution to federated learning with unknown worker unavailability patterns. We address the reviewer's concern on the definition of adaptivity and communication rounds as follows.
>
> **Q1**: Although the algorithm is claimed to be adaptive, upfront knowledge of   and  searching for the correct values (i.e., to set the learning rate) are still required.
>
> **A1**: We are sorry for the confusion. What we mean by “adapt to” in Line 52 is that our algorithm can handle various participation patterns without knowing the distribution statistics. We will clarify in our revision  that we did not claim our algorithm to be adaptive with respect to the learning rate. We remark that setting the learning rate in terms of $L$ and $\mu$ is standard in the theoretical analyses, e.g., classical analysis showing that gradient descent requires a learning rate smaller than $O(1/L)$ to converge for $L$-smooth functions.
>
> **Q2**: Although illustrating the superiority of the proposed methods over state of the art, the experimental section seems somewhat limited. To gain more confidence in the method and the related bounds, such experiments should have been run over several meta-parameters and unavailability schemes, datasets and model types.
>
> **A2**: Thank you for your suggestions. In the paper, we have experimented with different unavailability schemes (by varying the minimum participation probability), datasets (MNIST and CIFAR10), model types (linear model and LeNet-5). We also tested a few sets of hyper-parameters, and the results remained almost unchanged. We agree with the reviewer that the experimental section can be further strengthened, and will conduct more comprehensive experiments in our revision.
>
> **Q3**: The results presented in the introduction (lines 63, 64) use notation yet to be defined.
>
> **A3**: Thank you for pointing out the issue. We will add the interpretations in our revision. Here  $N, K, T$ stand for the number of devices, local updates and communication rounds, respectively. $\bar{\tau}_T$ and $\bar{\nu}$ characterize how actively devices participate in training.
>
> **Q4.1**: About the communication round. The notion of a communication round (as first mentioned in the introduction - line 48) is not reflected in Algorithm 1. E.g., how is the set of active devices determined? Is there a watchdog/timeout to be employed? When is a 'round' declared?
>
> **A4.1**: In the statement of Algorithm 1, we abstract the problem and neglect the implementation details to make the intuition clear. As the reviewer pointed out, a watchdog/timeout mechanism is a possible and wise implementation to determine the set of active devices. Our setup is motivated by scenarios where communication is much slower than computation. For example, the server may want to publish a new model at a specific time every day. In this case, a new round is declared when the server finishes updating the global model and publishes a new one. The set of active devices consists of the devices that send effective updates to the server during this round. We remark that the set of active devices is determined by the environment and the server has no control over it.
>
> **Q4.2**: What if an update arrives later, out of the round (that is, a combination of the delayed setup and 'our setup' of Figure 1)?
>
> **A4.2**: Our analysis does not take advantage of the delayed responses. As pointed out by [1], these updates only make up a tiny fraction in practical systems when the time interval of two consecutive updates is significantly longer than the computation time and the communication delay (e.g., when the communication is on a daily basis). Our theoretical results can likely be extended to the case where the delayed responses are used. We leave this problem as future work.
>
> **Q5**: Typos: founded (line 119), till (line 124).
>
> **A5**: Thank you a lot for pointing out the typos. We will correct them in our revision.
>
> [1] Bonawitz, Keith, et al. "Towards federated learning at scale: System design." arXiv preprint arXiv:1902.01046 (2019).

---

### Decision · Program_Chairs · 2021-09-27

**Decision:**

Accept (Poster)

**Comment:**

The authors focus on addresses some new challenges in Federated Learning. In particular they focus on the effect of straggler devices that may dropout which severely affect convergence of popular FL baselines. To ovecome this challenge they propose a new algorithm called Memory-augmented Impatient Federated Averaging (MIFA). The authors claim that their approach avoids the latency of inactive devices and can effectively correct for such gradient bias by using memorized updates from devices. The authors also show that this approaches achieve a minimax optimal convergence rate for non-i.i.d. data in certain cases. They also corroborate their theoretical findings with a variety of experiments. Most reviewers thought the paper was well written with a clearly presented approach to tackle a well motivated problem. The reviewers raised a variety of technical concerns some of which were addressed during the rebuttal. The conclusion of the discussion was that while the paper is not extremely novel/original and falls into the marginal category is a solid contribution and merits acceptance. I agree with this assessment and recommend acceptance with the condition that the authors clearly address the valid concerns raised by the reviewers.